# ON THE IMPORTANCE OF IN-DISTRIBUTION CLASS PRIOR FOR OUT-OF-DISTRIBUTION DETECTION

## ABSTRACT

Given a pre-trained *in-distribution* (ID) model, the task of inference-time *out-of-distribution* (OOD) detection methods aims to recognize upcoming OOD data in inference time. However, some representative methods share an unproven assumption that the probability that OOD data belong to every ID class should be the same, i.e., probabilities that OOD data belong to ID classes *form a uniform distribution*. In this paper, we theoretically and empirically show that this assumption makes these methods incapable of recognizing OOD data when the ID model is trained with *class-imbalanced data*. Fortunately, by analyzing the causal relations between ID/OOD classes and features, we identify several common scenarios where probabilities that OOD data belong to ID classes should be the ID-class-prior distribution. Based on the above finding, we propose two effective strategies to modify previous inference-time OOD detection methods: 1) if they explicitly use the uniform distribution, we can *replace* the uniform distribution with the ID-class-prior distribution; 2) otherwise, we can *reweight* their scores according to the similarity between the ID-class-prior distribution and the softmax outputs of the pre-trained model. Extensive experiments show that both strategies significantly improve the accuracy of recognizing OOD data when the ID model is pre-trained with imbalanced data. As a highlight, when evaluating on the iNaturalist dataset, our method can achieve ∼36% increase on AUROC and ∼61% decrease on FPR95, compared with the original Energy method, reflecting the importance of ID-class prior in the OOD detection, which *lights up a new road* to study this problem.

## 1 INTRODUCTION

How to reliably deploy machine learning models into real-world scenarios has been attracting more and more attention (Huang et al., 2021; Liang et al., 2018; Liu et al., 2020). In real-world scenarios, test data usually contain known and unknown classes (Hendrycks & Gimpel, 2017). We expect the deployed model to eliminate the interference of unknown classes while classifying known classes well. Nevertheless, current models tend to be overconfident in the unknown classes (Nguyen et al., 2015), and thus confusing known and unknown classes, which increases the risk of deploying these models in the real world. Especially if the scenarios are life-critical (e.g., car-driving scenarios), we cannot take the risks of deploying unreliable models in them. This motivates researchers to study *out-of-distribution* (OOD) detection, where we need to identify unknown classes (i.e., OOD classes) and classify known classes (i.e., *in-distribution* (ID) classes) well at the same time (Hendrycks & Gimpel, 2017; Hendrycks et al., 2019).

In the OOD detection, a well-known branch is to develop the inference-time/post hoc OOD detection methods (Huang et al., 2021; Liang et al., 2018; Liu et al., 2020; Hendrycks & Gimpel, 2017; Lee et al., 2018b; Sun et al., 2021), where we are given a pre-trained ID model and then aim to recognize upcoming OOD data well. The key advantage of inference-time OOD detection methods is that the classification performance on ID data will be unaffected since we only use the ID model instead of changing it. A general way to design a large-scale-friendly inference-time OOD detection method is to propose a score function by using the ID model's information. For example, *maximum softmax probability* (MSP) uses the ID model's outputs (Hendrycks & Gimpel, 2017), and GradNorm uses the ID model's gradients (Huang et al., 2021). If the score of a data point is smaller, then this data point is an OOD data point with a higher probability.

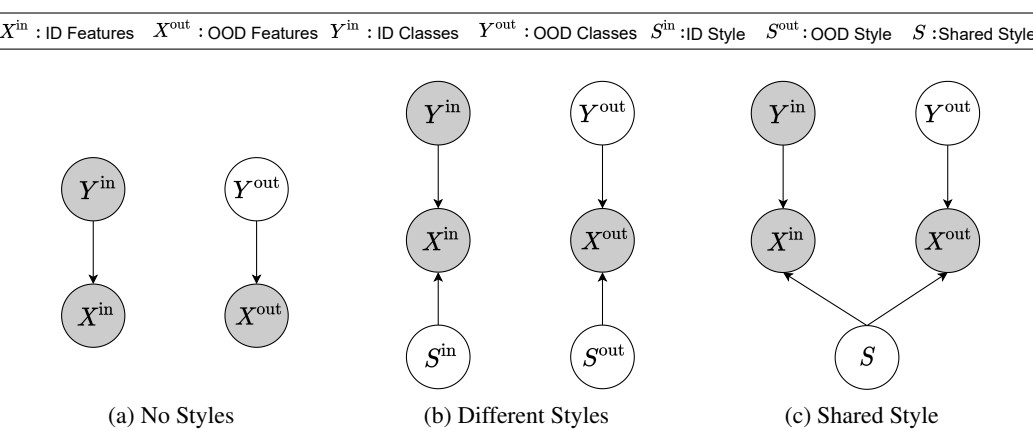

Figure 1: Three common causal graphs in OOD detection. Under these graphs, we prove that probabilities that an OOD data point $\mathbf{x}^{\mathrm{out}}$ belongs to ID classes should be the ID-class-prior distribution $\mathbb{P}_{Y^{\mathrm{in}}}$ (Theorem 1). However, some representative OOD detection methods (Huang et al., 2021; Hendrycks & Gimpel, 2017) assume such probabilities to be a uniform distribution $\mathbf{u}$ (e.g., GradNorm in Eq. 2). In this figure, each node represents a random variable, and gray ones indicate observable variables. $X$ stands for features, $Y$ stands for classes, and $S$ stands for styles. In the three graphs, features are generated by classes (i.e., $Y \to X$) (Gong et al., 2016; Stojanov et al., 2021) or generated by classes and styles (i.e., $Y \to X \leftarrow S$) (Yao et al., 2021). The three causal graphs broadly exist in our common datasets. For example, (a) corresponds to datasets consisting of sketch images, like ImageNet-Sketch (Wang et al., 2019) where ID classes could be cars and OOD classes could be animals; (b) and (c) correspond to datasets consisting of common images, like ImageNet (Deng et al., 2009) and MNIST (LeCun et al., 1998). In (b), the ID classes could be cars in ImageNet, and the OOD classes could be numbers in MNIST (different styles). In (c), the ID classes could be numbers in MNIST, and OOD classes could be classes in Fashion-MNIST (Xiao et al., 2017) (the same style). Through these graphs, it is clear that $Y^{\mathrm{in}} \perp\!\!\!\perp X^{\mathrm{out}}$, i.e., $Y^{\mathrm{in}}$ and $X^{\mathrm{out}}$ are independent.

However, some representative methods share *an unproven assumption*: the probability that an OOD data point $\mathbf{x}^{\mathrm{out}}$ belongs to each ID class $i$ is always the same. Namely, for any $\mathbf{x}^{\mathrm{out}}$, they *assume*

$$[\mathrm{Pr}(\mathbf{x}^{\mathrm{out}}\ \text{belongs to class } 1), \dots, \mathrm{Pr}(\mathbf{x}^{\mathrm{out}}\ \text{belongs to class } K)] = [1/K, \dots, 1/K]_{1 \times K} := \mathbf{u}, \quad (1)$$

where $\mathbf{u}$ is a uniform distribution and $K$ is the number of ID classes. Taking the GradNorm (Huang et al., 2021), a state-of-the-art OOD detection method, as an example[1], let $\mathbf{f}_{\boldsymbol{\Theta}}(\mathbf{x})$ be ID model's output of a data point $\mathbf{x}$, and the score function of GradNorm is

$$S_{\mathrm{GradNorm}}(\mathbf{f}_{\boldsymbol{\Theta}}, \mathbf{x}) = \left\| \frac{\partial D_{\mathrm{KL}}(\mathbf{u} \| \mathrm{softmax}(\mathbf{f}_{\boldsymbol{\Theta}}(\mathbf{x})))}{\partial \boldsymbol{\Theta}} \right\|_{L^1}, \quad (2)$$

where $\boldsymbol{\Theta}$ represents the ID model's parameters, $\mathrm{softmax}(\mathbf{f}_{\boldsymbol{\Theta}}(\mathbf{x}))$ is a vector consisting of predicted probabilities that $\mathbf{x}$ belongs to ID classes, and $D_{\mathrm{KL}}(\cdot \| \cdot)$ is the Kullback-Leibler divergence. It is clear that GradNorm considers $\mathbf{u}$ as a *reference distribution* to distinguish between ID and OOD data. If the divergence between $\mathrm{softmax}(\mathbf{f}_{\boldsymbol{\Theta}}(\mathbf{x}))$ and $\mathbf{u}$ is smaller, then $\mathbf{x}$ is an OOD data point with a higher probability. Nonetheless, since we do not have this assumption proven, we do not know whether it is correct. If not, the $\mathbf{u}$-based score functions (e.g., Eq. 2) are ill-defined because they cannot guarantee that the lowest score corresponds to the most OOD-ness data.

In this paper, we theoretically analyze the above assumption (i.e., Eq. 1) under three common causal graphs (Figure 1), and find that the above assumption holds only when the ID-class prior is $\mathbf{u}$, i.e., the ID model is trained with class-balanced data. In other cases, the reference distribution of OOD data *should be* the ID-class-prior distribution $\mathbb{P}_{Y^{\mathrm{in}}}$ (Theorem 1), i.e.,

$$[\mathrm{Pr}(\mathbf{x}^{\mathrm{out}}\ \text{belongs to class } 1), \dots, \mathrm{Pr}(\mathbf{x}^{\mathrm{out}}\ \text{belongs to class } K)] = \mathbb{P}_{Y^{\mathrm{in}}}. \quad (3)$$

Specifically, assume that we have $K$ classes in training data (i.e., ID data). Let $n_j$ be the number of samples in class $j$, then the total number of samples is $N = \sum_j^K n_j$. Thus, we have $\mathbb{P}_{Y^{\mathrm{in}}} = [n_1/N, n_2/N, ..., n_K/N]$.

---

[1]Note that, MSP also has this assumption, we will discuss it in Section 3.2.

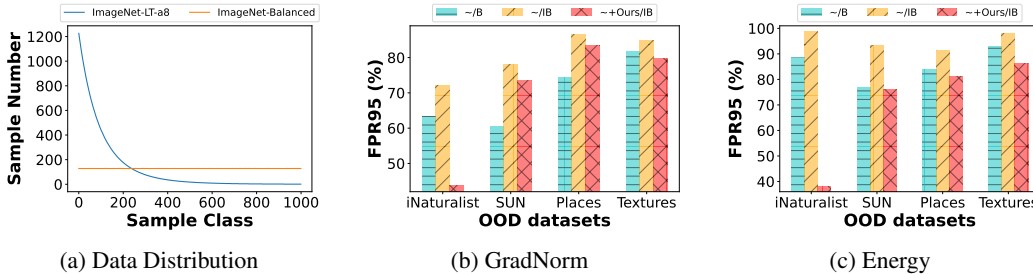

| (a) Data Distribution | (b) GradNorm | (c) Energy |

Figure 2: (a) Plot showing the data distribution of balanced and imbalanced datasets. OOD detection performances of (b) GradNorm and (c) Energy. Smaller FPR95 values are better. Cyan (left) bar: the original method on a balanced dataset. Yellow (middle) bar: the original method on an imbalanced dataset. Red (right) bar: the original method with our method on an imbalanced dataset. For a fair comparison, the sample numbers of balanced and imbalanced datasets are the same. More detailed results are shown in Appendix A.1.1.

Empirically, we test the performance of OOD detection methods when the data are not class-balanced (Figure 2a), i.e., $\mathbb{P}_{Y^{\mathrm{in}}} \neq \mathbf{u}$. We find that the GradNorm, a state-of-the-art OOD detection method, will suffer from the imbalanced situation (see cyan and yellow bars in Figure 2b). Besides, it is interesting to find that Energy (Liu et al., 2020), the other one of representative OOD detection methods that do not explicitly use $\mathbf{u}$, also suffers from this situation (see cyan and yellow bars in Figure 2c). Based on Theorem 1 and Eq. 3, we propose two effective strategies to modify previous score-based OOD detection methods using the ID-class-prior distribution: *replacing* (RP) strategy and *reweighting* (RW) strategy. In RP strategy, previous methods explicitly use the uniform distribution (like GradNorm), we can modify them by *replacing* $\mathbf{u}$ with the ID-class-prior distribution $\mathbb{P}_{Y^{\mathrm{in}}}$. For example, we can modify score functions of GradNorm by replacing $\mathbf{u}$ in Eq. 2 with $\mathbb{P}_{Y^{\mathrm{in}}}$:

$$S_{\mathrm{RP+GradNorm}}(\mathbf{f}_{\boldsymbol{\Theta}}, \mathbf{x}) = \left\| \frac{\partial D_{\mathrm{KL}}(\mathbb{P}_{Y^{\mathrm{in}}} \,\|\, \mathrm{softmax}(\mathbf{f}_{\boldsymbol{\Theta}}(\mathbf{x})))}{\partial \boldsymbol{\Theta}} \right\|_{L^1}. \tag{4}$$

For the methods that do not explicitly use the uniform distribution to compute scores (like Energy (Liu et al., 2020)), we can use the RW strategy to *reweight* their scores according to the similarity between the ID-class-prior distribution $\mathbb{P}_{Y^{\mathrm{in}}}$ and the softmax outputs of the pre-trained model $\mathrm{softmax}(\mathbf{f}_{\boldsymbol{\Theta}}(\mathbf{x}))$. Namely,

$$S_{\mathrm{RW+Method}}(\mathbf{f}_{\boldsymbol{\Theta}}, \mathbf{x}) = S_{\mathrm{Method}}(\mathbf{f}_{\boldsymbol{\Theta}}, \mathbf{x}) \cdot \cos(\mathrm{softmax}(\mathbf{f}_{\boldsymbol{\Theta}}(\mathbf{x})), \mathbb{P}_{Y^{\mathrm{in}}}), \tag{5}$$

where $S_{\mathrm{Method}}(\mathbf{f}_{\boldsymbol{\Theta}}, \mathbf{x})$ is a score function proposed in previous studies (like Energy (Liu et al., 2020)).

We conduct extensive experiments to verify the effectiveness of RP and RW strategies. After our modification, the results (red bars in Figure 2) show a significant improvement, which illustrates the effectiveness of our theory. Meanwhile, our method achieves state-of-the-art performance on four evaluation tasks. As a highlight, when evaluate the OOD detection performance on iNaturalist dataset, our method can achieve ∼36% increase on AUROC and ∼61% decrease on FPR95, compared with the original Energy (Liu et al., 2020) (see Table 1). It further validates that we cannot default the reference distribution of OOD data to the uniform distribution. To improve the generalizability of OOD detection methods, the class-prior distribution of the training data should be taken into account, which might benefit future researches in the community.

## 2 PRELIMINARIES

Let $\mathcal{X} \subset \mathbb{R}^d$ and $\mathcal{Y}^{\mathrm{in}} = \{1, ..., K\}$ be the feature space and the ID label space. Denote $X^{\mathrm{in}} \in \mathcal{X}, X^{\mathrm{out}} \in \mathcal{X}$ and $Y^{\mathrm{in}} \in \mathcal{Y}^{\mathrm{in}}$ by the random variables with respect to $\mathcal{X}$ and $\mathcal{Y}^{\mathrm{in}}$. $\mathbb{P}(X^{\mathrm{in}}, Y^{\mathrm{in}})$ is the ID joint distribution, $\mathbb{P}_{X^{\mathrm{in}}}$ is the ID marginal distribution, and $\mathbb{P}_{X^{\mathrm{out}}}$ is the OOD marginal distribution.

**OOD Detection.** Given the training data $\mathcal{D}_{\mathrm{in}}^{\mathrm{train}} = \{(\mathbf{x}^1, y^1), ..., (\mathbf{x}^n, y^n)\}$ *independent and identically distributed* (i.i.d.) drawn from $\mathbb{P}(X^{\mathrm{in}}, Y^{\mathrm{in}})$, the aim of OOD detection is to learn a model $h$

using $\mathcal{D}_{\text{in}}^{\text{train}}$ such that for any test data $\mathbf{x}$ drawn from $\mathbb{P}_{X^{\text{in}}}$ or $\mathbb{P}_{X^{\text{out}}}$: 1) if $\mathbf{x}$ is drawn from $\mathbb{P}_{X^{\text{in}}}$, then $h$ can classify $\mathbf{x}$ into correct ID classes; and 2) if $\mathbf{x}$ is drawn from $\mathbb{P}_{X^{\text{out}}}$, then $h$ can detect $\mathbf{x}$ as OOD data.

**Inference-time OOD Detection.** A well-known branch of OOD detection methods is to develop the inference-time OOD detection (or post hoc OOD detection) methods (Huang et al., 2021; Liang et al., 2018; Liu et al., 2020; Hendrycks & Gimpel, 2017; Lee et al., 2018b; Sun et al., 2021), where we are given a pre-trained ID model and then aim to recognize upcoming OOD data well. The key advantage of inference-time OOD detection methods is that the classification performance on ID data will be unaffected since we only use the ID model instead of changing it.

**Score Functions.** Many representative OOD detection methods use a score-based strategy: given a threshold $\gamma$, an ID model $\mathbf{f_\Theta}$ and a scoring function $S$, then $\mathbf{x}$ is detected as ID data if $S(\mathbf{f_\Theta}, \mathbf{x}) \geq \gamma$:

$$G_\gamma(\mathbf{x}) = \begin{cases} \text{ID}, & \text{if } S(\mathbf{f_\Theta}, \mathbf{x}) \geq \gamma \\ \text{OOD}, & \text{if } S(\mathbf{f_\Theta}, \mathbf{x}) < \gamma \end{cases} \tag{6}$$

The performance of OOD detection depends on how to design a scoring function $S$ to make OOD data obtain lower scores while ID data obtain higher scores—thus, we can recognize ID/OOD data.

## 3 METHODOLOGY

Clearly, without any assumptions or conditions, OOD detection cannot be addressed well due to the unavailability of OOD data (Zhang et al., 2021). Therefore, to investigate the feasibility of OOD detection, in this section, we consider a natural case that ID classes and OOD features do not interfere with each other.

### 3.1 ASSUMPTION AND THEOREM

**Assumption 1.** *Random variables $X^{\text{out}}$ and $Y^{\text{in}}$ are independent, i.e., $\mathbb{P}(X^{\text{out}}, Y^{\text{in}}) = \mathbb{P}_{X^{\text{out}}} \mathbb{P}_{Y^{\text{in}}}$.*

**Justification of Assumption 1.** To justify that Assumption 1 is realistic, we conclude three common causal graphs in Figure 1. These graphs illustrate how the data is generated through the lens of causality. Notably, in Figure 1c, we can observe that $X^{\text{in}}$ and $X^{\text{out}}$ are actually dependent, which is very common in our daily life. It seems that the dependence of $X^{\text{in}}$ and $X^{\text{out}}$ could result in the failure of Assumption 1. However, since $X^{\text{in}}$ and $X^{\text{out}}$ are dependent only because of the same style ($S$ in Figure 1) instead of classes ($Y$ in Figure 1) (Yao et al., 2021), the condition that $X^{\text{in}}$ and $X^{\text{out}}$ are dependent does not conflict with Assumption 1. In fact, there exist many practical scenarios which meet the causal structure in Figure 1c, e.g., MNIST and Fashion-MNIST (Xiao et al., 2017). According to this assumption, we can prove our main theorem, which provides the theoretical foundation for our paper.

**Theorem 1.** *If Assumption 1 holds, then $\mathbb{P}_{Y^{\text{in}}|X^{\text{out}}}(y|\mathbf{x}) = \mathbb{P}_{Y^{\text{in}}}(y)$, for any $y \in \mathcal{Y}^{\text{in}}$.*

The proof of Theorem 1 is in Appendix B.1. In the inference-time OOD detection, researchers assume that the pre-trained model $\mathbf{f_\Theta}(\mathbf{x})$ can simulate the ID conditional distribution, i.e., the probabilities that $\mathbf{x}$ belongs to each ID class: $\text{softmax}(\mathbf{f_\Theta}(\mathbf{x})) \approx [\mathbb{P}_{Y^{\text{in}}|X^{\text{out}}}(1|\mathbf{x}), ..., \mathbb{P}_{Y^{\text{in}}|X^{\text{out}}}(K|\mathbf{x})]$, when the data point $\mathbf{x}$ is from OOD distribution. Therefore, Theorem 1 implies that if Assumption 1 holds, then we hope that $\text{softmax}(\mathbf{f_\Theta}(\mathbf{x})) \approx [\mathbb{P}_{Y^{\text{in}}}(1), ..., \mathbb{P}_{Y^{\text{in}}}(K)] = \mathbb{P}_{Y^{\text{in}}}$, for any OOD data $\mathbf{x}$. Next, we discuss how to utilize this *novel observation* to improve existing score-based OOD methods. When the labels of the training dataset are not available, we can use the predictions made by the model as an alternative to simulate empirical ID-class-prior distribution $\mathbb{P}_{Y^{\text{in}}}$. The specific analysis and experiments can be found in the Appendix B.2.

### 3.2 RETHINKING MSP AND GRADNORM BY THEOREM 1

According to Eq. 6, we discover that the score-based strategy has an *implied assumption* that if a data point $\mathbf{x}$ has a lower score, then the data $\mathbf{x}$ has a higher probability detected as an OOD data point. Based on this assumption, we consider the *ideal* case that if a data point $\mathbf{x}$ has the smallest score, then what will happen? We answer this issue, when the score function is MSP.

**Rethinking MSP by Theorem 1.** We consider the MSP score and answer above issue by Theorem 2.

**Theorem 2.** *Given a data point* $\mathbf{x} \in \mathcal{X}$, *if* $\mathbf{f}_{\Theta}^*(\mathbf{x}) \in \arg\min_{\mathbf{f}_{\Theta}(\mathbf{x})} S_{\mathrm{MSP}}(\mathbf{f}_{\Theta}, \mathbf{x})$, *then*

$$\mathrm{softmax}(\mathbf{f}_{\Theta}^*(\mathbf{x})) = \mathbf{u}, \text{ where } \mathbf{u} = [1/K, \ldots, 1/K] \in \mathbb{R}^K.$$

The proof of Theorem 2 is in Appendix B.3. According to the implied assumption, we know that when the data point has the smallest score, then $\mathbf{x}$ has the largest probability detected as an OOD data point. Then, Theorem 2 shows that in the ideal case, the output of this data point $\mathbf{x}$ is a uniform distribution $\mathbf{u}$, which is conflict with our observation (i.e., $\mathrm{softmax}(\mathbf{f}_{\Theta}(\mathbf{x})) \approx \mathbb{P}_{Y^{\mathrm{in}}}$, if $\mathbf{x}$ is an OOD data point) in the ID class-imbalance case. Therefore, to avoid the contradiction, we can replace the uniform distribution in MSP as follows:

$$S_{\mathrm{RP+MSP}}(\mathbf{f}_{\Theta}, \mathbf{x}) = \max_{i \in \{1, \ldots, K\}} (\mathrm{softmax}_i(\mathbf{f}_{\Theta}(\mathbf{x})) - \mathbb{P}_{Y^{\mathrm{in}}}(i)). \tag{7}$$

**Rethinking GradNorm by Theorem 1.** Here, we discuss how to adjust the GradNorm score. By Eq. 2, it is clear that in the ideal case, we can conclude that $\mathrm{softmax}(\mathbf{f}_{\Theta}(\mathbf{x})) \approx \mathbf{u}$, i.e.,

$$\lim_{\gamma \to 0} \mathrm{softmax}(\mathbf{f}_{\Theta}(\mathbf{x})) = \mathbf{u}, \text{ where } \mathbf{f}_{\Theta}(\mathbf{x}) \text{ satisfies } S_{\mathrm{GradNorm}}(\mathbf{f}_{\Theta}, \mathbf{x}) < \gamma. \tag{8}$$

Therefore, Eq. 8 is inconsistent with our observation (i.e., $\mathrm{softmax}(\mathbf{f}_{\Theta}(\mathbf{x})) \approx \mathbb{P}_{Y^{\mathrm{in}}}$, if $\mathbf{x}$ is an OOD data point) in the ID class-imbalance case. Similar to the MSP scenario, the basic idea is to use the ID-class-prior distribution $\mathbb{P}_{Y^{\mathrm{in}}} = [\mathbb{P}_{Y^{\mathrm{in}}}(1), \ldots, \mathbb{P}_{Y^{\mathrm{in}}}(K)]$ to replace the uniform distribution $\mathbf{u}$, i.e.,

$$S_{\mathrm{RP+GradNorm}}(\mathbf{f}_{\Theta}, \mathbf{x}) = \left\| \frac{\partial D_{\mathrm{KL}}(\mathbb{P}_{Y^{\mathrm{in}}} \| \mathrm{softmax}(\mathbf{f}_{\Theta}(\mathbf{x}))}{\partial \Theta} \right\|_{L^1}. \tag{9}$$

### 3.3 OUR PROPOSAL: REPLACING AND REWEIGHTING STRATEGIES

**Replacing (RP) Strategy.** For those methods (e.g., MSP and GradNorm) whose score functions are deeply related to the uniform distribution $\mathbf{u}$, the simple and straight way of modifying them is to replace the uniform distribution $\mathbf{u}$ with the ID-class-prior distribution $\mathbb{P}_{Y^{\mathrm{in}}}$. As mentioned in Section 3.2, we modify the score functions of MSP and GradNorm as

$$S_{\mathrm{RP+MSP}}(\mathbf{f}_{\Theta}, \mathbf{x}) = \max_{i \in \{1, \ldots, K\}} (\mathrm{softmax}(\mathbf{f}_{\Theta}(\mathbf{x}))_i - \mathbb{P}_{Y^{\mathrm{in}}}(i)), \tag{10}$$

$$S_{\mathrm{RP+GradNorm}}(\mathbf{f}_{\Theta}, \mathbf{x}) = \left\| \frac{\partial D_{\mathrm{KL}}(\mathbb{P}_{Y^{\mathrm{in}}} \| \mathrm{softmax}(\mathbf{f}_{\Theta}(\mathbf{x})))}{\partial \Theta} \right\|_{L^1}. \tag{11}$$

**Reweighting (RW) Strategy.** For the methods that have no obvious correlations with the uniform distribution $\mathbf{u}$ (e.g., ODIN (Liang et al., 2018) and Energy (Liu et al., 2020)), we design the RW strategy as a complementary to the RP strategy. RW strategy *reweights* their scores according to a similarity between the ID-class-prior distribution $\mathbb{P}_{Y^{\mathrm{in}}}$ and $\mathrm{softmax}(\mathbf{f}_{\Theta}(\mathbf{x}))$. Here, we expect that the weights do not impact on the OOD scores seriously. In this paper, we use the cosine function as the weight function, which is one of the most popular distances and similarity functions in contrastive learning (Chen et al., 2020; Grill et al., 2020; He et al., 2020). The main reason we choose cosine function is that cosine is a bounded function and suitable as a weighting parameter after normalization. Specifically,

$$\begin{aligned} S_{\mathrm{RW+Method}}(\mathbf{f}_{\Theta}, \mathbf{x}) &= -S_{\mathrm{Method}}(\mathbf{f}_{\Theta}, \mathbf{x}) \cdot \cos(\mathrm{softmax}(\mathbf{f}_{\Theta}(\mathbf{x})), \mathbb{P}_{Y^{\mathrm{in}}}) \\ &= -S_{\mathrm{Method}}(\mathbf{f}_{\Theta}, \mathbf{x}) \cdot \frac{\mathrm{softmax}(\mathbf{f}_{\Theta}(\mathbf{x})) \cdot \mathbb{P}_{Y^{\mathrm{in}}}^{\top}}{\| \mathrm{softmax}(\mathbf{f}_{\Theta}(\mathbf{x})) \| \cdot \| \mathbb{P}_{Y^{\mathrm{in}}} \|}, \end{aligned} \tag{12}$$

where $S_{\mathrm{Method}}(\mathbf{f}_{\Theta}, \mathbf{x})$ is a score function proposed in previous studies, e.g., ODIN and Energy. Next, we introduce the details about the reweighted ODIN and reweighted Energy in the following.

Compared to MSP, the main improvement of ODIN is the use of a temperature scaling strategy. We can modify ODIN as follows:[2] for a temperature $T > 0$,

$$S_{\text{RW+ODIN}}(\mathbf{f}_\Theta, \mathbf{x}) = - \max_{i \in \{1,...,K\}} \frac{\exp\left(\mathbf{f}_i(\mathbf{x})/T\right)}{\sum_{j=1}^{K} \exp\left(\mathbf{f}_j(\mathbf{x})/T\right)} \cdot \cos(\text{softmax}(\mathbf{f}_\Theta(\mathbf{x})), \mathbb{P}_{Y^{\text{in}}}). \quad (13)$$

Energy (Liu et al., 2020) proposes to replace the softmax function with the energy function (LeCun et al., 2006) for OOD detection. The energy function has a property that is highly correlated with the distribution: the system with a more concentrated probability distribution has lower energy, while the system with a more divergent probability distribution (more similar to the uniform distribution) has higher energy (LeCun et al., 2006). Thus, the energy of ID data is smaller than OOD data. Based on Eq. 12, we modify Energy as follows:

$$S_{\text{RW+Energy}}(\mathbf{f}_\Theta, \mathbf{x}) = T \cdot \log \sum_{i=1}^{K} e^{\mathbf{f}_i(\mathbf{x})/T} \cdot \cos(\text{softmax}(\mathbf{f}_\Theta(\mathbf{x})), \mathbb{P}_{Y^{\text{in}}}). \quad (14)$$

In this paper, we mainly realize our strategies using Eqs. 10, 11, 13 and 14.

## 4 EXPERIMENTS

In this section, we construct a series of imbalanced ID datasets whose data are sampled from the ImageNet-1K (Deng et al., 2009). Then, we train the ID classifiers on them as pre-trained ID models, and use large-scale ImageNet OOD detection benchmark (Huang & Li, 2021) to evaluate our methods, i.e., RP+MSP (Eq. 10), RP+GradNorm (Eq. 11), RW+ODIN (Eq. 13), and RW+Energy (Eq. 14). In addition, we also evaluate our methods on a real-world imbalanced dataset iNaturalist (Horn et al., 2018), see Appendix A.2.

### 4.1 EXPERIMENT SETUP

**Dataset.** Following Liu et al. (2019), we construct a series of imbalanced datasets that are sampled by the Pareto distribution in ImageNet-1K dataset. The definition of Pareto distribution is in Eq. (15).

$$p(x) = \frac{am^a}{x^{a+1}}. \quad (15)$$

In Appendix B.5, we have shown that the parameter $m$ does not affect the level of imbalance. Thus, we set $m = 1$. Additionally, we note that the level of imbalance depends on the tail index $a$ (see Figure 3), thus, to evaluate the performance of our methods in different imbalanced cases, we take different tail index $a$.

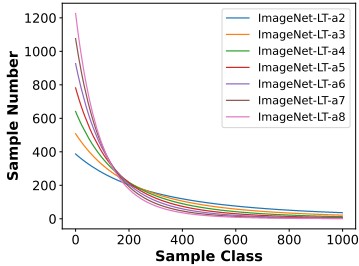

Figure 3: Data distribution with different tail index $a$.

The frequency distributions of classes of the sampled datasets are shown in Figure 3. As the increase of the tail index $a$, the sampled datasets become more imbalanced, thus, the ImageNet-LT-a8 dataset is the most imbalanced.

In the inference time, we use the large-scale benchmark proposed by Huang & Li (2021). In this benchmark, the OOD datasets include the subsets of iNaturalist (Horn et al., 2018), SUN (Xiao et al., 2010), Places (Zhou et al., 2018), and Textures (Cimpoi et al., 2014). Note that, there are no overlapping classes between ID datasets and OOD datasets (Huang & Li, 2021).

**Evaluation Metrics.** We use two common metrics to evaluate OOD detection methods (Huang et al., 2021): the false positive rate that OOD data are classified as ID data when 95% of ID data are correctly classified (FPR95) (Provost et al., 1998) and the *area under the receiver operating characteristic curve* (AUROC) (Huang et al., 2021).

---

[2]In fact, ODIN uses the modified softmax function with temperature $T$, which is also related to the uniform distribution, so we can also modify ODIN with RP strategy. We can map the class-prior distribution to the same feature space with ODIN's OOD scores by temperature $T$. However, if following the default setting ($T = 1000$) in ODIN, $\|\mathbb{P}_{Y^{\text{in}}} - \mathbf{u}\|/T \approx 0$. Thus, RP+ODIN may not work. We will discuss this issue in Appendix B.4.

Table 1: OOD detection performance comparison with other competitive score-based OOD detection methods. All methods are based on ResNet101 trained on ImageNet-LT-a8. ↑ indicates larger values are better and ↓ indicates smaller values are better. All values are percetages. The bold indicates the best performance while the underline indicates the second.

| Method | iNaturalist | | SUN | | Places | | Textures | | Average | |
|---|---|---|---|---|---|---|---|---|---|---|
| | AUROC↑ | FPR95↓ | AUROC↑ | FPR95↓ | AUROC↑ | FPR95↓ | AUROC↑ | FPR95↓ | AUROC↑ | FPR95↓ |
| MSP (Hendrycks & Gimpel, 2017) | 63.95 | 97.72 | 66.60 | 93.13 | 66.84 | 92.11 | 42.74 | 98.79 | 60.03 | 95.44 |
| ODIN (Liang et al., 2018) | 60.14 | 98.70 | 70.63 | 93.13 | 70.14 | 91.96 | 41.83 | 98.30 | 60.69 | 95.52 |
| Mahalanobis (Lee et al., 2018b) | 60.72 | 95.87 | 56.79 | 94.50 | 55.27 | 93.78 | 49.43 | 86.99 | 55.55 | 92.78 |
| Energy (Liu et al., 2020) | 55.99 | 98.74 | 71.12 | 93.11 | 70.24 | 91.30 | 42.38 | 98.07 | 59.93 | 95.30 |
| GradNorm (Huang et al., 2021) | 82.51 | 72.19 | 74.57 | 78.10 | 70.67 | 86.58 | 57.31 | 84.95 | 71.26 | 80.45 |
| Maxlogit (Hendrycks et al., 2022) | 60.14 | 98.70 | 70.64 | 93.13 | 70.15 | 91.96 | 41.83 | 98.30 | 60.69 | 95.52 |
| Dice (Sun & Li, 2022) | 85.80 | 58.96 | 73.17 | 76.90 | 67.83 | 87.89 | 58.43 | 80.59 | 71.31 | 76.11 |
| RP+MSP(Ours) | 64.95 | 96.44 | 67.39 | 91.79 | 67.46 | 91.16 | 43.05 | 98.51 | 60.71 | 94.48 |
| RW+ODIN(Ours) | 86.66 | 93.85 | 71.59 | 97.67 | 67.56 | 97.24 | **68.04** | 95.37 | 73.46 | 96.03 |
| RW+Energy(Ours) | **91.92** | **37.89** | **80.81** | 76.22 | **77.15** | **81.18** | 64.48 | 86.19 | **78.59** | 70.37 |
| RP+GradNorm(Ours) | 91.23 | 43.87 | 77.36 | **73.53** | 72.67 | 83.29 | 62.94 | **79.80** | 76.05 | **70.12** |
| RW+Maxlogit(Ours) | 78.16 | 77.68 | 77.26 | 81.91 | 75.15 | 83.28 | 48.34 | 94.06 | 69.73 | 84.23 |

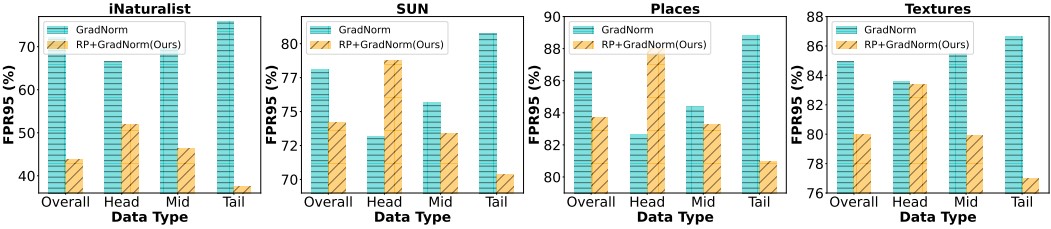

Figure 4: Performance comparison of different data type. The figures shows the OOD detection performance of GradNorm and RP+GradNorm in four OOD datasets.

**Baselines.** In order to verify the effectiveness of our strategies, we select MSP, ODIN, Energy, GradNorm and Dice as the baselines, where Dice is the *state-of-the-art* (SOTA) method. Following (Huang et al., 2021; Liang et al., 2018), the temperature parameter $T$ in ODIN is set to be 1000 and in GradNorm is 1.

**Models and Hyperparameters.** We use *mmclassification*[3] (Contributors, 2020) with Apache-2.0 license to train ID models. The training details of ResNet (He et al., 2016) and MobileNet (Howard et al., 2019) follow the default setting in *mmclassification*. Note that, all methods are realized by Pytorch 1.60 with CUDA 10.2, where we use several NVIDIA Tesla V100 GPUs.

## 4.2 EXPERIMENTAL RESULTS AND ANALYSIS

**Verification of Two Strategies.** Our strategies are applicable to various score functions used by OOD detection methods. The performance of our methods and baselines are shown in Table 1. Overall, after modifying previous methods using our strategies, their performance are significantly improved, indicating the effectiveness of our strategies. Specifically, RW+Energy achieves the highest AUROC (78.59%) compared to all methods. As a highlight, RW+Energy shows the most significant performance improvement on all four datasets: ~61% FPR95 decrease in iNaturalist, ~17% FPR95 decrease in SUN, ~10% FPR95 decrease in Places and ~12% FPR95 decrease in Textures.

Besides, our strategies outperform the existing baseline methods in all evaluation tasks. Compared with the best baseline, RW+Energy increases AUROC from 71.31% to 78.59%, while RP+GradNorm reduces FPR95 from 76.11% to 70.12%. Experimental results have shown that our strategies can significantly outperform the baselines in the ID-class-imbalanced scenarios.

**Analysis of Detection Results on Different ID Classes.** Since the training dataset is imbalanced, we follow Liu et al. (2019) to divide all ID classes into three categories (ID Head, ID Mid, ID Tail) for further analysis. In detail, ID Head category includes the classes containing more than 100 samples in the training dataset;ID Mid category includes the classes whose number of samples is between 20 and 100 in the training dataset; and ID Tail category includes the classes containing less than 20

---

[3]https://github.com/open-mmlab/mmclassification

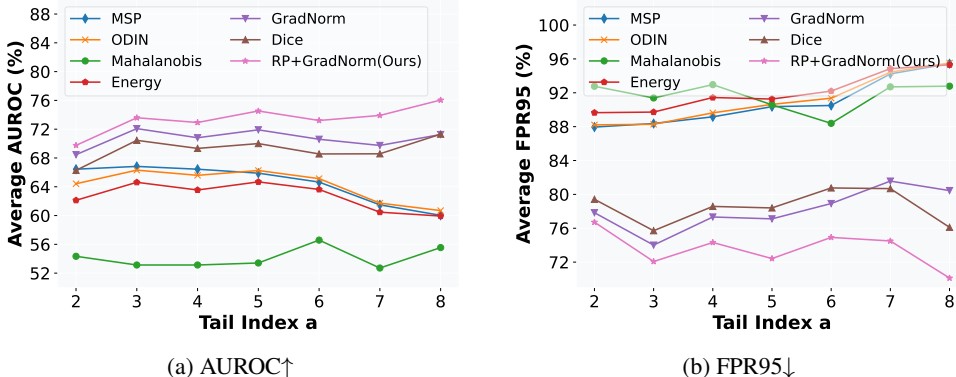

(a) AUROC↑             (b) FPR95↓

Figure 5: OOD detection performance with ResNet101 trained on different imbalanced ID datasets. ↑ indicates larger values are better and ↓ indicates smaller values are better.

samples in the training dataset. Then, we evaluate the OOD detection performance on three datasets: ID Head+OOD, ID Mid+OOD and ID Tail+OOD (details can be found in Appendix A.1.3). If the performance of one method on ID Tail+OOD is better than that on ID Head+OOD, then this method performs better when facing tailed classes and OOD data.

In the case of GradNorm, experiment results in Figure 4 show that our method RP+GradNorm improves the performance on the above three datasets (ID Head+OOD, ID Mid+OOD, and ID Tail+OOD). When we take a closer look at the performance improvement, we notice that the overall performance improvement of RP+GradNorm is mainly due to the significant improvement on the Tail+OOD dataset. This result might indicate that the previous method, like GradNorm, confuses OOD data and ID tailed classes, which hinders their OOD detection performance. And our strategies can overcome this issue. More detailed results are shown in Appendix A.1.3.

### 4.3 ABLATION STUDY

**Analysis regarding Tail Index $a$.** Here, we report the performance of our method and baselines when changing the tail index $a \in \{2, 3, \ldots, 8\}$. We conduct repeated experiments on these seven datasets (ImageNet-LT-a2, ImageNet-LT-a3,..., ImageNet-LT-a8), and the results are shown in Figure 5. Overall, our method RP+GradNorm always outperforms other baselines with different imbalance degrees. More importantly, the performance improvement between RP+GradNorm and each baseline gradually increases, as the increase of the imbalance degree. This indicates that RP+GradNorm can handle different imbalanced scenarios better. More detailed results are in Appendix A.1.2.

**Analysis regarding Network Architecture.** We evaluate all methods on a different network architecture, MobileNet-V3 (Howard et al., 2019). Experiment results in Table 2 show that our methods (RP+GradNorm and RW+Energy) still outperform baselines on four evaluation tasks even when we change the network architecture. In addition, RP+GradNorm has a better performance on FPR95 while RW+Energy has higher AUROC, corresponding to the performances of GradNorm and Energy.

**Analysis regarding Model Size.** We provide an experiment about the model size of RP+GradNorm. We compare ResNet50, ResNet101 and ResNet152 trained on ImageNet-LT-a8 datasets. The results are shown in Table 3. The optimal model is the smallest one (ResNet50), and we observe that as the increase of the model size, the performance decreases. One of the reasons may be that small models are more difficult to overfit and thus more suitable for OOD detection in imbalanced scenarios.

**More Experiments and Exploration.** First, we can also regard the cosine similarity weights in the RW strategy as a score function, and conduct several experiments in Appendix A.1.4. Second, to evaluate the stability of our strategies, we conduct 10 independent replicate experiments in Appendix A.1.5. Third, to further explore the collocation of existing methods and our design strategy, we conduct experiments regarding RW+MSP, RW+GradNorm, RW+RP+MSP/GradNorm, as shown in Appendix A.1.6. Then, we further consider what will happen if the ID-class prior is not accurate in the practical applications and conduct relevant experiments in Appendix A.1.7. Finally, we show in Appendix A.1.8 that our methods still work well when models are trained with long-tailed learning strategies (Cao et al., 2019; Park et al., 2022) during the training phase.

Table 2: OOD detection performance with MobileNet trained on ImageNet-LT-a8. ↑ indicates larger values are better and ↓ indicates smaller values are better. All values are percentages. The bold indicates the best performance while the underline indicates the second.

| Method | iNaturalist | | SUN | | Places | | Textures | | Average | |
|---|---|---|---|---|---|---|---|---|---|---|
| | AUROC↑ | FPR95↓ | AUROC↑ | FPR95↓ | AUROC↑ | FPR95↓ | AUROC↑ | FPR95↓ | AUROC↑ | FPR95↓ |
| MSP (Hendrycks & Gimpel, 2017) | 63.47 | 92.38 | 67.27 | 85.62 | 64.16 | 89.62 | 59.88 | 89.06 | 63.69 | 89.17 |
| ODIN (Liang et al., 2018) | 64.68 | 93.78 | 74.29 | 79.42 | 69.94 | 89.70 | 69.06 | 82.23 | 69.49 | 86.28 |
| Energy (Liu et al., 2020) | 63.42 | 96.37 | 74.95 | 77.86 | 70.30 | 90.50 | 70.69 | 80.83 | 69.84 | 86.39 |
| GradNorm (Huang et al., 2021) | 70.87 | 78.12 | 69.70 | 67.59 | 66.00 | 85.75 | 63.09 | 74.89 | 67.41 | 76.59 |
| Dice (Sun & Li, 2022) | 65.61 | 86.40 | 69.35 | 66.38 | 65.95 | 88.42 | 68.85 | 68.19 | 67.44 | 77.35 |
| RP+MSP(Ours) | 63.76 | 91.76 | 67.56 | 85.11 | 64.37 | 89.41 | 60.02 | 88.62 | 63.93 | 88.73 |
| RW+ODIN(Ours) | 80.20 | 73.32 | 72.46 | 77.48 | 68.25 | 89.08 | 67.62 | 73.24 | 72.13 | 78.28 |
| RW+Energy(Ours) | 81.64 | 69.94 | 76.10 | 80.36 | 70.41 | 86.71 | 65.58 | 85.82 | 73.43 | 80.71 |
| RP+GradNorm(Ours) | 77.25 | 68.61 | 72.49 | 66.02 | 68.56 | 82.30 | 64.69 | 71.86 | 70.75 | 72.20 |

Table 3: OOD detection performance with model size increases. The RP+GradNorm method is trained on ImageNet-LT-a8. All values are percentages.

| Model | iNaturalist | | SUN | | Places | | Textures | | Average | |
|---|---|---|---|---|---|---|---|---|---|---|
| | AUROC↑ | FPR95↓ | AUROC↑ | FPR95↓ | AUROC↑ | FPR95↓ | AUROC↑ | FPR95↓ | AUROC↑ | FPR95↓ |
| ResNet50 | 89.85 | 50.03 | 80.73 | 64.52 | 74.69 | 78.18 | 63.31 | 77.73 | 77.14 | 67.62 |
| ResNet101 | 91.23 | 43.87 | 77.36 | 73.53 | 72.67 | 83.29 | 62.94 | 79.80 | 76.05 | 70.12 |
| ResNet152 | 88.24 | 53.14 | 73.45 | 78.96 | 68.61 | 87.14 | 59.41 | 83.71 | 72.43 | 75.74 |

## 5 RELATED WORKS

**Inference-time/post-hoc OOD Detection:** Some methods (Huang et al., 2021; Liang et al., 2018; Liu et al., 2020; Hendrycks & Gimpel, 2017; Lee et al., 2018b; Sun et al., 2021) focus on designing OOD score functions for OOD detection in the inference time and are easy to use without changing the model's parameters. This property is important for deploying OOD detection methods in real-world scenarios where the cost of re-training is prohibitively expensive and time-consuming. MSP (Hendrycks & Gimpel, 2017) directly takes the maximum value of the model's prediction as the OOD score function. Based on MSP, ODIN (Liang et al., 2018) uses a temperature scaling strategy and input perturbation to improve OOD detection performance. Moreover, Liu et al. (2020) and Wang et al. (2021a) propose to replace the softmax function with the energy functions for OOD detection. Recently, GradNorm (Huang et al., 2021) uses the similarity of the model-predicted probability distribution and the uniform distribution to improve OOD detection and achieve state-of-the-art performance. In this paper, we mainly work on the inference-time OOD detection methods and aim at improving the generalizability of OOD detection in real-world scenarios.

**Training-time OOD Detection:** Other methods (Hsu et al., 2020; Hein et al., 2019; Bitterwolf et al., 2020; Wang et al., 2021b) will complete ID tasks and OOD detection simultaneously in the training time. Bitterwolf et al. (2020) uses adversarial learning to process OOD data in training time and make the model predict lower confidence scores for them. Wang et al. (2021b) generates pseudo OOD data by adversarial learning to re-training a K+1 model for OOD detection. These methods usually require auxiliary OOD data available in the training process. Thus, the model will be affected by both ID data and OOD data. It is important for these method to explore an inherent trade-off (Liu et al., 2019; Vaze et al., 2022; Yang et al., 2021) between ID tasks and OOD detection.

## 6 CONCLUSION

This paper theoretically and empirically shows that the unproven assumption of uniform distribution in previous methods is not valid when the training dataset is imbalanced. Moreover, by analyzing the causal relations between ID/OOD classes and features, we point out that the best reference distribution for OOD data is the ID-class-prior distribution. Based on this, we propose two simple and effective strategies to modify the uniform distribution assumption in previous inference-time OOD detection methods. RP strategy is suitable for the methods that directly use the uniform distribution to design the OOD score function, while RW strategy is designed for methods that potentially use the assumption. Extensive experiments show that both strategies can significantly improve the performance of OOD detection on large-scale image classification benchmarks.

## 7 ETHIC STATEMENT

This paper does not raise any ethics concerns. This study does not involve any human subjects, practices to data set releases, potentially harmful insights, methodologies and applications, potential conflicts of interest and sponsorship, discrimination/bias/fairness concerns, privacy and security issues, legal compliance, and research integrity issues.

## 8 REPRODUCIBILITY STATEMENT

To ensure the reproducibility of experimental results, we provide main codes in the Appendix D. The experimental setups for training and evaluation as well as the hyperparamters are detailedly described in Section 4.1.

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

# Table of Contents of Appendix

# A    FURTHER EXPERIMENTS

## A.1    EVALUATION ON IMAGENET BENCHMARK

### A.1.1    EVALUATION ON IMBALANCED DATA AND BALANCED DATA

We randomly sample a balanced dataset from ImageNet-1K dataset, which has the same sample numbers with the imbalanced datasets. We conduct experiments on the balanced data as shown in Figure 6 and Figure 7. All methods shows a similar trend, i.e., the performance drop a lot when the training dataset becomes imbalanced (from cyan bars to yellow bars). Moreover, our method shows a significant improvement with previous methods on all evaluation tasks (from yellow bars to red bars).

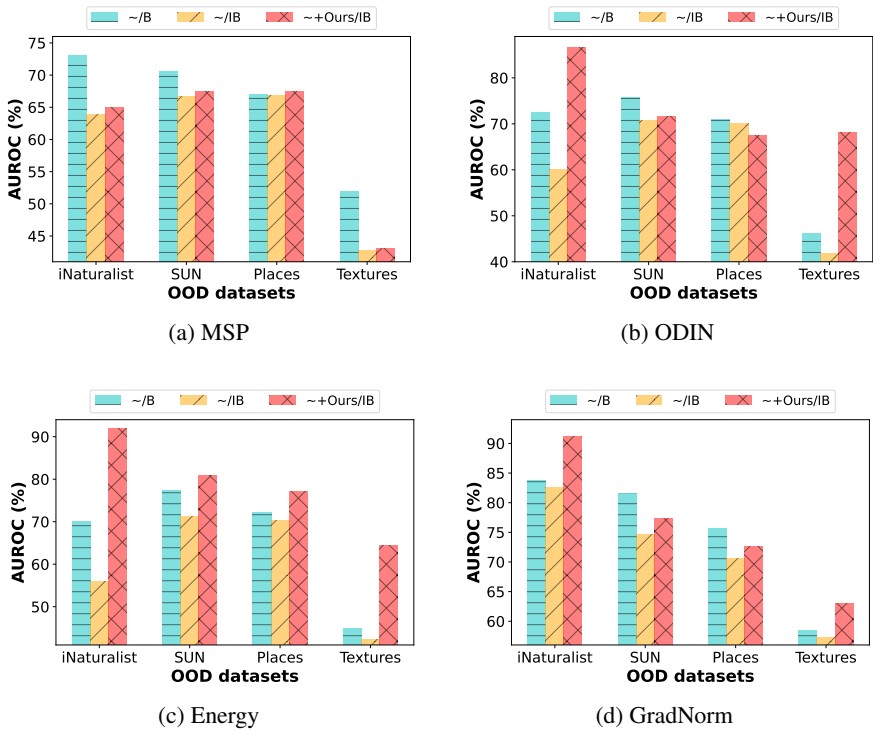

Figure 6: OOD detection performance (AUROC) of (a) MSP (b) ODIN (c) Energy and (d) GradNorm. Larger AUROC values are better. Cyan (left) bar: the original method on balanced dataset. Yellow (middle) bar: the original method on imbalanced dataset. Red (right) bar: the original method with our method on imbalanced dataset.

### A.1.2    ANALYSIS REGARDING TAIL INDEX

We can obtain sampled datasets with different levels of imbalance based on the Pareto distribution. ImageNet-LT-a8 dataset is the most imbalanced, while ImageNet-LT-a2 is the most balanced. We conduct repeated experiments on these seven datasets and results are shown in Table 4. Obviously, the more imbalanced the training ID dataset becomes, the more our methods (RP+GradNorm and Cosine Similarity) demonstrates their superior performance of OOD detection, compared to other methods on all evaluation tasks.

It is noticeable that the detection performance of GradNorm is relatively stable no matter how imbalanced the ratio changes, compared with other existing methods (such as MSP, ODIN, Energy). These methods explicitly/implicitly use the discrepancy between the classifier's output and the uniform distribution. Thus, they will be affected a lot if the prior distribution changes from the uniform distribution to an imbalanced/tailed one.

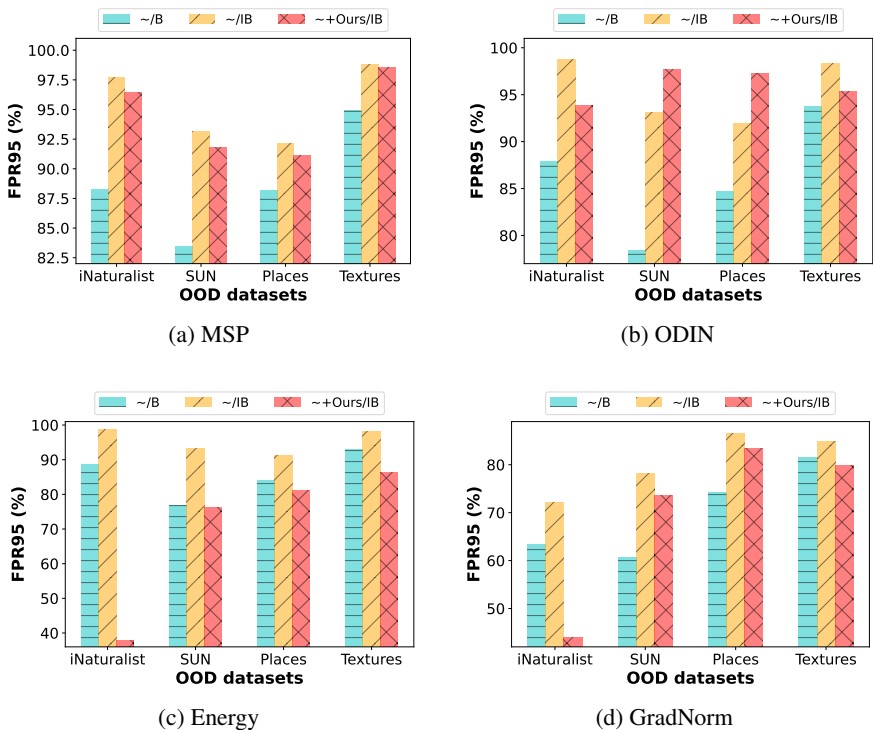

Figure 7: OOD detection performance (FPR95) of (a) MSP (b) ODIN (c) Energy and (d) GradNorm. Smaller FPR95 values are better. Cyan (left) bar: the original method on balanced dataset. Yellow (middle) bar: the original method on imbalanced dataset. Red (right) bar: the original method with our method on imbalanced dataset.

As for GradNorm, we conjecture that considering the gradient space might be robust to the changes of priors (such as from a uniform prior to an imbalanced prior). To verify this conjecture, we conduct the experiment that KL divergence is directly used to measure the discrepancy between the output of the classifier and the uniform distribution (i.e., GradNorm without gradient-norm process). The results are shown in Table 5. Obviously, this KL-based method is also significantly affected by the imbalanced situation, then we can verify this conjecture. Thus, we confirm that GradNorm's robustness of the imbalanced ratio depends on the gradient space.

### A.1.3 ANALYSIS OF DETECTION RESULTS ON DIFFERENT ID CLASSES

We calculate the evaluation metrics for the three categories (ID-Head, ID-Mid, ID-Tail) by randomly sampling OOD data in equal proportions corresponding to the number of samples in each category. For example, we have 50000 ID samples and 10000 OOD samples in total. If the number of samples in category Head is 10000 and accordingly we will sample 2000 OOD samples, then we use the 12000 samples to calculate AUROC and FPR95. The results reflects the confusion degree between ID Head data and OOD data in the view of OOD detection methods. We conduct experiments to analyze the performance of different data types, as shown in Table 6.

We try to analyze tailed categories using a confusion matrix, and there are some cases in Table 7. $\begin{bmatrix} A & B \\ C & D \end{bmatrix}$ is the result, where A represents the number of ID samples in the current class that are correctly classified as ID while C represents the number of ID samples in the current class that are misclassified as ID. D represents the number of OOD samples close to ID samples in the current class that are correctly classified as OOD while B represents the number of OOD samples close to ID samples in the current class that are misclassified as OOD.

In Class #88 and #94, more OOD samples are correctly classified after applying our strategies, while the performance of Class #671 remains stable. In Class #671, more ID samples are correctly classified,

Table 4: OOD detection performance with ResNet101 trained on different imbalanced ID datasets. ↑ indicates larger values are better and ↓ indicates smaller values are better. All values are percentages.

| Datasets | Method | iNaturalist AUROC↑ | FPR95↓ | SUN AUROC↑ | FPR95↓ | Places AUROC↑ | FPR95↓ | Textures AUROC↑ | FPR95↓ | Average AUROC↑ | FPR95↓ |
|---|---|---|---|---|---|---|---|---|---|---|---|
| ImageNet-LT-a2 | MSP Hendrycks & Gimpel (2017) | 71.57 | 89.71 | 73.74 | 80.64 | 70.76 | 85.17 | 49.68 | 96.28 | 66.44 | 87.95 |
| | ODIN Liang et al. (2018) | 65.92 | 93.17 | **76.32** | 79.60 | **72.77** | 84.29 | 42.67 | 95.76 | 64.42 | 88.21 |
| | Mahalanobis (Lee et al., 2018b) | 61.12 | 92.51 | 52.79 | 95.47 | 52.26 | 94.96 | 51.19 | 88.14 | 54.34 | 92.77 |
| | Energy Liu et al. (2020) | 60.26 | 95.63 | 75.57 | 81.89 | 72.00 | 85.90 | 40.67 | 95.18 | 62.12 | 89.65 |
| | GradNorm Huang et al. (2021) | 75.82 | 75.86 | 75.33 | **68.03** | 70.13 | **80.61** | 52.58 | 86.86 | 68.46 | 77.84 |
| | Dice (Sun & Li, 2022) | 72.17 | 77.31 | 73.94 | 70.28 | 68.88 | 83.69 | 50.12 | **86.38** | 66.28 | 79.42 |
| | RP+GradNorm(Ours) | 79.66 | 70.18 | 75.70 | 68.50 | 70.33 | 81.34 | **53.30** | 86.88 | **69.75** | **76.72** |
| | Cosine Similarity(Ours) | **81.84** | **67.06** | 73.26 | 74.27 | 68.78 | 81.75 | 47.94 | 92.18 | 67.95 | 78.82 |
| ImageNet-LT-a3 | MSP Hendrycks & Gimpel (2017) | 73.97 | 89.10 | 73.38 | 80.91 | 69.91 | 87.20 | 50.07 | 96.28 | 66.83 | 88.37 |
| | ODIN Liang et al. (2018) | 72.04 | 92.06 | 75.92 | 79.71 | **71.49** | 86.30 | 45.79 | 95.04 | 66.31 | 88.28 |
| | Mahalanobis (Lee et al., 2018b) | 48.51 | 94.60 | 53.81 | 92.63 | 53.85 | 91.84 | 56.36 | 86.48 | 53.13 | 91.39 |
| | Energy Liu et al. (2020) | 67.94 | 94.42 | 75.24 | 82.36 | 70.48 | 88.07 | 44.81 | 94.02 | 64.62 | 89.72 |
| | GradNorm Huang et al. (2021) | 83.41 | 65.27 | 76.93 | **67.10** | 70.94 | **81.43** | 57.03 | **82.25** | 72.08 | 74.01 |
| | Dice (Sun & Li, 2022) | 81.72 | 65.63 | 75.60 | 70.30 | 69.81 | 84.44 | 54.68 | 82.48 | 70.45 | 75.71 |
| | RP+GradNorm(Ours) | 87.13 | 56.16 | **77.47** | 67.71 | 71.42 | 81.96 | **58.35** | 82.45 | **73.59** | **72.07** |
| | Cosine Similarity(Ours) | **87.81** | **50.66** | 74.15 | 73.80 | 68.87 | 82.92 | 52.98 | 87.73 | 70.96 | 72.78 |
| ImageNet-LT-a4 | MSP Hendrycks & Gimpel (2017) | 72.41 | 89.92 | 73.00 | 83.34 | 70.38 | 86.97 | 49.96 | 96.42 | 66.44 | 89.16 |
| | ODIN Liang et al. (2018) | 70.35 | 92.76 | 74.66 | 82.95 | **71.35** | 87.30 | 46.00 | 95.50 | 65.59 | 89.63 |
| | Mahalanobis (Lee et al., 2018b) | 49.37 | 96.48 | 56.58 | 93.36 | 54.98 | 93.75 | 51.59 | 85.45 | 53.13 | 92.26 |
| | Energy Liu et al. (2020) | 66.27 | 95.05 | 73.17 | 85.85 | 69.63 | 89.71 | 45.13 | 95.14 | 63.55 | 91.44 |
| | GradNorm Huang et al. (2021) | 80.87 | 71.75 | 74.64 | **71.60** | 69.86 | 84.24 | 57.87 | 81.72 | 70.81 | 77.33 |
| | Dice (Sun & Li, 2022) | 78.60 | 72.35 | 74.02 | 73.34 | 68.94 | 86.45 | 55.75 | 82.18 | 69.33 | 78.58 |
| | RP+GradNorm(Ours) | 86.20 | 60.40 | 75.29 | 71.93 | 70.40 | 84.43 | **59.85** | 80.51 | **72.93** | 74.32 |
| | Cosine Similarity(Ours) | **89.21** | **48.91** | 74.13 | 74.53 | 70.06 | **82.41** | 57.25 | 86.60 | 72.66 | **73.11** |
| ImageNet-LT-a5 | MSP Hendrycks & Gimpel (2017) | 72.18 | 91.15 | 72.43 | 85.42 | 70.38 | 87.73 | 48.59 | 97.13 | 65.89 | 90.36 |
| | ODIN Liang et al. (2018) | 69.53 | 94.68 | 75.92 | 84.34 | 73.07 | 87.04 | 46.49 | 96.47 | 66.26 | 90.63 |
| | Mahalanobis (Lee et al., 2018b) | 48.35 | 96.03 | 58.07 | 91.81 | 57.20 | 90.52 | 50.00 | 83.95 | 53.41 | 90.58 |
| | Energy Liu et al. (2020) | 64.85 | 95.98 | 75.38 | 85.29 | 72.25 | 87.89 | 46.25 | 95.89 | 64.68 | 91.26 |
| | GradNorm Huang et al. (2021) | 84.01 | 67.28 | 75.30 | 73.25 | 69.82 | 85.38 | 58.53 | 82.52 | 71.91 | 77.11 |
| | Dice (Sun & Li, 2022) | 82.88 | 65.51 | 73.14 | 76.80 | 66.94 | 89.04 | 57.04 | 82.25 | 70.00 | 78.40 |
| | RP+GradNorm(Ours) | 89.59 | 51.01 | 76.55 | 72.93 | 70.91 | 84.68 | 61.06 | **81.05** | 74.53 | 72.42 |
| | Cosine Similarity(Ours) | **91.53** | **41.44** | **78.57** | **71.83** | **74.41** | **79.89** | 60.13 | 86.74 | **76.16** | **69.97** |
| ImageNet-LT-a6 | MSP Hendrycks & Gimpel (2017) | 70.99 | 91.04 | 71.63 | 86.67 | 70.50 | 86.62 | 45.49 | 97.66 | 64.65 | 90.50 |
| | ODIN Liang et al. (2018) | 70.82 | 92.57 | 74.04 | 87.81 | 72.18 | 88.25 | 43.49 | 96.83 | 65.13 | 91.36 |
| | Mahalanobis (Lee et al., 2018b) | 57.62 | 89.82 | 60.18 | 89.38 | 59.07 | 89.46 | 49.49 | 84.92 | 56.59 | 88.39 |
| | Energy Liu et al. (2020) | 67.93 | 93.88 | 72.62 | 89.16 | 70.44 | 89.81 | 43.53 | 95.94 | 63.63 | 92.20 |
| | GradNorm Huang et al. (2021) | 83.49 | 67.76 | 74.02 | 76.58 | 69.12 | 85.87 | 55.75 | 85.46 | 70.60 | 78.92 |
| | Dice (Sun & Li, 2022) | 82.31 | 68.27 | 71.70 | 79.35 | 66.20 | 89.82 | 54.04 | 85.60 | 68.56 | 80.76 |
| | RP+GradNorm(Ours) | 89.07 | 51.55 | 75.09 | 77.47 | 69.89 | 86.28 | 58.80 | **84.38** | 73.21 | 74.92 |
| | Cosine Similarity(Ours) | **91.22** | **43.01** | **78.35** | 78.35 | **74.64** | **82.94** | 60.08 | 88.63 | **76.07** | **73.23** |
| ImageNet-LT-a7 | MSP Hendrycks & Gimpel (2017) | 65.60 | 96.81 | 68.50 | 90.68 | 67.22 | 91.26 | 44.65 | 98.16 | 61.49 | 94.23 |
| | ODIN Liang et al. (2018) | 63.13 | 97.72 | 71.29 | 91.17 | 69.16 | 91.20 | 43.33 | 97.64 | 61.73 | 94.43 |
| | Mahalanobis Lee et al. (2018b) | 50.52 | 97.86 | 54.18 | 92.36 | 54.19 | 92.31 | 51.96 | 88.28 | 52.71 | 92.70 |
| | Energy Liu et al. (2020) | 59.46 | 98.20 | 70.56 | 91.89 | 68.07 | 91.90 | 43.77 | 97.41 | 60.47 | 94.85 |
| | GradNorm Huang et al. (2021) | 80.38 | 77.04 | 73.07 | 77.43 | 66.86 | 88.60 | 58.65 | 83.21 | 69.74 | 81.57 |
| | Dice (Sun & Li, 2022) | 81.61 | 70.42 | 71.19 | 78.93 | 63.79 | 91.40 | 57.72 | 81.95 | 68.58 | 80.68 |
| | RP+GradNorm(Ours) | 88.59 | 55.40 | 75.17 | **75.35** | 68.78 | 86.41 | **63.08** | **80.83** | 73.91 | 74.50 |
| | Cosine Similarity(Ours) | **90.12** | **49.49** | **78.62** | 77.87 | **73.94** | **82.90** | 62.43 | 87.54 | **76.28** | **74.45** |
| ImageNet-LT-a8 | MSP Hendrycks & Gimpel (2017) | 63.95 | 97.72 | 66.60 | 93.13 | 66.84 | 92.11 | 42.74 | 98.79 | 60.03 | 95.44 |
| | ODIN Liang et al. (2018) | 60.14 | 98.70 | 70.63 | 93.13 | 70.14 | 91.96 | 41.83 | 98.30 | 60.69 | 95.52 |
| | Mahalanobis (Lee et al., 2018b) | 60.72 | 95.87 | 56.79 | 94.50 | 55.27 | 93.78 | 49.43 | 86.99 | 55.55 | 92.78 |
| | Energy Liu et al. (2020) | 55.99 | 98.74 | 71.12 | 93.11 | 70.24 | 91.30 | 42.28 | 98.07 | 59.93 | 95.30 |
| | GradNorm Huang et al. (2021) | 82.51 | 72.19 | 74.57 | 78.10 | 70.67 | 86.58 | 57.31 | 84.95 | 71.26 | 80.45 |
| | Dice (Sun & Li, 2022) | 85.80 | 58.96 | 73.17 | 76.90 | 67.83 | 87.89 | 58.43 | 80.69 | 71.31 | 76.11 |
| | RP+GradNorm(Ours) | **91.23** | **43.87** | 77.36 | 73.53 | 72.67 | 83.29 | **62.94** | **79.80** | 76.05 | **70.12** |
| | Cosine Similarity(Ours) | 89.81 | 51.42 | **81.55** | 73.25 | **77.47** | **78.38** | 62.41 | 87.59 | **77.65** | 72.66 |

Table 5: OOD detection performance with ResNet101 trained on different imbalanced ID datasets. KL stands for using only KL divergence as the OOD detection function. ↑ indicates larger values are better and ↓ indicates smaller values are better. All values are percentages.

| Datasets | Method | iNaturalist AUROC↑ | FPR95↓ | SUN AUROC↑ | FPR95↓ | Places AUROC↑ | FPR95↓ | Textures AUROC↑ | FPR95↓ | Average AUROC↑ | FPR95↓ |
|---|---|---|---|---|---|---|---|---|---|---|---|
| ImageNet-LT-a2 | KL | 60.66 | 95.51 | 75.21 | 83.06 | 71.94 | 86.15 | 40.36 | 95.25 | 62.04 | 89.99 |
| ImageNet-LT-a3 | KL | 68.14 | 94.26 | 74.91 | 83.22 | 70.32 | 88.10 | 44.55 | 94.10 | 64.48 | 89.92 |
| ImageNet-LT-a4 | KL | 66.58 | 94.78 | 72.97 | 86.35 | 69.66 | 89.62 | 44.97 | 95.25 | 63.54 | 91.50 |
| ImageNet-LT-a5 | KL | 65.14 | 95.88 | 75.02 | 86.18 | 72.12 | 88.03 | 45.96 | 96.01 | 64.56 | 91.53 |
| ImageNet-LT-a6 | KL | 68.14 | 93.77 | 72.30 | 89.82 | 70.44 | 89.86 | 43.10 | 96.15 | 63.50 | 92.40 |
| ImageNet-LT-a7 | KL | 59.71 | 98.10 | 70.32 | 92.18 | 68.12 | 91.81 | 43.49 | 97.41 | 60.41 | 94.88 |
| ImageNet-LT-a8 | KL | 56.33 | 98.72 | 71.09 | 93.29 | 70.43 | 91.13 | 42.27 | 98.12 | 60.03 | 95.32 |

while the performance of Class #94 of ID samples is sacrificed for more performance improvement of OOD samples.

Moreover, we visualize the OOD score distributions in Figure 8-11. Obviously, the results and figures show that the previous methods tend to confuse OOD data and the minority classes, which hinders their performance of OOD detection. And our strategies can reduce the confusion to improve OOD detection performance.

Table 6: Performance comparison of different data type. All methods are based on ResNet101 trained on ImageNet-LT-a8. All values are percentages.

| Method | Data Type | iNaturalist | | SUN | | Places | | Textures | |
|---|---|---|---|---|---|---|---|---|---|
| | | AUROC↑ | FPR95↓ | AUROC↑ | FPR95↓ | AUROC↑ | FPR95↓ | AUROC↑ | FPR95↓ |
| GradNorm (Huang et al., 2021) | Overall | 82.51 | 72.19 | 74.57 | 78.10 | 70.67 | 86.58 | 57.31 | 84.95 |
| | Head | 82.68 | 66.48 | 73.13 | 73.18 | 68.86 | 82.66 | 51.61 | 83.59 |
| | Mid | 83.85 | 69.50 | 75.88 | 75.68 | 71.91 | 84.41 | 55.99 | 85.48 |
| | Tail | 82.51 | 75.77 | 75.13 | 80.78 | 71.64 | 88.85 | 58.62 | 86.62 |
| RP+GradNorm(Ours) | Overall | 91.23(+8.72) | 43.83(-28.36) | 77.05(+2.48) | 74.20(-3.90) | 72.36(+1.69) | 83.73(-2.85) | 63.05(+5.74) | 79.95(-5.00) |
| | Head | 87.56(+4.88) | 51.94(-14.54) | 68.13(-5.00) | 78.77(+5.59) | 61.71(-7.15) | 88.02(+5.36) | 52.56(+0.94) | 83.39(-0.20) |
| | Mid | 91.42(+7.56) | 42.26(-27.24) | 76.87(+0.99) | 73.40(-2.29) | 72.12(+0.21) | 83.30(-1.11) | 63.32(+7.33) | 79.92(-5.57) |
| | Tail | 93.38(+10.87) | 37.58(-38.19) | 81.82(+6.69) | 70.33(-10.45) | 78.51(+6.87) | 80.98(-7.87) | 69.40(+10.78) | 77.01(-9.61) |

Table 7: Confusion matrix in some tailed categories.

| Class id | GradNorm | RP+GradNorm |
|---|---|---|
| 88 | $\begin{bmatrix} 47 & 14 \\ 3 & 8 \end{bmatrix}$ | $\begin{bmatrix} 47 & 4 \\ 3 & 18 \end{bmatrix}$ |
| 94 | $\begin{bmatrix} 33 & 106 \\ 17 & 182 \end{bmatrix}$ | $\begin{bmatrix} 29 & 61 \\ 21 & 227 \end{bmatrix}$ |
| 671 | $\begin{bmatrix} 35 & 2 \\ 15 & 1 \end{bmatrix}$ | $\begin{bmatrix} 41 & 2 \\ 9 & 1 \end{bmatrix}$ |

Table 8: Performance comparison under random sampling. All methods are based on ResNet101 trained on different imbalanced ID dataset with tail index $a = 8$. The results are means $\pm$ standard errors among ten randomly sampled datasets. ↑ indicates larger values are better and ↓ indicates smaller values are better. All values are percentages.

| Method | iNaturalist | | SUN | | Places | | Textures | |
|---|---|---|---|---|---|---|---|---|
| | AUROC↑ | FPR95↓ | AUROC↑ | FPR95↓ | AUROC↑ | FPR95↓ | AUROC↑ | FPR95↓ |
| MSP (Hendrycks & Gimpel, 2017) | 65.19±0.95 | 93.77±0.58 | 66.18±2.91 | 91.49±0.62 | 59.83±1.19 | 93.44±0.47 | 46.94±0.81 | 97.16±0.26 |
| ODIN (Liang et al., 2018) | 59.30±0.86 | 96.99±0.31 | 66.82±3.36 | 94.2±0.48 | 59.84±1.35 | 94.65±0.40 | 39.79±0.62 | 98.32±0.10 |
| Mahalanobis (Lee et al., 2018b) | 52.70±1.05 | 94.56±0.72 | 50.12±5.09 | 95.79±1.32 | 49.72±3.50 | 95.66±1.04 | 54.79±3.94 | 91.36±2.52 |
| Energy (Liu et al., 2020) | 53.96±0.77 | 98.20±0.17 | 65.44±3.72 | 96.26±0.49 | 58.67±1.46 | 95.54±0.51 | 37.09±0.76 | 99.07±0.11 |
| GradNorm (Huang et al., 2021) | 81.03±0.56 | 68.73±1.40 | 76.03±1.77 | 70.39±1.74 | 70.87±0.77 | 83.39±1.18 | 61.78±0.83 | 79.80±0.80 |
| RP+GradNorm(Ours) | **83.48±1.03** | **61.93±2.26** | **78.74±2.43** | **62.66±1.81** | **74.20±0.78** | **77.35±1.42** | **64.11±0.73** | **76.80±0.69** |
| Cosine Similarity(Ours) | 71.99±7.05 | 84.29±13.22 | 71.72±3.87 | 82.04±3.06 | 66.63±4.70 | 87.99±3.45 | 51.46±4.44 | 92.95±1.93 |

Table 9: OOD detection performance with model size increasing. The RP+GradNorm method is trained on ImageNet-LT-a8. All values are percentages.

| Model | iNaturalist | | SUN | | Places | | Textures | | Average | |
|---|---|---|---|---|---|---|---|---|---|---|
| | AUROC↑ | FPR95↓ | AUROC↑ | FPR95↓ | AUROC↑ | FPR95↓ | AUROC↑ | FPR95↓ | AUROC↑ | FPR95↓ |
| ResNet50 | 89.18 | 51.42 | 81.55 | 73.25 | 77.47 | 78.38 | 62.41 | 87.59 | 77.65 | 72.66 |
| ResNet101 | 90.34 | 47.35 | 81.30 | 75.55 | 77.84 | 80.01 | 62.04 | 88.05 | 77.88 | 72.74 |
| ResNet152 | 89.19 | 50.08 | 79.53 | 79.20 | 76.25 | 82.63 | 61.41 | 89.17 | 76.59 | 75.27 |
| MobileNet | 82.55 | 65.76 | 79.73 | 69.31 | 73.84 | 79.47 | 69.09 | 78.21 | 76.30 | 73.19 |

### A.1.4 Cosine Similarity as A Score Function

We can also regard the cosine similarity weights in the RW strategy as a score function, and conduct several experiments in Table 4, Table 8 and Table 9. We notice that the cosine similarity also achieves a significant improvement compared with baselines in main evaluation tasks. Yet we notice that the cosine similarity is sensitive to the ID data distribution, since the performance in random sampling experiments (see Table 8) is not good enough compared with RP+GradNorm.

### A.1.5 Performance Evaluation under Random Sampling

To further evaluate the performance of our method RP+GradNorm, we conduct experiments in 10 different ID datasets, which are generated randomly by the Pareto distribution with $a = 2$ and $a = 8$ from ImageNet-1K. The results on ImageNet-a8 dataset are reported in Table 8. From this table, it can be observed that, our method RP+GradNorm can outperform baseline methods on all evaluation tasks. As a highlight, RP+GradNorm reduces FPR95 from 70.39% to 62.66%. All these results show the our method RP+GradNorm still outperforms all baselines under random sampling. ImageNet-a2 dataset is more similar to the balanced dataset. The results are reported in Table 10. It can be observed that, our method RP+GradNorm can still outperform baseline methods on all evaluation tasks.

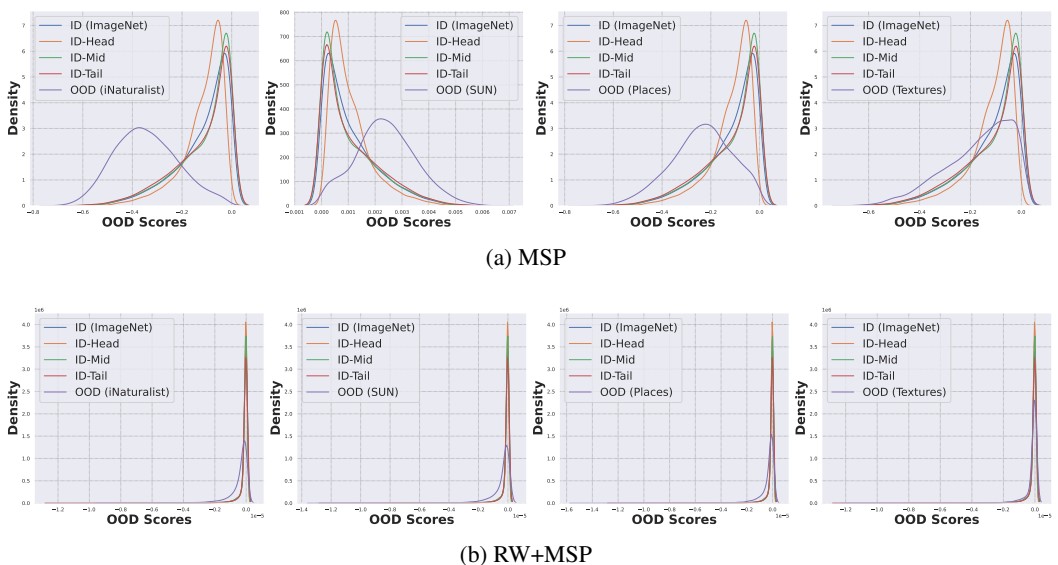

(a) MSP

(b) RW+MSP

Figure 8: OOD score distribution of (a) MSP and (b) RP+MSP.

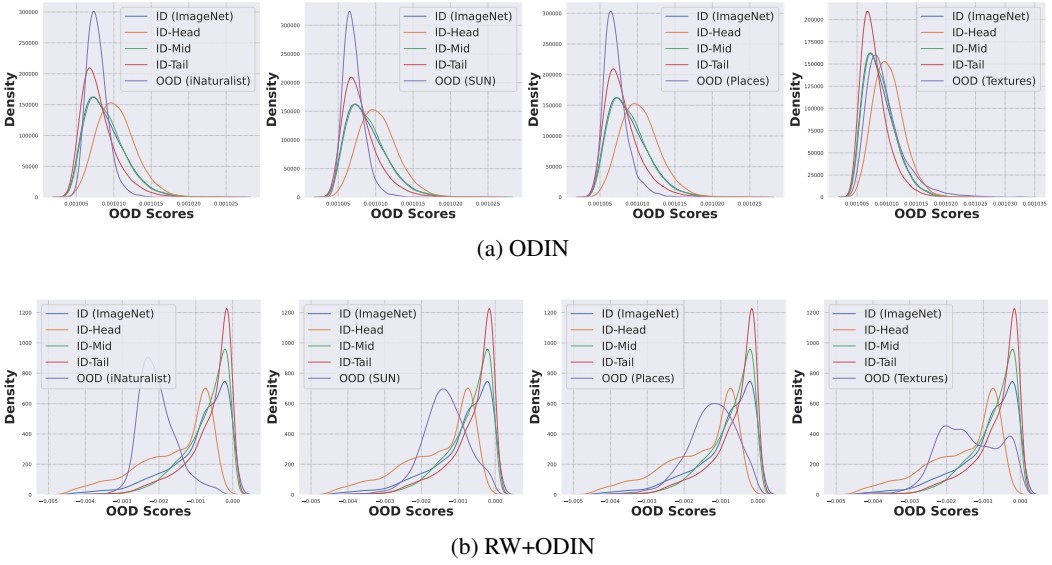

(a) ODIN

(b) RW+ODIN

Figure 9: OOD score distribution of (a) ODIN and (b) RW+ODIN.

Table 10: Performance comparison under random sampling. All methods are based on ResNet101 trained on different imbalanced ID dataset with tail index $a = 2$. The results are means $\pm$ standard errors among ten randomly sampled datasets. $\uparrow$ indicates larger values are better and $\downarrow$ indicates smaller values are better. All values are percentages.

| Method | iNaturalist | | SUN | | Places | | Textures | |
|---|---|---|---|---|---|---|---|---|
| | AUROC↑ | FPR95↓ | AUROC↑ | FPR95↓ | AUROC↑ | FPR95↓ | AUROC↑ | FPR95↓ |
| MSP (Hendrycks & Gimpel, 2017) | 72.55±0.39 | 87.64±0.59 | 68.23±0.35 | 87.33±0.27 | 64.82±0.25 | 90.21±0.27 | 50.58±0.48 | 96.02±0.26 |
| ODIN (Liang et al., 2018) | 70.70±0.63 | 89.41±0.75 | 72.03±0.43 | 85.31±0.55 | 67.26±0.38 | 89.28±0.46 | 44.86±0.69 | 95.57±0.28 |
| Energy (Liu et al., 2020) | 67.36±0.89 | 91.14±0.83 | 72.52±0.63 | 85.27±0.88 | 67.33±0.54 | 89.67±0.61 | 43.19±0.74 | 95.42±0.27 |
| GradNorm (Huang et al., 2021) | 83.66±0.93 | 61.90±2.25 | 78.40±0.76 | 66.78±1.31 | 72.09±0.67 | 79.47±1.09 | 61.80±0.72 | 80.21±0.88 |
| RP+GradNorm(Ours) | **83.77±0.95** | **61.83±2.30** | **78.84±0.77** | **65.79±1.40** | **72.50±0.69** | **78.65±1.17** | **61.91±0.74** | **80.12±0.89** |

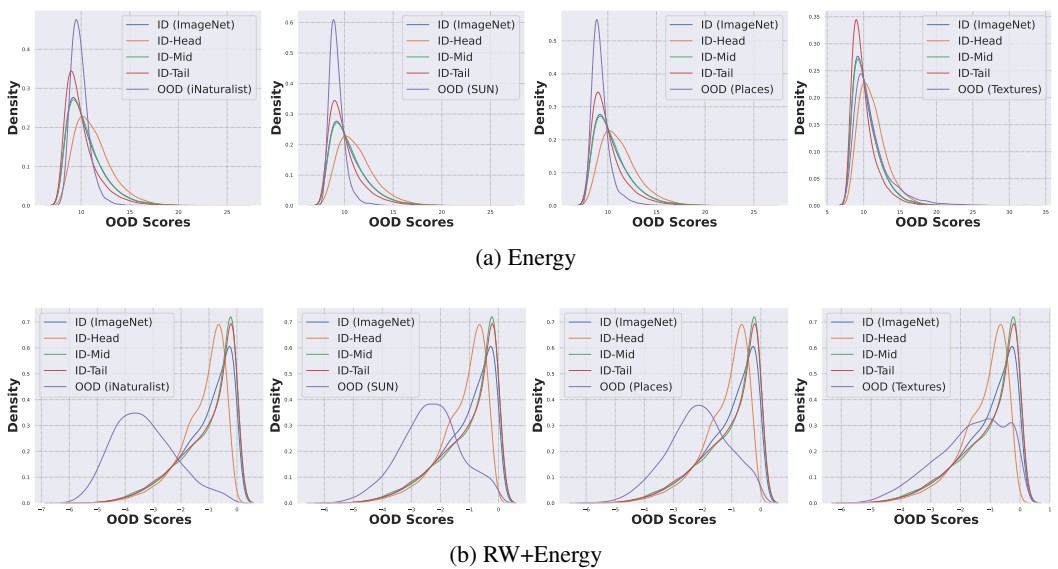

(a) Energy

(b) RW+Energy

Figure 10: OOD score distribution of (a) Energy and (b) RW+Energy.

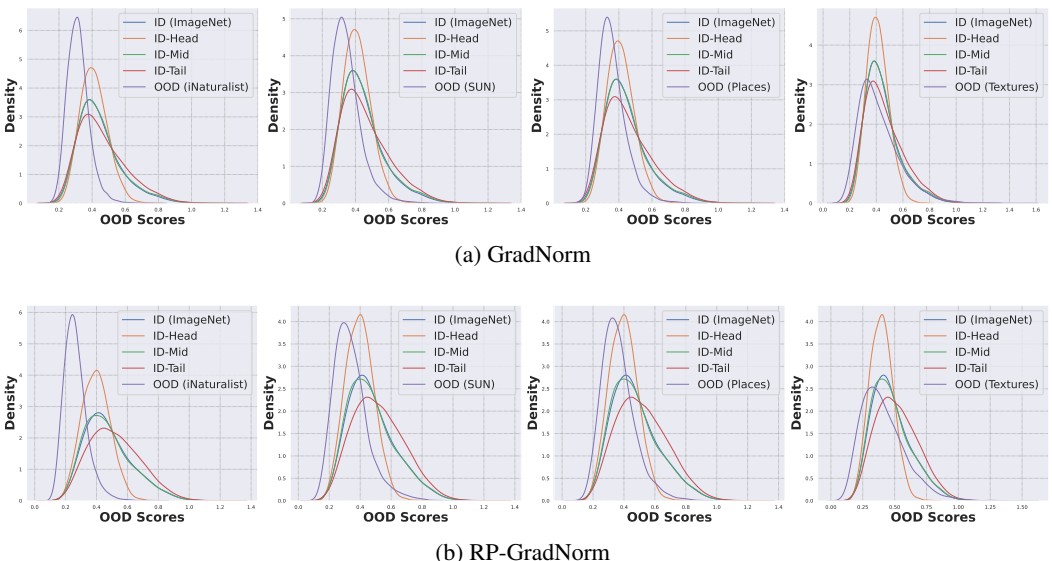

(a) GradNorm

(b) RP-GradNorm

Figure 11: OOD score distribution of (a) GradNorm and (b) RW+GradNorm.

### A.1.6 ABLATION STUDY BETWEEN PROPOSED STRATEGIES

To further explore the collocation of existing methods and our design strategy, we conduct experiments regarding RW+MSP, RW+GradNorm, RW+RP+MSP/GradNorm, as shown in the table 11. Since the cosine distance is superior to the performance of MSP, RW strategy can modify the output of MSP to improve it better than RP strategy. However, for GradNorm, RP strategy performs better than RW strategy because the cosine distance does not perform much better than GradNorm. In contrast, RP strategy better matches the idea of the original method. Therefore, we suggest that the choice of the two strategies follows the idea of preferentially matching the original method. As for RP+RW+MSP/GradNorm, they do not perform well. Taking GradNorm as an example, we think the main reason for this phenomenon is that RP+GradNorm has an outstanding performance, but RW strategy performance is worse than RP+GradNorm. Thus, adding RW strategy to RP+GradNorm has a negative effect, so the performance is significantly reduced. On the other hand, as for MSP,

Table 11: Ablation study between proposed strategies. All methods are trained on ImageNet-LT-a8 dataset with ResNet101.

| Method | Strategy | | iNaturalist | | SUN | | Places | | Textures | | Average | |
|---|---|---|---|---|---|---|---|---|---|---|---|---|
| | RP | RW | AUROC↑ | FPR95↓ | AUROC↑ | FPR95↓ | AUROC↑ | FPR95↓ | AUROC↑ | FPR95↓ | AUROC↑ | FPR95↓ |
| MSP | w/o | w/o | 63.95 | 97.72 | 66.60 | 93.13 | 66.84 | 92.11 | 42.74 | 98.79 | 60.03 | 95.44 |
| | w/ | w/o | 64.95 | 96.44 | 67.39 | 91.79 | 67.46 | 91.16 | 43.05 | 98.51 | 60.71 | 94.48 |
| | w/o | w/ | **81.77** | **78.77** | **77.83** | **78.10** | **75.67** | **81.98** | **51.24** | **90.44** | **71.63** | **82.32** |
| | w/ | w/ | 67.40 | 94.79 | 68.23 | 91.12 | 68.30 | 90.34 | 43.71 | 98.09 | 61.91 | 93.58 |
| GradNorm | w/o | w/o | 82.51 | 72.19 | 74.57 | 78.10 | 70.67 | 86.58 | 57.31 | 84.95 | 71.26 | 80.45 |
| | w/ | w/o | **91.23** | **43.87** | 77.36 | **73.53** | 72.67 | 83.29 | **62.94** | **79.80** | 76.05 | **70.12** |
| | w/o | w/ | 88.96 | 56.32 | **80.05** | 82.45 | **77.47** | 83.93 | 62.08 | 92.39 | **77.14** | 78.77 |
| | w/ | w/ | 82.56 | 85.92 | 77.89 | 89.06 | 76.30 | **88.23** | 58.73 | 96.37 | 73.87 | 89.89 |

Table 12: OOD detection performances with different level of noises.

| Method | Noise Intensity $k$ | iNaturalist | | SUN | | Places | | Textures | | Average | |
|---|---|---|---|---|---|---|---|---|---|---|---|
| | | AUROC↑ | FPR95↓ | AUROC↑ | FPR95↓ | AUROC↑ | FPR95↓ | AUROC↑ | FPR95↓ | AUROC↑ | FPR95↓ |
| RP+MSP | 0 | 64.95 | 96.44 | 67.39 | 91.79 | 67.46 | 91.16 | 43.05 | 98.51 | 60.71 | 94.48 |
| | 0.1 | 65.12 | 96.40 | 67.24 | 91.92 | 67.55 | 90.97 | 42.88 | 98.49 | 60.70 | 94.45 |
| | 0.2 | 65.06 | 96.41 | 67.19 | 92.07 | 67.51 | 91.08 | 42.86 | 98.51 | 60.65 | 94.52 |
| | 0.5 | 64.94 | 96.53 | 67.04 | 92.29 | 67.39 | 91.51 | 42.82 | 98.60 | 60.50 | 94.74 |
| | 1 | 64.80 | 96.75 | 67.04 | 92.23 | 67.38 | 91.32 | 42.79 | 98.67 | 60.28 | 95.11 |
| | 2 | 64.48 | 97.15 | 66.75 | 92.82 | 67.18 | 91.73 | 42.72 | 98.76 | 60.28 | 95.11 |
| RW+ODIN | 0 | 86.66 | 93.85 | 71.59 | 97.67 | 67.56 | 97.24 | 68.04 | 95.37 | 73.46 | 96.03 |
| | 0.1 | 74.81 | 82.02 | 63.77 | 95.77 | 62.39 | 96.07 | 52.72 | 93.69 | 63.42 | 91.89 |
| | 0.2 | 73.16 | 81.94 | 62.75 | 96.25 | 61.46 | 96.68 | 52.55 | 94.27 | 62.48 | 92.29 |
| | 0.5 | 65.59 | 83.52 | 57.79 | 96.08 | 56.86 | 96.46 | 51.25 | 94.36 | 57.87 | 92.61 |
| | 1 | 65.83 | 84.55 | 58.22 | 95.89 | 57.31 | 96.22 | 50.84 | 94.75 | 58.05 | 92.85 |
| | 2 | 65.75 | 86.32 | 58.25 | 95.87 | 57.49 | 95.77 | 50.28 | 94.45 | 57.94 | 93.10 |
| RW+Energy | 0 | 91.92 | 37.89 | 80.81 | 76.22 | 77.15 | 81.18 | 64.48 | 86.19 | 78.59 | 70.37 |
| | 0.1 | 91.79 | 37.93 | 80.65 | 76.82 | 77.12 | 81.24 | 64.33 | 86.49 | 78.47 | 70.62 |
| | 0.2 | 91.76 | 38.06 | 80.65 | 76.90 | 77.21 | 81.29 | 64.07 | 86.38 | 78.42 | 70.66 |
| | 0.5 | 91.41 | 39.09 | 80.31 | 77.42 | 76.91 | 81.51 | 62.79 | 87.38 | 77.86 | 71.35 |
| | 1 | 89.53 | 45.44 | 79.06 | 78.57 | 76.44 | 81.36 | 60.48 | 89.17 | 76.38 | 73.63 |
| | 2 | 87.24 | 51.32 | 76.98 | 79.61 | 74.70 | 82.04 | 58.05 | 88.92 | 74.24 | 75.47 |
| RP+GradNorm | 0 | 91.23 | 43.87 | 77.36 | 73.53 | 72.67 | 83.29 | 62.94 | 79.80 | 76.05 | 70.12 |
| | 0.1 | 91.08 | 44.37 | 77.02 | 74.18 | 72.52 | 83.30 | 62.46 | 80.11 | 75.77 | 70.49 |
| | 0.2 | 90.89 | 44.70 | 76.69 | 74.65 | 72.22 | 83.67 | 62.24 | 80.09 | 75.51 | 70.78 |
| | 0.5 | 90.27 | 47.63 | 76.43 | 75.06 | 72.07 | 83.79 | 61.76 | 80.21 | 75.13 | 71.67 |
| | 1 | 89.89 | 48.70 | 75.66 | 75.94 | 71.24 | 84.70 | 61.37 | 80.59 | 74.54 | 72.48 |
| | 2 | 88.05 | 55.07 | 74.82 | 76.33 | 70.51 | 85.30 | 60.72 | 80.60 | 73.52 | 74.33 |

the RP strategy has a negative impact on RW+MSP, resulting in the degradation of OOD detection performance.

### A.1.7 ROBUSTNESS TO INACCURATE CLASS PRIOR DISTRIBUTION

We further consider what will happen if the ID-class prior is not accurate in the practical applications. In this regard, we conduct relevant experiments. Specifically, we simulate this kind of error regarding prior by adding noises with different intensities to the ID-class-prior distribution. Assume that the standard deviation of the ID-class-prior distribution is std. The noises we add will follow $N(0, k \cdot \text{std})$, where $k$ controls noise intensity. The results are shown in Table 12. After adding noise, the performances of OOD detection do decrease, but not much, which shows that our method is robust to inaccurate class prior distributions.

### A.1.8 OOD DETECTION WITH LONG-TAILED LEARNING

We train ResNet50 with some long-tailed methods, like LDAM (Cao et al., 2019) and CMO (Park et al., 2022). Then we evaluate it with different OOD detection methods and our strategies, as shown in Table 13. The models with LDAM loss do perform better than those with CrossEntropy loss, but after applying our strategies, there are also significant improvements in all methods. However, CMO does not bring the performance improvement of OOD detection as LDAM does, and even performs worse than CrossEntropy. We think that this phenomenon indicates that not all long-tailed training methods are helpful to improve the OOD detector. But the results show that our strategies still works well while the models try to overcome the class imbalance in training time.

At last, We would like to reiterate our view on class-imbalanced OOD detection:

Table 13: OOD detection performance with long-tailed learning methods.

| Method | Long-tailed Method | iNaturalist | | SUN | | Places | | Textures | | Average | |
|---|---|---|---|---|---|---|---|---|---|---|---|
| | | AUROC↑ | FPR95↓ | AUROC↑ | FPR95↓ | AUROC↑ | FPR95↓ | AUROC↑ | FPR95↓ | AUROC↑ | FPR95↓ |
| MSP | CrossEntropy | 62.50 | 97.30 | 67.46 | 92.04 | 66.89 | 91.49 | 42.64 | 98.44 | 59.87 | 94.82 |
| RP+MSP | CrossEntropy | 63.45 | 95.88 | 67.83 | 90.85 | 67.41 | 90.47 | 42.74 | 98.26 | 60.36 | 93.87 |
| MSP | LDAM | 68.12 | 95.22 | 70.04 | 89.39 | 69.72 | 87.71 | 43.04 | 98.07 | 62.73 | 92.60 |
| RP+MSP | LDAM | 68.95 | 93.66 | **70.67** | **88.07** | **70.19** | **86.87** | 43.27 | 97.78 | **63.27** | 91.60 |
| MSP | CMO | **71.13** | **83.77** | 59.35 | 93.81 | 60.35 | 92.77 | **49.13** | **93.58** | 59.99 | **90.98** |
| RP+MSP | CMO | 70.82 | 84.59 | 58.92 | 94.62 | 59.96 | 93.49 | 48.99 | 93.85 | 59.67 | 91.64 |
| ODIN | CrossEntropy | 59.34 | 98.59 | 72.57 | 91.68 | 70.85 | 90.83 | 42.00 | 98.16 | 61.19 | 94.81 |
| RW+ODIN | CrossEntropy | 84.04 | 76.58 | 68.25 | 95.94 | 65.69 | 96.43 | 53.00 | 92.16 | 67.75 | 90.28 |
| ODIN | LDAM | 64.08 | 98.28 | 74.40 | **90.79** | **72.93** | **89.01** | 39.74 | 98.37 | 62.79 | 94.11 |
| RW+ODIN | LDAM | **85.94** | **71.28** | **68.59** | 96.92 | 66.77 | 96.63 | 50.30 | 93.53 | **67.90** | 89.59 |
| ODIN | CMO | 73.67 | 81.36 | 58.94 | 93.39 | 59.57 | 92.54 | 49.34 | 92.41 | 60.38 | 89.93 |
| RW+ODIN | CMO | 77.19 | 81.55 | 53.21 | 95.26 | 56.84 | 93.90 | **57.03** | **83.67** | 61.07 | **88.60** |
| Energy | CrossEntropy | 56.25 | 98.95 | 73.60 | 91.56 | 71.32 | 90.37 | 42.68 | 98.10 | 60.96 | 94.75 |
| RW+Energy | CrossEntropy | 90.57 | 42.68 | 80.09 | 76.41 | 76.12 | 80.78 | **64.68** | **86.17** | 77.87 | 71.51 |
| Energy | LDAM | 59.93 | 98.80 | 74.74 | 91.99 | 72.74 | 90.39 | 39.51 | 98.37 | 61.73 | 94.89 |
| RW+Energy | LDAM | **92.32** | **35.76** | **82.38** | **74.25** | **78.61** | **79.08** | 63.32 | 86.35 | **79.16** | **68.86** |
| Energy | CMO | 73.24 | 85.77 | 55.50 | 95.64 | 55.64 | 95.54 | 48.93 | 91.56 | 58.33 | 92.13 |
| RW+Energy | CMO | 79.75 | 71.54 | 54.32 | 97.12 | 53.63 | 96.51 | 52.31 | 90.32 | 60.00 | 88.87 |
| GradNorm | CrossEntropy | 80.61 | 74.33 | 78.73 | 68.77 | 72.78 | 82.15 | 58.12 | 81.97 | 72.56 | 76.80 |
| RP+GradNorm | CrossEntropy | 89.85 | 50.03 | 80.73 | 64.52 | 74.69 | 78.18 | **63.31** | **77.73** | 77.14 | 67.62 |
| GradNorm | LDAM | 87.05 | 58.11 | 81.20 | 66.00 | 76.01 | 78.42 | 55.95 | 81.86 | 75.05 | 71.10 |
| RP+GradNorm | LDAM | **92.87** | **36.75** | **82.70** | **63.74** | **77.41** | **77.21** | 60.50 | 79.01 | **78.37** | **64.18** |
| GradNorm | CMO | 78.56 | 77.25 | 57.91 | 95.74 | 56.94 | 96.06 | 52.86 | 89.79 | 61.57 | 89.71 |
| RP+GradNorm | CMO | 84.90 | 63.05 | 63.65 | 92.25 | 61.24 | 94.42 | 59.17 | 86.06 | 67.24 | 83.95 |

- Data imbalance is a common phenomenon, and even a slight imbalance (like the ImageNet-LT-a2 dataset) can still lead to a decrease in the performance of the OOD detector. After applying our strategies, this phenomenon can be improved.

- Developers do not necessarily use strategies to overcome data imbalance during the training phase of the model, depending on whether developers need to pay more attention to the minority in specific applications.

- Even if developers use strategies to overcome data imbalance during the training time, it is very hard to obtain a class-balanced classifier. Experiment results show that our method can achieve performance improvements with or without a strategy to overcome data imbalance.

### A.1.9 EVALUATION ON THE BALANCED DATASET

As for RP strategy, when the training dataset is balanced (the class-prior distribution is uniform distribution in Eqs. 10 and 11), RP+Method will be same as the original method.

For RW strategy, we conduct experiments on full ImageNet dataset (note that it is balanced) and the results are shown in Table 14. The performances of Energy and ODIN are quite close to the performnce of Energy and ODIN using RW strategy on the balanced dataset.

Table 14: OOD detection performances on full ImageNet dataset.

| Method | RW Strategy | iNaturalist | | SUN | | Places | | Textures | | Average | |
|---|---|---|---|---|---|---|---|---|---|---|---|
| | | AUROC | FPR95 | AUROC | FPR95 | AUROC | FPR95 | AUROC | FPR95 | AUROC | FPR95 |
| ODIN | w/o | 89.81 | 45.74 | 83.01 | 63.84 | 82.50 | 66.48 | 81.31 | 65.51 | 84.16 | 60.39 |
| | w/ | 89.87 | 45.37 | 83.05 | 63.59 | 82.54 | 66.32 | 81.36 | 65.27 | 84.20 | 60.14 |
| Energy | w/o | 93.28 | 37.62 | 88.60 | 49.48 | 87.36 | 54.61 | 86.80 | 53.40 | 89.01 | 48.78 |
| | w/ | 93.48 | 35.86 | 88.48 | 50.13 | 87.29 | 54.80 | 86.72 | 53.24 | 88.99 | 48.51 |

### A.2 EVALUATION ON INATURALIST BENCHMARK

### A.2.1 EXPERIMENT SETUP

**Datasets.** We use iNaturalist as ID dataset, which is a real long-tailed dataset, and use Texture as OOD dataset, which has no overlapping classes with the former.

Table 15: OOD detection performance on the iNaturalist benchmark.

| Method | Textures | | | |
|---|---|---|---|---|
| | AUROC↑ | AUPR-IN↑ | AUPR-OUT↑ | FPR95↓ |
| MSP | 51.41 | 91.10 | 9.66 | 98.16 |
| RP+MSP(Ours) | 51.54 | 91.12 | 9.70 | 97.96 |
| ODIN | 37.65 | 86.53 | 7.44 | 99.56 |
| RW+ODIN(Ours) | **70.80** | **94.96** | **19.73** | **85.00** |
| Energy | 30.54 | 84.61 | 6.68 | 99.93 |
| RW+Energy(Ours) | 66.99 | 94.38 | 17.32 | 88.03 |
| GradNorm | 31.29 | 83.84 | 6.75 | 99.61 |
| RP+GradNorm(Ours) | 36.26 | 85.10 | 7.34 | 98.71 |
| Cosine Similarity(Ours) | 63.64 | 93.66 | 14.58 | 91.26 |

**Model and Hyperparameters.** We use *mmclassification*[4] (Contributors, 2020) with Apache-2.0 license to train ID models. The training details of ResNet (He et al., 2016) follow the default setting in *mmclassification*. We use the model trained in ImageNet as the pre-trained model and finetune it in iNaturalist. Note that, all methods are realized by Pytorch 1.60 with CUDA 10.2, where we use several NVIDIA Tesla V100 GPUs.

### A.2.2 EXPERIMENTAL RESULTS

We evaluate our methods and previous methods on the proposed benchmark. The results in Table 15 illustrate that our methods show a significant improvement on OOD detection, compared with previous methods. In addition, the RW strategy appears to be more insensitive to the performance of the original method than the RP strategy. Specifically, the RW strategy is able to bring substantial growth when the original method performs particularly poorly, but the RP strategy is unable to do so.

## B FURTHER ANALYSIS

### B.1 PROOF OF THEOREM 1

**Theorem 1.** If Assumption 1 holds, then $\mathbb{P}_{Y^{\text{in}}|X^{\text{out}}}(y|\mathbf{x}) = \mathbb{P}_{Y^{\text{in}}}(y)$, for any $y \in \mathcal{Y}^{\text{in}}$.

*Proof.* Using Assumption 1 in the second equation, we have

$$\mathbb{P}_{Y^{\text{in}}|X^{\text{out}}}(y|\mathbf{x}) = \frac{\mathbb{P}(Y^{\text{in}} = y \wedge X^{\text{out}} = \mathbf{x})}{\mathbb{P}(X^{\text{out}} = \mathbf{x})} = \frac{\mathbb{P}(Y^{\text{in}} = y)\mathbb{P}(X^{\text{out}} = \mathbf{x})}{\mathbb{P}(X^{\text{out}} = \mathbf{x})} = \mathbb{P}_{Y^{\text{in}}}(y).$$

□

### B.2 ALTERNATIVE CHOICE FOR ID-CLASS-PRIOR DISTRIBUTION

When the labels of the training dataset are not available, we can use the predictions made by the model as an alternative to simulate empirical ID-class-prior distribution. Specifically, for each sample $\mathbf{x_i}$ in training dataset, the prediction made by the model is $\text{softmax}(\mathbf{f_\Theta}(\mathbf{x_i}))$. Thus, we have $\mathbb{P}_{Y^{\text{in}}} = 1/N * \sum_{i=0}^{N} \text{softmax}(\mathbf{f_\Theta}(\mathbf{x_i}))$.

We also conduct experiments to confirm the assumption, and the results are shown in Table 16. Noticeably, OOD detection performances with two kinds of ID-class-prior distribution are similar.

---

[4]https://github.com/open-mmlab/mmclassification

Table 16: Performance comparison of two different ID-class-prior distribution acquisition methods. All methods are trained on ImageNet-LT-a8 dataset with ResNet50.

| Method | ID-class-prior Distribution | iNaturalist | | SUN | | Places | | Texture | | Average | |
|---|---|---|---|---|---|---|---|---|---|---|---|
| | | AUROC↑ | FPR95↓ | AUROC↑ | FPR95↓ | AUROC↑ | FPR95↓ | AUROC↑ | FPR95↓ | AUROC↑ | FPR95↓ |
| RP+MSP | Model Prediction | 63.36 | 96.05 | 68.19 | 90.40 | 67.44 | 90.58 | 42.93 | 98.17 | 60.48 | 93.80 |
| | Data Label | 63.34 | 96.09 | 68.19 | 90.44 | 67.43 | 90.64 | 42.93 | 98.17 | 60.48 | 93.84 |
| RW+ODIN | Model Prediction | 84.02 | 76.52 | 68.62 | 95.56 | 65.85 | 96.28 | 53.21 | 90.22 | 67.93 | 89.65 |
| | Data Label | 84.04 | 76.49 | 68.57 | 95.59 | 65.81 | 96.29 | 53.23 | 91.99 | 67.91 | 90.09 |
| RW+Energy | Model Prediction | 90.64 | 42.30 | 80.51 | 75.14 | 76.39 | 80.29 | 64.64 | 85.78 | 78.05 | 70.88 |
| | Data Label | 90.56 | 42.69 | 80.46 | 75.34 | 76.27 | 80.48 | 64.79 | 85.90 | 78.02 | 71.10 |
| RP+GradNorm | Model Prediction | 89.89 | 49.89 | 80.75 | 64.33 | 74.70 | 78.11 | 63.36 | 77.70 | 77.17 | 67.51 |
| | Data Label | 89.85 | 50.03 | 80.73 | 64.52 | 74.69 | 78.18 | 63.31 | 77.73 | 77.14 | 67.60 |

### B.3 PROOF OF THEOREM 2

*Proof of Theorem 2.* According to the definition of softmax function, it is clear that

$$\sum_{i=1}^{K} \text{softmax}_i(\mathbf{f}_{\Theta}(\mathbf{x})) = 1 \text{ and } \text{softmax}_i(\mathbf{f}_{\Theta}(\mathbf{x})) \geq 0, \text{ for } \forall\, i = 1, ..., K.$$

**Existence.** If we assume that for all $i = 1, ..., K$, $\text{softmax}_i(\mathbf{f}_{\Theta}(\mathbf{x})) < \frac{1}{K}$, then

$$\sum_{i=1}^{K} \text{softmax}_i(\mathbf{f}_{\Theta}(\mathbf{x})) < 1, \text{ which is conflict with } \sum_{i=1}^{K} \text{softmax}_i(\mathbf{f}_{\Theta}(\mathbf{x})) = 1.$$

Therefore, there is at least one $i$ such that $\text{softmax}_i(\mathbf{f}_{\Theta}(\mathbf{x})) \geq \frac{1}{K}$, which implies that

$$\min_{\mathbf{f}_{\Theta}(\mathbf{x})} S_{\text{MSP}}(\mathbf{f}_{\Theta}, \mathbf{x}) \geq \frac{1}{K}.$$

Note that when $\text{softmax}(\mathbf{f}_{\Theta}(\mathbf{x})) = \mathbf{u}$, $S_{\text{MSP}}(\mathbf{f}_{\Theta}, \mathbf{x}) = \frac{1}{K}$, which implies that there exists $\tilde{\mathbf{f}}_{\Theta}(\mathbf{x}) \in \arg\min_{\mathbf{f}_{\Theta}(\mathbf{x})} S_{\text{MSP}}(\mathbf{f}_{\Theta}, \mathbf{x})$ such that

$$\mathbf{u} = \text{softmax}(\tilde{\mathbf{f}}_{\Theta}(\mathbf{x})).$$

**Uniqueness.** If there is $\mathbf{f}^*_{\Theta}(\mathbf{x}) \in \arg\min_{\mathbf{f}_{\Theta}(\mathbf{x})} S_{\text{MSP}}(\mathbf{f}_{\Theta}, \mathbf{x})$ such that $\text{softmax}(\mathbf{f}^*_{\Theta}(\mathbf{x})) \neq \mathbf{u}$, it is clear that

$$\sum_{i=1}^{K} \text{softmax}_i(\mathbf{f}^*_{\Theta}(\mathbf{x})) < 1,$$

which is conflict with $\sum_{i=1}^{K} \text{softmax}_i(\mathbf{f}_{\Theta}(\mathbf{x})) = 1$. Therefore, $\text{softmax}(\mathbf{f}^*_{\Theta}(\mathbf{x})) = \mathbf{u}$.

Combining the results in existence and uniqueness, we have completed this proof. □

### B.4 DISCUSSION ABOUT RP+ODIN

ODIN (Liang et al., 2018) is an enhanced version of MSP, whose main improvement is the introduction of a temperature scaling strategy. The temperature parameter $T$ smoothes the prediction distribution of the softmax function and thus making the prediction sparser and more similar to the uniform distribution.

$$S_{\text{ODIN}}(\mathbf{f}_{\Theta}, \mathbf{x}) = \max_i \frac{\exp\left(\mathbf{f}_i(\mathbf{x})/T\right)}{\sum_{j=1}^{C} \exp\left(\mathbf{f}_j(\mathbf{x})/T\right)} \tag{16}$$

Since ODIN maps the prediction distribution of the softmax layer to another distribution space while we need to measure the similarity between the class-prior distribution and the model-predicted distribution, we need to use the same mapping method to deal with the class-prior distribution $\mathbb{P}_{Y^{\text{in}}} = [p_1, p_2, ..., p_C]$, as follows:

$$\mathbb{P}'_{Y^{\text{in}}} = \left[ \frac{\exp\left(p_1/T\right)}{\sum_{j=1}^{C} \exp\left(p_j/T\right)}, \frac{\exp\left(p_2/T\right)}{\sum_{j=1}^{C} \exp\left(p_j/T\right)}, \cdots, \frac{\exp\left(p_C/T\right)}{\sum_{j=1}^{C} \exp\left(p_j/T\right)} \right] \tag{17}$$

Then, we use this new class-prior distribution $\mathbb{P}'_{Y^{\text{in}}}$ to modify ODIN with RP strategy as Eq. (19).

$$\mathbf{h}_{\Theta}(\mathbf{x}) = \left[ \frac{\exp\left(\mathbf{f}_1(\mathbf{x})/T\right)}{\sum_{j=1}^{C} \exp\left(\mathbf{f}_j(\mathbf{x})/T\right)}, \frac{\exp\left(\mathbf{f}_2(\mathbf{x})/T\right)}{\sum_{j=1}^{C} \exp\left(\mathbf{f}_j(\mathbf{x})/T\right)}, \cdots, \frac{\exp\left(\mathbf{f}_C(\mathbf{x})/T\right)}{\sum_{j=1}^{C} \exp\left(\mathbf{f}_j(\mathbf{x})/T\right)} \right] \quad (18)$$

$$S_{\text{RP+ODIN}}(\mathbf{f}_{\Theta}, \mathbf{x}) = \max\left(\mathbf{h}_{\Theta}(\mathbf{x}) - \mathbb{P}'_{Y^{\text{in}}}\right) \quad (19)$$

When we follow the default setting $T = 1000$ in ODIN, we notice that $\mathbb{P}'_{Y^{\text{in}}}$ will be quite close to the uniform distribution, where each element is close to $1/K$. Thus, Eq. (19) can be regarded as $\mathbf{h}_{\Theta}(\mathbf{x})$ minus a constant.

### B.5 DISCUSSION ABOUT M IN PARETO DISTRIBUTION

For each class $x_i$, the sample number is

$$y_i = N \times p(x_i) = N \times \frac{am^a}{x_i^{a+1}}, \quad (20)$$

where $a$ is tail index, $m$ is a constant and $N$ is the sample number of the ImageNet-1K dataset.

After sampling, the new data distribution for each class is

$$p(y_i) = \frac{y_i}{\sum_{i=1}^{K} y_i} = \frac{N \times \frac{am^a}{x_i^{a+1}}}{\sum_{i=1}^{K}(N \times \frac{am^a}{x_i^{a+1}})} = \frac{1}{x_i^{a+1} \sum_{i=1}^{K} \frac{1}{x_i^{a+1}}}. \quad (21)$$

Obviously, the value of $m$ do not affect the imbalance degree of sampled datasets. Thus, we keep $m = 1$ unchanged.

### B.6 DISCUSSION ABOUT FEATURE-BASED METHODS

Feature-based methods, like KNN (Sun et al., 2022), need a training set to generate class prototypes, i.e., an average feature vector for each category. Under class-imbalanced situations, prototypes of tailed classes would be more unreliable than the majority due to the limitation of training samples. We think using ID-class-prior distribution to reweight features may be an effective way to solve the imbalanced problem in feature space.

### B.7 DISCUSSION ABOUT CLASS-DEPENDENT THRESHOLDING

The paper (Guarrera et al., 2022) designs an optimization for threshold selection rather than designing an OOD score. Class-dependent thresholding is designed for $p_{train} \neq p_{test}$ and not for data imbalance issues. This approach only affects the precision and recall metrics at the deployment stage of the OOD detector, not the AUROC and FPR95 metrics that our paper focuses on. This can be seen in Table 1 in Guarrera et al. (2022).

### B.8 DISCUSSION ABOUT POSSIBILITY FOR RP+MSP.

In order to discuss about the possibility for aligning the minimizer of the score function with the class priors, we conduct experiments for $\max_i(\text{softmax}_i(\mathbf{f}(\mathbf{x}))/\mathbb{P}_{Y^{\text{in}}}(i))$ and show the corresponding results in the below table. Our experiments show that $\max_i(\text{softmax}_i(\mathbf{f}(\mathbf{x}))/\mathbb{P}_{Y^{\text{in}}}(i))$ also performs very well.

Table 17: OOD detection performances on ImageNet-LT-a8 dataset.

| Method | iNaturalist | | SUN | | Places | | Textures | | Average | |
|---|---|---|---|---|---|---|---|---|---|---|
| | AUROC | FPR95 | AUROC | FPR95 | AUROC | FPR95 | AUROC | FPR95 | AUROC | FPR95 |
| $\max_i(\text{softmax}_i(\mathbf{f}(\mathbf{x}))/\mathbb{P}_{Y^{\text{in}}}(i))$ | 81.76 | 69.75 | 57.80 | 94.80 | 54.90 | 94.68 | 52.01 | 88.97 | 61.62 | 87.05 |

## C  DETAILED RELATED WORKS

**OOD Detection.** OOD detection is a crucial problem for reliably deploying machine learning models into real-world scenarios. OOD detection can be divided into two categories according to whether the classifier will be re-trained for OOD detection or not.

1) Inference-time/post hoc OOD Detection: Some methods (Huang et al., 2021; Liang et al., 2018; Liu et al., 2020; Hendrycks & Gimpel, 2017; Lee et al., 2018b; Sun et al., 2021) focus on designing OOD score functions for OOD detection in the inference time and are easy to use without changing the model's parameters. This property is important for deploying OOD detection methods in real-world scenarios where the cost of re-training is prohibitively expensive and time-consuming. MSP (Hendrycks & Gimpel, 2017) directly takes the maximum value of the model's prediction as the OOD score function. Based on MSP, ODIN (Liang et al., 2018) uses a temperature scaling strategy and input perturbation to improve OOD detection performance. Moreover, Liu et al. (2020) and Wang et al. (2021a) propose to replace the softmax function with the energy functions for OOD detection. Recently, GradNorm (Huang et al., 2021) uses the similarity of the model-predicted probability distribution and the uniform distribution to improve OOD detection and achieve state-of-the-art performance. In this paper, we mainly work on the inference-time OOD detection methods and aim at improving the generalizability of OOD detection in real-world scenarios.

2) Training-time OOD Detection: Other methods (Hsu et al., 2020; Hein et al., 2019; Bitterwolf et al., 2020; Wang et al., 2021b) will complete ID tasks and OOD detection simultaneously in the training time. Bitterwolf et al. (2020) uses adversarial learning to process OOD data in training time and make the model predict lower confidence scores for them. Wang et al. (2021b) generates pseudo OOD data by adversarial learning to re-training a K+1 model for OOD detection. These methods usually require auxiliary OOD data available in the training process. Thus, the model will be affected by both ID data and OOD data. It is important for these method to explore an inherent trade-off (Liu et al., 2019; Vaze et al., 2022; Yang et al., 2021) between ID tasks and OOD detection.

The paper (Wang et al., 2022) is training-time OOD detection and uses OOD data to train the model. After being finetuned, the model can deal with the imbalanced issue and OOD problem. The problem is similar to our paper, but the setting is completely different with our paper (inference-time OOD detection). In our paper, we do not change any parameters of the model and design methods to deal with the imbalanced issue on OOD detection. Note that our work and this work are not comparable due to the different problem settings.

**Open Set Recognition.** In open set recognition, machine learning models (Huang & Li, 2021; Lee et al., 2018a; Perera & Patel, 2019; Perera et al., 2020; Shalev et al., 2018; Radford et al., 2021; Fort et al., 2021) are required to both correctly classify the known data (ID) from the closed set and detect unknown data (OOD) from the open set. Some works (Lee et al., 2018a; Huang & Li, 2021) use the information in the label space for OOD detection, and they divide the large semantic space into multiple levels for models to easily understand. Perera & Patel (2019) designs two parallel networks training on different dataset and use the membership loss to encourage high activations for ID data while reducing activations for OOD data. Perera et al. (2020) uses self-supervision and data augmentation to improve the network's ability to detect OOD data. Input images are augmented with the representation obtained from a generative model. In this paper, we consider a more complex open set, large scale and imbalanced, to achieve OOD detccion.

## D  CODES

```python
import torch
import numpy as np
from torch.autograd import Variable

def MSP(data_loader, model):
    ood_scores = []
    m = torch.nn.Softmax(dim=-1).cuda()
    for b, (x, y) in enumerate(data_loader):
```

```python
        with torch.no_grad():
            x = x.cuda()
            logits, _ = model_forward(model, x)
            softmax_output = m(logits)
            ood_score, _ = torch.max(softmax_output, dim=-1)
            ood_scores.extend(ood_score.data)
    return ood_scores

def MSP_RP(data_loader, model, ID_Prior):
    ood_scores = []
    m = torch.nn.Softmax(dim=-1).cuda()
    for b, (x, y) in enumerate(data_loader):
        with torch.no_grad():
            x = x.cuda()
            logits, _ = model_forward(model, x)
            softmax_output = m(logits)
            # RP strategy
            sim = softmax_output - ID_Prior
            ood_score, _ = torch.max(sim, dim=-1)
            ood_scores.extend(ood_score.data)
    return ood_scores

def ODIN(data_loader, model, epsilon, temper):
    criterion = torch.nn.CrossEntropyLoss().cuda()
    ood_scores = []
    for b, (x, y) in enumerate(data_loader):
        x = Variable(x.cuda(), requires_grad=True)
        outputs, _ = model_forward(model, x)
        maxIndexTemp = np.argmax(outputs.data.cpu().numpy(), axis=1)
        outputs = outputs / temper
        labels = Variable(torch.LongTensor(maxIndexTemp).cuda())
        loss = criterion(outputs, labels)
        loss.backward()
        # Normalizing the gradient to binary in {0, 1}
        gradient = torch.ge(x.grad.data, 0)
        gradient = (gradient.float() - 0.5) * 2
        # Adding small perturbations to images
        tempInputs = torch.add(x.data, -epsilon, gradient)
        outputs, _ = model_forward(model, Variable(tempInputs))
        outputs = outputs / temper
        # Calculating the confidence after adding perturbations
        nnOutputs = outputs.data.cpu()
        nnOutputs = nnOutputs.numpy()
        nnOutputs = nnOutputs - np.max(nnOutputs, axis=1, keepdims=True)
        nnOutputs = np.exp(nnOutputs) / np.sum(np.exp(nnOutputs),
        ↪  axis=1, keepdims=True)
        ood_scores.extend(np.max(nnOutputs, axis=1))
    return ood_scores

def ODIN_RW(data_loader, model, epsilon, temper, ID_Prior):
    criterion = torch.nn.CrossEntropyLoss().cuda()
    ood_scores = []
```

```python
    m = torch.nn.Softmax(dim=-1).cuda()
    for b, (x, y) in enumerate(data_loader):
        x = Variable(x.cuda(), requires_grad=True)
        outputs, _ = model_forward(model, x)
        softmax_output = m(outputs)
        softmax_output = softmax_output.data.cpu()
        softmax_output = softmax_output.numpy()
        maxIndexTemp = np.argmax(outputs.data.cpu().numpy(), axis=1)
        outputs = outputs / temper
        labels = Variable(torch.LongTensor(maxIndexTemp).cuda())
        loss = criterion(outputs, labels)
        loss.backward()
        # Normalizing the gradient to binary in {0, 1}
        gradient = torch.ge(x.grad.data, 0)
        gradient = (gradient.float() - 0.5) * 2
        # Adding small perturbations to images
        tempInputs = torch.add(x.data, -epsilon, gradient)
        outputs, _ = model_forward(model, Variable(tempInputs))
        outputs = outputs / temper
        # Calculating the confidence after adding perturbations
        nnOutputs = outputs.data.cpu()
        nnOutputs = nnOutputs.numpy()
        nnOutputs = nnOutputs - np.max(nnOutputs, axis=1, keepdims=True)
        nnOutputs = np.exp(nnOutputs) / np.sum(np.exp(nnOutputs),
        ↪  axis=1, keepdims=True)
        # RW strategy
        sim = -softmax_output * ID_Prior
        sim = sim.sum(axis=1) / (np.linalg.norm(nnOutputs, axis=-1) *
        ↪  np.linalg.norm(ID_Prior, axis=-1))
        sim = np.expand_dims(sim, axis=1)
        nnOutputs = sim * nnOutputs
        ood_scores.extend(np.max(nnOutputs, axis=1))
    return ood_scores

def Energy(data_loader, model, temper):
    ood_scores = []
    for b, (x, y) in enumerate(data_loader):
        with torch.no_grad():
            x = x.cuda()
            logits, _ = model_forward(model, x)
            ood_score = temper * torch.logsumexp(logits / temper, dim=1)
            ood_scores.extend(ood_score.data)
    return ood_scores

def Energy_RW(data_loader, model, temper, ID_Prior):
    ood_scores = []
    m = torch.nn.Softmax(dim=-1).cuda()
    for b, (x, y) in enumerate(data_loader):
        with torch.no_grad():
            x = x.cuda()
            logits, _ = model_forward(model, x)
            conf = temper * torch.logsumexp(logits / temper, dim=1)
            # RW strategty
```

```python
            softmax_output = m(logits)
            sim = -softmax_output * ID_Prior
            sim = sim.sum(1) / (torch.norm(softmax_output, dim=1) *
            ↪  torch.norm(ID_Prior, dim=1))
            ood_score = conf * sim
            ood_scores.extend(ood_score.data)
    return ood_scores

def GradNorm(data_loader, model, temperature, num_classes):
    ood_scores = []
    logsoftmax = torch.nn.LogSoftmax(dim=-1).cuda()
    for b, (x, y) in enumerate(data_loader):
        inputs = Variable(x.cuda(), requires_grad=True)
        model.zero_grad()
        outputs, _ = model_forward(model, inputs)
        targets = torch.ones((inputs.shape[0], num_classes)).cuda()
        outputs = outputs / temperature
        loss = torch.sum(torch.mean(-targets * logsoftmax(outputs),
        ↪  dim=-1))
        loss.backward()
        layer_grad = model.head.conv.weight.grad.data
        ood_score = torch.sum(torch.abs(layer_grad))
        ood_scores.append(ood_score)
    return ood_scores

def GradNorm_RP(data_loader, model, temperature, num_classes,ID_Prior):
    ood_scores = []
    logsoftmax = torch.nn.LogSoftmax(dim=-1).cuda()
    for b, (x, y) in enumerate(data_loader):
        inputs = Variable(x.cuda(), requires_grad=True)
        model.zero_grad()
        outputs, _ = model_forward(model, inputs)
        # RP strategy
        target = torch.tensor(ID_Prior)
        targets = target.unsqueeze(dim=0).cuda()
        outputs = outputs / temperature
        loss = torch.sum(-targets * logsoftmax(outputs), dim=-1)
        loss.backward()
        layer_grad = model.fc.weight.grad.data
        ood_score = torch.sum(torch.abs(layer_grad))
        ood_scores.append(ood_score)
    return ood_scores

def Cosine_Similarity(data_loader, model, ID_Prior):
    ood_scores = []
    m = torch.nn.Softmax(dim=-1).cuda()
    for b, (x, y) in enumerate(data_loader):
        with torch.no_grad():
            x = x.cuda()
            logits, _ = model_forward(model, x)
            softmax_output = m(logits)
            sim = -softmax_output * ID_Prior
            sim = sim.sum(1) / (torch.norm(softmax_output, dim=1) *
            ↪  torch.norm(ID_Prior, dim=1))
```

```
        ood_score = sim.unsqueeze(1)
        ood_scores.extend(ood_score.data)
return ood_scores
```

