# OpenReview forum: "On the Importance of In-distribution Class Prior for Out-of-distribution Detection"
_ICLR.cc/2023/Conference — Submitted to ICLR 2023_

### Official Review · Reviewer_vuMZ · 2022-10-24

**Confidence:** 4
**Correctness:** 3
**Technical Novelty And Significance:** 3
**Empirical Novelty And Significance:** 3
**Recommendation:** 8

**Clarity, Quality, Novelty And Reproducibility:**

I think the paper is relatively well written, with strong empirical findings and a strong motivation. My main issue on clarity is that I believe the causal framework proposed to analyse the problem obfuscates rather than clarities the problem.

**Strength And Weaknesses:**

Strengths:
* The authors tackle an interesting problem, OoD detection for long-tailed training data. It is interesting because real-world data is rarely balanced and, as the authors observe, the performance of existing methods drops substantially in this regime.
* The re-weighting strategies which the authors propose are intuitive and easy to implement. It makes sense to incorporate the empirical class prior of the in distribution data to the OoD scoring rule, and the authors show an easy way in which it can be done. Despite this, they markedly improve the performance of a number of baselines.
* The authors provide a detailed ablation, controlling for model architecture, different amounts of data long-tallness, as well as a useful breakdown of the OoD performance when only the 'Head','Middle' or 'Tail' classes are taken from the closed-set categories. Intuitively, they find that the biggest problem is distinguishing 'Tail' closed-set categories from open-set classes.

Weaknesses:
* The main weakness for me is that the authors don't seem to mention or compare to [1], which is published work on the problem of OoD detection for long-tail recognition. This method also reports substantial boosts over baselines (though on small-scale OoD benchmarks). The authors should discuss and compare to this method if possible, or else explain why it cannot be done.
* While I appreciate the efforts to impose a theoretical framework on the problem, I believe it makes the paper more confusing than clarified. I do not think that the proofs and arguments derived from the causal framework add anything to the argument which could not be more clearly expressed through a few lines describing the intuitions.
* (Minor): There are a number of typos throughout the paper. For example, P_xout * P_yin = P_xout * P_yin in Assumption 1.

[1] Partial and Asymmetric Contrastive Learning for Out-of-Distribution Detection in Long-Tailed Recognition, Wang et al., ICML 22

**Summary Of The Paper:**

This paper tackles the problem of OoD detection in the case where the distribution of training classes is not-uniform. In other words, the authors tackle OoD detection in the case of long-tail image recognition. The authors first make the observation that a number of existing OoD methods (either implicitly or explicitly) make the assumption that the training set is balanced, and then propose a simple re-weighting strategies to address this.

On an ImageNet evaluation, with varying degrees of long-tailness, the authors show their re-weighting strategies boost the performance of a number of baselines (including maximum softmax probability and the more recent GradNorm). Specifically, the re-weighting takes into account the empirical distribution of training classes.

The authors theoretically analyze the problem through the lens of causality, by presenting different candidate causal graphs which model the data generation process.

**Summary Of The Review:**

Overall, I think this is an interesting paper which proposes a simple solution to an important problem, which nonetheless demonstrates substantial boosts on a large-scale benchmark. Significant extra exploration is undertaken to provide additional details to explain the method. My main issue is that prior work in this space is not discussed or compared to, and hence I am hesitant to recommend a strong acceptance.

**UPDATE AFTER AUTHOR RESPONSE:**

After reading the other reviews and the (especially) the authors' responses, I have decided to upgrade my rating. Overall, I like this paper because it provides strong empirical results with a very simple method, to tackle an important and realistic problem.

However, I would strongly suggest the following improvements to the paper:
1) The authors should discuss [1]. Though it is a valid distinction that [1] trains with OoD data, it is not a completely different task. For instance, Outlier Exposure is still often referred to inference-only OoD detection papers, with reasonable discussion given on why inference-only methods operate in a different setting.
2) I concur with the other reviewers that the theoretical causal framework does not add much to the paper's narrative, but rather distracts from the main findings.

[1] Partial and Asymmetric Contrastive Learning for Out-of-Distribution Detection in Long-Tailed Recognition, Wang et al., ICML 22

---

> ### Author Response · Authors · 2022-11-15
> **Response to your reviews**
>
> Thanks for your comments! We will answer them as follows:
>
>
> >Q1: The main weakness for me is that the authors don't seem to mention or compare to [R1], which is published work on the problem of OoD detection for long-tail recognition. This method also reports substantial boosts over baselines (though on small-scale OoD benchmarks). The authors should discuss and compare to this method if possible, or else explain why it cannot be done.
>
> **A1**: Thanks for your suggestion! This paper [R1] is training-time OOD detection and uses OOD data to train the model. After being finetuned, the model can deal with the imbalanced issue and OOD detection problem. The setting is completely different with our paper (inference-time OOD detection). In our paper, we do not change any parameters of the model and design methods to deal with the imbalanced issue on OOD detection. Our paper and the metioned paper are not comparable due to different problem settings. We will cite and add more discussions about these two papers in our related work.
>
>
> >Q2: While I appreciate the efforts to impose a theoretical framework on the problem, I believe it makes the paper more confusing than clarified. I do not think that the proofs and arguments derived from the causal framework add anything to the argument which could not be more clearly expressed through a few lines describing the intuitions. My main issue on clarity is that I believe the causal framework proposed to analyse the problem obfuscates rather than clarities the problem.
>
> **A2**:  Causal graphs are used to support Assumption 1 that can hold in most practical cases. According to [R2][R3], we can use three causal graphs to describe most situations in OOD detection. Figure 1$($a$)$ corresponds to images without styles,  Figure 1$($b$)$ corresponds to images with shared styles, and Figure 1$($c$)$ corresponds to images with different styles. Through these graphs, it is clear that $Y^{\rm in}$ and $X^{\rm out}$ are independent. Thus, our assumption 1 is a mild assumption, which holds in these situations.
>
> >Q3(Minor): There are a number of typos throughout the paper. For example, P_xout * P_yin = P_xout * P_yin in Assumption 1.
>
> **A3**：In Assumption 1, $\mathbb{P}_{X^{\rm out}{Y}^{\rm in}}$ represents the joint distribution. Maybe the notation $\mathbb{P}({X^{\rm out},{Y}^{\rm in}})$ is better. We will follow your suggestion to revise it.
>
> **General Response:** We have addressed your concerns about our paper. If you have more suggestions, please discuss them with us in the openreview system. We will try our best to address your further concerns and merge your comments into our revision as well!
>
> **Reference**
>
> [R1] Partial and Asymmetric Contrastive Learning for Out-of-Distribution Detection in Long-Tailed Recognition, Wang et al., ICML 22.
>
> [R2] Mingming Gong, Kun Zhang, Tongliang Liu, Dacheng Tao, Clark Glymour, and Bernhard Schölkopf. Domain adaptation with conditional transferable components. In ICML, 2016 .
>
> [R3] Petar Stojanov, Zijian Li, Mingming Gong, Ruichu Cai, Jaime G. Carbonell, and Kun Zhang. Domain adaptation with invariant representation learning: What transformations to learn? In NeurIPS, 2021.

---

> ### Author Response · Authors · 2022-11-17
> **Looking forward to your responses or further suggestions/comments!**
>
> Dear Reviewer vuMZ,
>
> We have addressed your initial concerns regarding our paper. We are happy to discuss them with you in the openreview system if you feel that there still are some concerns/questions. We also welcome new suggestions/comments from you!
>
> Best regards,
>
> Authors of #3905

---

> ### Author Response · Authors · 2022-11-18
> **Looking forward to your responses or further suggestions/comments!**
>
> Dear Reviewer vuMZ,
>
> We have addressed your initial concerns regarding our paper. We are happy to discuss them with you in the openreview system if you feel that there still are some concerns/questions. We also welcome new suggestions/comments from you!
>
> Best regards,
>
> Authors of #3905

---

> ### Author Response · Authors · 2022-11-19
> **Looking forward to your responses or further suggestions/comments!**
>
> Dear Reviewer vuMZ,
>
> We have addressed your initial concerns regarding our paper. We are happy to discuss them with you in the openreview system if you feel that there still are some concerns/questions. We also welcome new suggestions/comments from you!
>
> Best regards,
>
> Authors of #3905

---

> ### Author Response · Authors · 2022-11-20
> **Looking forward to your responses or further suggestions/comments!**
>
> Dear Reviewer vuMZ,
>
> We have addressed your initial concerns regarding our paper. We are happy to discuss them with you in the openreview system if you feel that there still are some concerns/questions. We also welcome new suggestions/comments from you!
>
> Best regards,
>
> Authors of #3905

---

> ### Author Response · Authors · 2022-11-21
> **Looking forward to your responses or further suggestions/comments!**
>
> Dear Reviewer vuMZ,
>
> We have addressed your initial concerns regarding our paper. We are happy to discuss them with you in the openreview system if you feel that there still are some concerns/questions. We also welcome new suggestions/comments from you!
>
> Best regards,
>
> Authors of #3905

---

> ### Author Response · Authors · 2022-11-22
> **Looking forward to your responses or further suggestions/comments!**
>
> Dear Reviewer vuMZ,
>
> We have addressed your initial concerns regarding our paper. We are happy to discuss them with you in the openreview system if you feel that there still are some concerns/questions. We also welcome new suggestions/comments from you!
>
> Best regards,
>
> Authors of #3905

---

> ### Author Response · Authors · 2022-11-23
> **Looking forward to your responses or further suggestions/comments!**
>
> Dear Reviewer vuMZ,
>
> We have addressed your initial concerns regarding our paper. We are happy to discuss them with you in the openreview system if you feel that there still are some concerns/questions. We also welcome new suggestions/comments from you!
>
> Best regards,
>
> Authors of #3905

---

> ### Author Response · Authors · 2022-11-24
> **Looking forward to your responses or further suggestions/comments!**
>
> Dear Reviewer vuMZ,
>
> We have addressed your initial concerns regarding our paper. We are happy to discuss them with you in the openreview system if you feel that there still are some concerns/questions. We also welcome new suggestions/comments from you!
>
> Best regards,
>
> Authors of #3905

---

> ### Author Response · Authors · 2022-11-25
> **Looking forward to your responses or further suggestions/comments!**
>
> Dear Reviewer vuMZ,
>
> We have addressed your initial concerns regarding our paper. We are happy to discuss them with you in the openreview system if you feel that there still are some concerns/questions. We also welcome new suggestions/comments from you!
>
> Best regards,
>
> Authors of #3905

---

> ### Author Response · Authors · 2022-11-26
> **Looking forward to your responses or further suggestions/comments!**
>
> Dear Reviewer vuMZ,
>
> We have addressed your initial concerns regarding our paper. We are happy to discuss them with you in the openreview system if you feel that there still are some concerns/questions. We also welcome new suggestions/comments from you!
>
> Best regards,
>
> Authors of #3905

---

> ### Author Response · Authors · 2022-12-06
> **Thanks for updating your score from 6 to 8**
>
> Thanks for your suggestions! We will add more discussions about [R1] in our paper. In addition, we will reorganize the paper's structure to introduce our approach directly and then use the causal framework as the theoretical explanation of our approach.
>
> **Reference**
>
> [R1] Partial and Asymmetric Contrastive Learning for Out-of-Distribution Detection in Long-Tailed Recognition, Wang et al., ICML 22

---

### Official Review · Reviewer_G6VC · 2022-10-24

**Confidence:** 3
**Correctness:** 2
**Technical Novelty And Significance:** 2
**Empirical Novelty And Significance:** 3
**Recommendation:** 3

**Clarity, Quality, Novelty And Reproducibility:**

- While the paper contains quite many grammar errors, the general presentation is clear and easy to follow.
- There are not very many previous works on OOD detection with imbalanced classes, and the proposed methods are to my knowledge novel in this area.
- The experiments are described in detail and code for the evaluations in included in the appendix. The model checkpoints are not available; they would be useful to reproduce the exact results and compare them with other scoring methods.

**Strength And Weaknesses:**

Strengths:

- The paper tackles the important problem of transferring OOD detection methods that are known to work for class-balanced distributions to unbalanced ones.

- The experimental results show good improvements compared to the presented implementations of the baseline methods.

Weaknesses and open questions:

- Assumption 1: What does $P_{X^{out} Y^{in}}$ mean exactly? This is central for the paper and as I read it it is not explained.

- It is not clear why causal graphs are necessary to understand the independence of certain variables.

- The claim that "researchers assume that the pre-trained model f can simulate the ID conditional distribution" should be substantiated. The methods proposed by this paper are motivated by achieving this simulation, so it is important to exhibit why it is necessary or desired.

- The presented theorems are trivial.

- Eq. (7) should be justified, as this is not the only possibility for aligning the minimizer of the score function with the class priors. For example, $max_i softmax_i f(x) / P_{Y^{in}}(i)$ would have the same minimizer but different behaviour away from the minimum.

- For the reweighting strategy of Eq. (12), having a minus sign in front of the base score seems extremely unintuitive. That the cosine should get a minus sign is clear, but why does it also affect $S_{Method}$?

- "the system with a more concentrated probability distribution has lower energy, while the system with a more divergent probability distribution (more similar to the uniform distribution) has higher energy (LeCun et al., 2006). Thus, the energy of ID data is smaller than OOD data." -- This is not clear for the logits of a classifier trained with cross-entropy loss. The cited slides (LeCun et al., 2006) mainly describe models where the energy is regularized. How can those be connected with inference-time OOD detection?

- The sign of Eq. (14) is wrong, since the *negative* energy is the base OOD detection score $S_{Method}$. This means that $-S_{Method}(f,x) = -T·\log \sum_k e^{f_i(x)/T}$. What does this mean that the AUROCs in Table 1 for RW+Energy?

- In Figure 4, I would suggest showing only one of the similar metrics AUROC and FPR, and using the space to show all datasets.

- Results for the balanced standard datasets should definitely be shown. Both for assessing the baselines compared to other publications and for checking how the proposed method (should only affect RW) impacts performance.

- I think this paper could benefit from an analysis how the base methods which do not account for class imbalance fail. E.g. is it because ID samples from rare classes are being rejected too often, or because OOD samples receive imbalanced predictions that are close to the class priors rather than to the uniform distribution? The "Analysis of Detection Results on Different ID Classes" answers this partially, but it focuses on the improvements rather than on nature of the original problems.

- The following previous works investigate class-dependent thresholding and should be discussed and potentially compared to.
  - [Guarrera et al. 2022: Class-wise Thresholding for Robust Out-of-Distribution Detection]
  - [Wang et al. 2022: Partial and Asymmetric Contrastive Learning for Out-of-Distribution Detection in Long-Tailed Recognition]

- It is not clear to me whether the compared and improved methods are state of the art in the regarded setting as the paper claims. Many publications have worked with Vision Transformers pretrained on larger datasets -- which can be considered a different setting for OOD detection in particular as pointed out by [Hendrycks et al. 2022: Scaling Out-of-Distribution Detection for Real-World Settings] -- and achieve much better performance. This as well as other related works that work only with ID data should be discussed and compared to in order to provide context. It could also help emphasize that all methods that ignore potentially imbalanced classes need to be reconsidered.

**Summary Of The Paper:**

The paper modifies OOD detection methods which assume a balanced class distribution such that they are applicable to distributions with unbalanced classes.

**Summary Of The Review:**

The paper contains interesting approaches to OOD detection with a class-imbalanced in-distribution. However, as detailed above, particularly its methodology currently contains several problematic issues.

---

> ### Author Response · Authors · 2022-11-15
> **Response to your reviews (part4)**
>
>
> >Q12: The following previous works investigate class-dependent thresholding and should be discussed and potentially compared to.
>
> **A12**: Thanks for your suggestion!
> [Guarrera et al. 2022: Class-wise Thresholding for Robust Out-of-Distribution Detection] The paper designs an optimization for threshold selection rather than designing an OOD score. Class-dependent thresholding is designed for $p_{train} \neq p_{test}$ and not for data imbalance issues. This approach only affects the precision and recall metrics at the deployment stage of the OOD detector, not the AUROC and FPR95 metrics that this paper focuses on. This can be seen in Table 1 in [Guarrera et al. 2022: Class-wise Thresholding for Robust Out-of-Distribution Detection].
>
> [Wang et al. 2022: Partial and Asymmetric Contrastive Learning for Out-of-Distribution Detection in Long-Tailed Recognition] This paper is training-time OOD detection and uses OOD data to train the model. After being finetuned, the model can deal with the imbalanced issue and OOD problem. The setting is completely different with our paper (inference-time OOD detection). In our paper, we do not change any parameters of the model and design methods to deal with the imbalanced issue on OOD detection. Note that our work and the mentioned work are not comparable due to the different problem settings.
>
> We will cite and add more discussions about these two papers in our related work.
>
>
> >Q13: It is not clear to me whether the compared and improved methods are state of the art in the regarded setting as the paper claims. Many publications have worked with Vision Transformers pretrained on larger datasets -- which can be considered a different setting for OOD detection in particular as pointed out by [Hendrycks et al. 2022: Scaling Out-of-Distribution Detection for Real-World Settings] -- and achieve much better performance. This as well as other related works that work only with ID data should be discussed and compared to in order to provide context. It could also help emphasize that all methods that ignore potentially imbalanced classes need to be reconsidered.
>
> **A13**: Thanks for your suggestion! In our paper, we use Dice [R8] as the baseline. [R8] is published in CVPR 2022 (this June) and can be regarded as the state-of-the-art published method. However, the large-scale OOD benchmark ImageNet-21k-P (cotaining 14 million images) proposed by [Hendrycks et al. 2022: Scaling Out-of-Distribution Detection for Real-World Settings] has not been released yet. It is hard for us to evaluate our methods on a dataset which is larger than ImageNet-1k in the short term. We will update the results on ImageNet-21k-P benchmark when it is publicly available. Meanwhile, we will try ViT and update the corresponding results soon.
>
> **General Response:** We have addressed your concerns about our paper. If you have more suggestions, please discuss them with us in the openreview system. We will try our best to address your further concerns and merge your comments into our revision as well!
>
> **Reference**
>
> [R1] Mingming Gong, Kun Zhang, Tongliang Liu, Dacheng Tao, Clark Glymour, and Bernhard Schölkopf. Domain adaptation with conditional transferable components. In ICML, 2016 .
>
> [R2] Petar Stojanov, Zijian Li, Mingming Gong, Ruichu Cai, Jaime G. Carbonell, and Kun Zhang. Domain adaptation with invariant representation learning: What transformations to learn? In NeurIPS, 2021.
>
> [R3] Weitang Liu, Xiaoyun Wang, John D. Owens, and Yixuan Li. Energy-based out-of-distribution detection. In NeurIPS, 2020.
>
> [R4] Dan Hendrycks and Kevin Gimpel. A baseline for detecting misclassified and out-of-distribution examples in neural networks. In ICLR, 2017.
>
> [R5] Shiyu Liang, Yixuan Li, and R. Srikant. Enhancing the reliability of out-of-distribution image detection in neural networks. In ICLR, 2018.
>
> [R6] Rui Huang, Andrew Geng, and Yixuan Li. On the importance of gradients for detecting distributional shifts in the wild. In NeurIPS, 2021.
>
> [R7] Yann LeCun, Sumit Chopra, Raia Hadsell, M Ranzato, and F Huang. A tutorial on energy-based learning. Predicting structured data, 1(0), 2006
>
> [R8] Sun and Yixuan Li. Dice: Leveraging sparsification for out-of-distribution detection. In ECCV,2022.

---

> > ### Author Response · Authors · 2022-11-19
> > **Update for ViT experiment**
> >
> > We follow the paper [R1] to reproduce MSP and Maxlogit with ViT on our imbalanced dataset ImageNet-LT-a8. The results are shown in below table. The last two rows are the results after applying our strateges. Compared with the first two rows, our strateges significantly improve the performance of the previous method.
> >
> > | Model |    Method   | iNaturalist |        |   SUN  |        | Places |        | Textures |        | Average |        |
> > |:-----:|:-----------:|:-----------:|:------:|:------:|:------:|:------:|:------:|:--------:|:------:|:-------:|:------:|
> > |       |             |    AUROC    |  FPR95 |  AUROC |  FPR95 |  AUROC |  FPR95 |   AUROC  |  FPR95 |  AUROC  |  FPR95 |
> > | ViT-B |     MSP     |    79.03%   | 87.90% | 75.86% | 86.60% | 73.03% | 87.30% |  77.70%  | 81.70% |  76.41% | 85.88% |
> > | ViT-B |   Maxlogit  |    60.47%   | 99.10% | 69.46% | 99.70% | 67.09% | 98.70% |  64.03%  | 91.00% |  65.27% | 97.13% |
> > | ViT-B |    RP+MSP   |    79.23%   | 87.10% | 75.88% | 86.60% | 73.08% | 87.10% |  77.78%  | 81.70% |  76.49% | 85.63% |
> > | ViT-B | RW+Maxlogit |    79.03%   | 87.90% | 75.86% | 86.60% | 73.03% | 87.30% |  77.70%  | 81.70% |  76.41% | 85.88% |
> >
> > [R1] Dan Hendrycks, Steven Basart, Mantas Mazeika, Andy Zou, Joe Kwon, Mohammadreza Mostajabi, Jacob Steinhardt, and Dawn Song. Scaling out-of-distribution detection for real-world settings. ICML, 2022

---

> > ### Comment · Reviewer_G6VC · 2022-11-25
> > **Responses 12 - 13 and reference list**
> >
> > **Re A12:** Thanks for the discussions. I agree that the settings considered in those two works seem to be different. Including these discussions of the differences with the related work will certainly be helpful for the reader.
> >
> > **Re A13:** While [2] is also concerned with new datasets (which would indeed be interesting to see results on in the future), I was referring to their page 3 passage following "Recent work has suggested that stronger representations from Vision Transformers pre-trained on ImageNet-21K ...". A general discussion of the effects of pre-training would in my opinion be a nice addition to the current paper, also for future reference.
> >
> > [1] Kolmogorov 1933: Foundations of the Theory of Probability, p.12.
> > [2] Hendrycks et al. 2022: Scaling Out-of-Distribution Detection for Real-World Settings

---

> > > ### Author Response · Authors · 2022-11-28
> > > **Response to your comments**
> > >
> > > >Re A13: While [2] is also concerned with new datasets (which would indeed be interesting to see results on in the future), I was referring to their page 3 passage following "Recent work has suggested that stronger representations from Vision Transformers pre-trained on ImageNet-21K ...". A general discussion of the effects of pre-training would in my opinion be a nice addition to the current paper, also for future reference.
> > >
> > > **Answer**：Thank you for your comments! Based on current results in Table 1 and updated ViT experiments, comparing the performance of MSP and Maxlogit on different models, the ViT model pretrained on the ImageNet-21k dataset does have better representation ability than the randomly initialized ResNet101, and have better performance on OOD detection.
> > >
> > > Concretely, models that are pre-trained on large-scale data can lead to better performance on ID tasks (i.e., image classification). Taking MSP and Maxlogit as examples, they judge OOD data based on the class-wise prediction of the model. If the model has higher classification accuracy, i.e., the prediction of ID samples is closer to the one-hot distribution, then it would also have better performance of OOD.
> > >
> > > On the other hand, in our study, we find that the main reason for the poor performance of OOD detection on imbalanced data is that ID-tailed data have lower classification confidence and are easily confused with OOD data, thus reducing detection performance. The pre-trained model has stronger representation ability, which can enhance the classification accuracy of the network for the tailed data, thus improving the performance of OOD detection.
> > >
> > > **Reference**
> > >
> > > [R1] Weitang Liu, Xiaoyun Wang, John D. Owens, and Yixuan Li. Energy-based out-of-distribution detection. In NeurIPS, 2020.
> > >
> > > [R2] Rui Huang and Yixuan Li. MOS: towards scaling out-of-distribution detection for large semantic space. In CVPR, 2021.
> > >
> > > [R3] Rui Huang, Andrew Geng, and Yixuan Li. On the importance of gradients for detecting distributional shifts in the wild. In NeurIPS, 2021.

---

> ### Author Response · Authors · 2022-11-15
> **Response to your reviews (part3)**
>
> >Q8: The sign of Eq. (14) is wrong, since the negative energy is the base OOD detection score $S_{Method}$. This means that $-S_{M e t h o d}(f, x)=-T \cdot \log \sum_k e^{f_i(x) / T}$. What does this mean that the AUROCs in Table 1 for RW+Energy?
>
> **A8**: According to [R3], the original Energy score is $S_{Energy}(f, x)=-T \cdot \log \sum_k e^{f_i(x) / T}$. Thus, we have $S_{RW+Energy}(f, x)=(-T \cdot \log \sum_k e^{f_i(x) / T}) \cdot (-\cos(softmax(f(x)),P_{Y^{\rm in}}))$, which is equal to $T \cdot \log \sum_k e^{f_i(x) / T}\cdot\cos(softmax(f(x)),P_{Y^{\rm in}})$. The sign of Eq. (14) is correct since there are two minus symbols, i.e., $(-1) \cdot (-1)=1$.
>
> >Q9: In Figure 4, I would suggest showing only one of the similar metrics AUROC and FPR, and using the space to show all datasets.
>
> **A9**: Thanks for your suggestion! The complete experimental results of Figure 4 are in Appendix A.1.3. We will revise Figure 4 according to your suggestion. Your suggestion will make our paper more clear and increase readability.
>
> >Q10: Results for the balanced standard datasets should definitely be shown. Both for assessing the baselines compared to other publications and for checking how the proposed method (should only affect RW) impacts performance.
>
> **A10**: As for RP strategy, when the training dataset is balanced (the class-prior distribution is uniform distribution in Eq. (10) and Eq. (11)), RP+Method will be same as the original method. Specifically.
>
> For RW strategy, we conduct experiments on full ImageNet dataset (note that it is balanced) and the results are shown below. The performances of Energy and ODIN are quite close to the performnce of Energy and ODIN using RW strategy on the balanced dataset.
>
> | Method | RW Strategy | iNaturalist |        |   SUN  |        | Places |        | Textures |        | Average |        |
> |:------:|:-----------:|:-----------:|:------:|:------:|:------:|:------:|:------:|:--------:|:------:|:-------:|:------:|
> |        |             |    AUROC    |  FPR95 |  AUROC |  FPR95 |  AUROC |  FPR95 |   AUROC  |  FPR95 |  AUROC  |  FPR95 |
> |  ODIN  |     w/o     |      89.81% | 45.74% | 83.01% | 63.84% | 82.50% | 66.48% |   81.31% | 65.51% |  84.16% | 60.39% |
> |        |      w/     |      89.87% | 45.37% | 83.05% | 63.59% | 82.54% | 66.32% |   81.36% | 65.27% |  84.20% | 60.14% |
> | Energy |     w/o     |      93.28% | 37.62% | 88.60% | 49.48% | 87.36% | 54.61% |   86.80% | 53.40% |  89.01% | 48.78% |
> |        |      w/     |      93.48% | 35.86% | 88.48% | 50.13% | 87.29% | 54.80% |   86.72% | 53.24% |  88.99% | 48.51% |
>
> >Q11: I think this paper could benefit from an analysis how the base methods which do not account for class imbalance fail. E.g. is it because ID samples from rare classes are being rejected too often, or because OOD samples receive imbalanced predictions that are close to the class priors rather than to the uniform distribution? The "Analysis of Detection Results on Different ID Classes" answers this partially, but it focuses on the improvements rather than on nature of the original problems.
>
> **A11**: Thanks for your suggestion! The main reason why the base methods do not account for class imbalance fail is that previous methods explicitly/implicitly use the uniform distribution assumption. For example, GradNorm takes gradients of KL divergence between model prediction and uniform distribution as the OOD score function. GradNorm assumes that OOD data are closer to uniform distribution than ID data.
>
> Under imbalanced scenarios, the model-predicted distribution of tailed samples may be similar to uniform distribution due to their low confidence in their ground-truth class. Moreover, OOD samples are not similar to uniform distribution as they would have higher confidence in major classes. As a result, under imbalanced scenarios, OOD detectors may make mistakes like rejecting tailed ID samples or accepting some OOD samples. In order to solve the problem, we propose our RW/RP strategies, which is important for OOD detectors with imbalanced issues.

---

> > ### Comment · Reviewer_G6VC · 2022-11-25
> > **Responses 8-11**
> >
> > **Re A8:** As stated above, [R3] uses the **negative energy** as their score.
> > The number $-T·\log \sum_k e^{f_i(x)/T}$ is the **positive energy.**
> > See Eq. (7) in [R3], where they also state: "Here we use negative energy scores, −E(x; f), to align with the conventional definition where positive (in-distribution) samples have
> > higher scores."
> >
> > In the submission, the same conventional definition is used, stated in Eq. (6).
> >
> > Thus I think that the crucial point on the minus sign still holds.
> >
> > **Re A10:** Thanks for the numbers. It is helpful information that there are no disadvantages for using the methods also in the balanced regime are observed in the standard settings.
> >
> > **Re A11:** Thanks for formulating the likely reasons for the shortcomings of standard methods. What I meant in the original comment is that it could be nice to show a quantitative analysis for which of them plays the more important role, and maybe an example of ID or OOD inputs which demonstrate the failure modes. Not mentioned in the answer, the updates in A.1.3 are alredy quite interesting in that regard.

---

> ### Author Response · Authors · 2022-11-15
> **Response to your reviews (part2)**
>
>
> >Q5: Eq. (7) should be justified, as this is not the only possibility for aligning the minimizer of the score function with the class priors. For example, $\max_{i}( softmax_{i}(\mathbf{f}(\mathbf{x})) /\mathbb{P}_{Y^{\rm in}}(i))$ would have the same minimizer but different behaviour away from the minimum.
>
> **A5**: Thanks for your comments! First of all, we want to emphasize  this paper's main contribution: our paper is first work to use **ID class prior distribution** to address data imbalance issue on OOD detection. We propose two effective strategies and achieve significant improvements. In our paper,  $\max_{i}(softmax_i(\mathbf{f}(\mathbf{x})) - \mathbb{P}_{Y^{\rm in}}(i))$ is a reasonable and effective selection.
>
> Your suggestion is very interesting and important, and also quite inspiring. There may be various selections for Eq. (7). According to your example, we conduct experiments for  $\max_{i}(softmax_i(f(x)) / P_{Y^{\rm in}}(i))$ and show the corresponding results in the below table. Our experiments show that $\max_{i}( softmax_i(\mathbf{f}(\mathbf{x})) / P_{Y^{\rm in}}(i))$ also performs very well. Based on your constructive suggestion, we will add more discussions in our revised version.
>
>
> | Method | iNaturalist |        |   SUN  |        | Places |        | Textures |        | Average |        |
> |:------:|:-----------:|:------:|:------:|:------:|:------:|:------:|:--------:|:------:|:-------:|:------:|
> |        |    AUROC    |  FPR95 |  AUROC |  FPR95 |  AUROC |  FPR95 |   AUROC  |  FPR95 |  AUROC  |  FPR95 |
> |    $\max_{i}( softmax_i(\mathbf{f}(\mathbf{x})) / \mathbb{P}_{Y^{\rm in}}(i))$   |    81.76%   | 69.75% | 57.80% | 94.80% | 54.90% | 94.68% |  52.01%  | 88.97% |  61.62% | 87.05% |
>
>
> >Q6: For the reweighting strategy of Eq. (12), having a minus sign in front of the base score seems extremely unintuitive. That the cosine should get a minus sign is clear, but why does it also affect $S_{Method}$?
>
> **A6**: The minus belongs to the cosine function and it is the cosine function that affects $S_{Method}$.
> $S_\text{RW+Method}(f,x)=S_\text{Method}(f,x) \cdot (-\cos(softmax(f(x)),P_{Y^{\rm in}}))$
> In detail,  ID data are far from the ID class prior distribution, and their cosine similarity is smaller than OOD data. The directions of $S_{Method}$ and cosine function are consistent.
>
> >Q7: "the system with a more concentrated probability distribution has lower energy, while the system with a more divergent probability distribution (more similar to the uniform distribution) has higher energy (LeCun et al., 2006). Thus, the energy of ID data is smaller than OOD data." -- This is not clear for the logits of a classifier trained with cross-entropy loss. The cited slides (LeCun et al., 2006) mainly describe models where the energy is regularized. How can those be connected with inference-time OOD detection?
>
> **A7**: Energy [R3] is the classic and representive OOD detection method, which is the first work using the energy function [R7] to address the OOD detection issue. We just follow this work [R3] to conduct our experiments. In the paper [R3], the authors claimed that "The learning process shapes the energy surface to assign low energy values to the ID data and higher energy values to OOD data." and their experiments also support the claim. Based on this observation, the authors in [R3] designed the energy-based method by using the value of energy to distinguish ID and OOD test data. That is the basic idea of [R3] to connect energy and inference-time OOD.

---

> > ### Comment · Reviewer_G6VC · 2022-11-25
> > **Responses 5-7**
> >
> > **Re A5:** Thanks for the interesting experimental results. It would be interesting to read a discussion which approach to align the minimum with the class prior is 'the right approach', but I agree that as a first step, studying any approach that does it makes sense.
> >
> > **Re A6:** Writing the minus sign in front of the second factor does not change the value. If we assume a fixed $softmax(f(x))$, then the second factor $-cos(softmax(f(x)), P_{Y^in})$ is a fixed negative number (both vectors are in the positive orthant), and the resulting score is lower for higher $S_{Method}(f,x)$, which does not make sense.
> > Using a product of two positive scores where one is positively correlated with the input being ID and the other one negatively is not theoretically sound. This issue is not fixed by multiplying the product by $-1$.
> >
> > **Re A7:** The stated claim in [R3] reads in full:
> > "At training time, we propose an energy-bounded learning objective to fine-tune the network. The learning process shapes the energy surface to assign low energy values to the indistribution data and higher energy values to OOD training data. Specifically, we regularize the energy using two square hinge loss terms, which explicitly create the energy gap between in- and out-of-distribution training data."
> > I.e. this is about models that are fine-tuned using OOD data (from a separate training distribution of natural images) during training.
> >
> > This is not the setting considered in this submission, which does inference-time OOD detection.
> > For models that did not have access to OOD data, [R3] observes only a weak (and for some datasets detrimental) effect of using energy compared to softmax.

---

> > > ### Author Response · Authors · 2022-11-28
> > > **Response to your comments!**
> > >
> > > >Re A6: Writing the minus sign in front of the second factor does not change the value. If we assume a fixed , then the second factor  is a fixed negative number (both vectors are in the positive orthant), and the resulting score is lower for higher , which does not make sense. Using a product of two positive scores where one is positively correlated with the input being ID and the other one negatively is not theoretically sound. This issue is not fixed by multiplying the product by .
> > >
> > >
> > > >Re A8: As stated above, [R3] uses the negative energy as their score. The number  is the positive energy. See Eq. (7) in [R3], where they also state: "Here we use negative energy scores, −E(x; f), to align with the conventional definition where positive (in-distribution) samples have higher scores."
> > >
> > > >In the submission, the same conventional definition is used, stated in Eq. (6).
> > >
> > > >Thus I think that the crucial point on the minus sign still holds.
> > >
> > > **Answer**: There might be misunderstanding about the cosine similarity, so we would like to help you understand with an example calculation.
> > >
> > > Assume that there are five classes in ID data, the sample number of each class in training set is $[7, 3,20,36,34]$. So the ID class prior distribution $P_{Y_{in}} = [0.07,0.03,0.2,0.36,0.34]$. As shown in GradNorm [R3], the prediction of ID data tends to concentrate on the ground-truth class, which is more similar to one-hot distribution. Given an ID sample in class #3, the prediction of the classifier may be $[0.01,0.01,0.95,0.01,0.02]$. In the meanwhile, according to Theorem 1 in our paper, the prediction of OOD data tends to be spread across multiple classes, which we think is closer to the ID class prior distribution $P_{Y_{in}}$. So the prediction of one OOD sample may be $[0.08, 0.02, 0.2, 0.36, 0.34]$. Thus we have $cos(ID, P_{Y_{in}}) = 0.3928$ and $cos(OOD, P_{Y_{in}}) = 0.9997$. It is obvious that the cosine similarity of OOD data is larger than that of ID data, so we need to add a negative sign to make the weight calculated from ID data larger than that of OOD data. Then we have $-cos(ID, P_{Y_{in}}) = -0.3928$ and $-cos(OOD, P_{Y_{in}}) = -0.9997$.
> > >
> > > This is consistent with the direction of Energy score, i.e., the OOD score of ID data is larger than that of OOD data. We actually use a product of two scores where they are both positively correlated with the input being ID.
> > >
> > > In principal, the cosine similarity in our paper is $s(u,v)=cos(\theta_{u,v}) = \frac{u^Tv}{\left \| u \right \|\left \|v \right \|},s(u,v) \in [0,1]$.
> > > When the cosine value $cos ( \theta_{u,v})$ is close to 1 (i.e. $\theta_{u,v} \rightarrow 0^{\circ}$), it indicates that the two vectors are more similar. When the cosine value $cos ( \theta_{u,v})$ is close to 0 (i.e. $\theta_{u,v}$ $\rightarrow$ $90^{\circ}$), it indicates that the two vectors are more different.
> > >
> > > >Re A7: The stated claim in [R3] reads in full: "At training time, we propose an energy-bounded learning objective to fine-tune the network. The learning process shapes the energy surface to assign low energy values to the indistribution data and higher energy values to OOD training data. Specifically, we regularize the energy using two square hinge loss terms, which explicitly create the energy gap between in- and out-of-distribution training data." I.e. this is about models that are fine-tuned using OOD data (from a separate training distribution of natural images) during training.
> > >
> > > >This is not the setting considered in this submission, which does inference-time OOD detection. For models that did not have access to OOD data, [R3] observes only a weak (and for some datasets detrimental) effect of using energy compared to softmax.
> > >
> > >
> > > **Answer**：**We think there is a misunderstanding in Energy [R1].** Energy [R1] proposes two designs: one without finetuning and one with finetuning, as shown in the contributions on page 2 and Table 1 on page 7 in Energy [R1]. So Energy [R1] contains two kinds of settings: training-time OOD detection and inference-time OOD detection.
> > > We follow the benchmark and baselines proposed by MOS [R2], where Energy is also regarded as a kind of inference-time OOD detection, i.e., directly using the energy function as the OOD score function. This is clearly seen in the source code of MOS: https://github.com/deeplearning-wisc/large_scale_ood/blob/master/test_baselines.py.

---

> ### Author Response · Authors · 2022-11-15
> **Response to your reviews (part1)**
>
> Thanks for your comments! We will answer them as follows:
>
> >Q1: Assumption 1: What does $P_{X_{out}Y_{in}}$  mean exactly? This is central for the paper and as I read it it is not explained.
>
> **A1**: $P_{X_{out}Y_{in}}$ is the joint distribution of $X_{out}$ and $Y_{in}$. Maybe $\mathbb{P}({X^{\rm out},{Y}^{\rm in}})$ is better. We will follow your suggestion to revise it.
>
> >Q2: It is not clear why causal graphs are necessary to understand the independence of certain variables.
>
> **A2**: Causal graphs are used to support Assumption 1 that can hold in most cases. According to [R1][R2], we can use three causal graphs to describe most situations in OOD detection. Figure 1$($a$)$ corresponds to images without styles,  Figure 1$($b$)$ corresponds to images with shared styles and Figure 1$($c$)$ corresponds to images with different styles. Through these graphs, it is clear that $Y^{\rm in}$ and $X^{\rm out}$ are independent. Thus, our assumption 1 is a general assumption that can hold in common situations.
>
> >Q3: The claim that "researchers assume that the pre-trained model f can simulate the ID conditional distribution" should be substantiated. The methods proposed by this paper are motivated by achieving this simulation, so it is important to exhibit why it is necessary or desired.
>
> **A3**: This is a consensus in OOD detection and has been potentially used in previous works [R3-R6].
>
> >Q4: The presented theorems are trivial.
>
> **A4**: Whether a theorem is trivial or not trival depends on whether the theorem can inspire us to address the pracitcal issues or not. If a theorem can help us to address the practical issues, the theorem is not trivial, even if the techniques used to prove the theorem are trival. For example, the proof of Bayes' Theorem is simple, but few people believe that Bayes' Theorem is trival. We want to argue that the simple proof does not imply the trivial theoretical result. Due to the mild and practical assumption in our theorem, our theorem is very general and practical, and can be used to guide the development of OOD detection algorithms.
>
> Then, we want to restate the meaning and impact of our theorem as follows.
>
> **Meaning of our Theorem.** There exists an open issue in OOD detection, that is how the **multi-class** ID data affects the detection of OOD data. Our theorem is the first theorem to answer the issue under mild and practical assumption: **multi-class** ID data is deeply related to the detection of OOD data and the ID class priors play an important role to affecet the detection of OOD data. Additionally, our algorithms are rooted in this theorem and achieve good performance, which indicates that our theorem can be used to guide the development of OOD detection.
>
> **Impact of our Theorem.** The two implementations RP and RW based on Theorem 1 have also been proven to have significant effects in experiments. As a highlight, when evaluating the OOD detection performance on iNaturalist dataset, our method can achieve ∼36% increase on AUROC and ∼61% decrease on FPR95, compared with the original Energy. The empirical results support our theorem and that our theorem can be used to guide the development of OOD detection.

---

> > ### Comment · Reviewer_G6VC · 2022-11-25
> > **Responses 1-4**
> >
> > Thanks for the detailed responses. I will address all of them below. I think that the updated passages and experimental results all improve the paper.
> >
> > I still have major concerns about the theoretical contributions. Particularly points 6 and 8 are important errors in my current understanding, but I would be happy to be proven wrong here if I am mistaken about these points.
> >
> > **Re A1-A3:** I am still not convinced that the theoretical considerations based on the causal graphs and on the independence of $Y^{in}$ and $X^{out}$ meaningfully lead to exactly the conclusions that are made.
> >
> > I also agree with the similar point made by Reviewer vuMZ.
> >
> > As I read it, the main conclusions do not arise from the theoretical derivations (in which I do not see any surprising/novel contribution), but from the assumptions supposedly made by previous works (see Q3) and by the arbitrarily chosen re-balancing method (see Q5).
> >
> > **Re A1:** The meaning of the joint distribution of $Y^{in}$ and $X^{out}$ is still not clear to me. Can it be interpreted as an experiment where they are drawn in a specific setting?
> >
> > **Re A3:** I cannot find the assumption stated in any of the works cited as potentially using it. It might be that in an ideal case, the methods would work well if $softmax(f(x)) = [P_{Y^{in}}(1), \ldots, P_{Y^{in}}(K)]$, but similarly one could say that they hope for example that $max_c softmax(f(x))[c]$ is small for $x$ being OOD. There seems not to be a clear distinction made by the previous works that allows for the conclusion that the notion of conditional probability (and not just a number that coincides but could be used for different reasons) used in the paper is regarded by those works, or even seen as a consensus.
> >
> > **Re A4:** Theorem 1 is trivial because its statement is a very well known basic fact of probability theory [1]: "If two random variables A and B are independent, then $P(A|B) = P(A)$.", with other variable names for $A$ and $B$.
> >
> > *The issue is that it is a known basic fact, not that the proof is simple.*
> >
> > If the theorem could be restated in a way that it does include reasoning about ID and OOD distributions, it might make more sense as a theorem and a novel contribution.
> > Particularly, the fact stated in the theorem is only relevant if one is interested in $P_{Y^{in}|X^{out}}(y|x)$, which is only motivated later.
> > Likewise, Theorem 2 also is a well known fact.

---

> > > ### Author Response · Authors · 2022-11-28
> > > **Response to your comments!**
> > >
> > > >Re A1: The meaning of the joint distribution of  and  is still not clear to me. Can it be interpreted as an experiment where they are drawn in a specific setting?
> > >
> > > **Answer:** Given two random variables that are defined on the same probability space, the joint probability distribution is the corresponding probability distribution on all possible pairs of outputs. For any object $x$ and label $y$,
> > > $P(X^{\rm out}=x,{Y}^{\rm in}=y) = P(X^{\rm out}=x \ and \ {Y}^{\rm in}=y)=P({Y}^{\rm in}=y|X^{\rm out}=x) \cdot P(X^{\rm out}=x)$. The $P(X^{\rm out}=x \ and \ {Y}^{\rm in}=y)$ represents the probability that the event $X^{\rm out}=x \ and \ {Y}^{\rm in}=y$ happens.
> > >
> > > The joint distribution of $X^{\rm out}$ and ${Y}^{\rm in}$ consists of two parts: the distribution of OOD data and the probability distribution that one OOD sample belongs to each ID class. If $X^{\rm out}$ and ${Y}^{\rm in}$ are independent, we have $P(X^{\rm out},{Y}^{\rm in}) = P_{X^{\rm out}}P_{Y^{\rm in}}$.
> > >
> > > >Re A3: I cannot find the assumption stated in any of the works cited as potentially using it. It might be that in an ideal case, the methods would work well if , but similarly one could say that they hope for example that  is small for  being OOD. There seems not to be a clear distinction made by the previous works that allows for the conclusion that the notion of conditional probability (and not just a number that coincides but could be used for different reasons) used in the paper is regarded by those works, or even seen as a consensus.
> > >
> > > **Answer:**
> > > Thanks for your comments! We agree that if we directly claim "researchers assume that the pre-trained model f can simulate the ID conditional distribution", it may mislead the readers, so we will add more justifications about how this assumption is used in previous methods in the latest version of our paper, which can avoid this being regarded as a consensus.
> > >
> > > For example, GradNorm uses KL distance to measure the similarity of model output prediction and uniform distribution to complete OOD detection. Since GradNorm's training data is balanced, the uniform distribution is the class-prior distribution of the training set. GradNorm thinks the OOD data are more similar to the uniform distribution, which is potentially using the fact that "the pre-trained model can simulate the ID conditional distribution."
> > >
> > > >Re A4: Theorem 1 is trivial because its statement is a very well known basic fact of probability theory [1]: "If two random variables A and B are independent, then .", with other variable names for  and .
> > >
> > >
> > > >The issue is that it is a known basic fact, not that the proof is simple.
> > >
> > > >If the theorem could be restated in a way that it does include reasoning about ID and OOD distributions, it might make more sense as a theorem and a novel contribution. Particularly, the fact stated in the theorem is only relevant if one is interested in , which is only motivated later. Likewise, Theorem 2 also is a well known fact.
> > >
> > >
> > > **Answer:** Theorem 1 is a property of the conditional probability distribution itself, but we are concerned with whether $X_{out}$ and $Y_{in}$ are independent under the general hypothesis. Therefore, we use causal graphs to describe most situations in OOD detection and discuss the independence of $X_{out}$ and $Y_{in}$ one by one. Figure 1 illustrates that under most conditions, Assumption 1 holds, i.e., $X_{out}$ and $Y_{in}$ are independent. So $X_{out}$ and $Y_{in}$ are independent under the general hypothesis, and thus Theorem 1 also holds in general cases.

---

> ### Author Response · Authors · 2022-11-17
> **Looking forward to your responses or further suggestions/comments!**
>
> Dear Reviewer G6VC,
>
> We have addressed your initial concerns regarding our paper. We are happy to discuss them with you in the openreview system if you feel that there still are some concerns/questions. We also welcome new suggestions/comments from you!
>
> Best regards,
>
> Authors of #3905

---

> ### Author Response · Authors · 2022-11-18
> **Looking forward to your responses or further suggestions/comments!**
>
> Dear Reviewer G6VC,
>
> We have addressed your initial concerns regarding our paper. We are happy to discuss them with you in the openreview system if you feel that there still are some concerns/questions. We also welcome new suggestions/comments from you!
>
> Best regards,
>
> Authors of #3905

---

> ### Author Response · Authors · 2022-11-19
> **Looking forward to your responses or further suggestions/comments!**
>
> Dear Reviewer G6VC,
>
> We have addressed your initial concerns regarding our paper. We are happy to discuss them with you in the openreview system if you feel that there still are some concerns/questions. We also welcome new suggestions/comments from you!
>
> Best regards,
>
> Authors of #3905

---

> ### Author Response · Authors · 2022-11-20
> **Looking forward to your responses or further suggestions/comments!**
>
> Dear Reviewer G6VC,
>
> We have addressed your initial concerns regarding our paper. We are happy to discuss them with you in the openreview system if you feel that there still are some concerns/questions. We also welcome new suggestions/comments from you!
>
> Best regards,
>
> Authors of #3905

---

> ### Author Response · Authors · 2022-11-21
> **Looking forward to your responses or further suggestions/comments!**
>
> Dear Reviewer G6VC,
>
> We have addressed your initial concerns regarding our paper. We are happy to discuss them with you in the openreview system if you feel that there still are some concerns/questions. We also welcome new suggestions/comments from you!
>
> Best regards,
>
> Authors of #3905

---

> ### Author Response · Authors · 2022-11-22
> **Looking forward to your responses or further suggestions/comments!**
>
> Dear Reviewer G6VC,
>
> We have addressed your initial concerns regarding our paper. We are happy to discuss them with you in the openreview system if you feel that there still are some concerns/questions. We also welcome new suggestions/comments from you!
>
> Best regards,
>
> Authors of #3905

---

> ### Author Response · Authors · 2022-11-23
> **Looking forward to your responses or further suggestions/comments!**
>
> Dear Reviewer G6VC,
>
> We have addressed your initial concerns regarding our paper. We are happy to discuss them with you in the openreview system if you feel that there still are some concerns/questions. We also welcome new suggestions/comments from you!
>
> Best regards,
>
> Authors of #3905

---

> ### Author Response · Authors · 2022-11-24
> **Looking forward to your responses or further suggestions/comments!**
>
> Dear Reviewer G6VC,
>
> We have addressed your initial concerns regarding our paper. We are happy to discuss them with you in the openreview system if you feel that there still are some concerns/questions. We also welcome new suggestions/comments from you!
>
> Best regards,
>
> Authors of #3905

---

> ### Author Response · Authors · 2022-11-25
> **Looking forward to your responses or further suggestions/comments!**
>
> Dear Reviewer G6VC,
>
> We have addressed your initial concerns regarding our paper. We are happy to discuss them with you in the openreview system if you feel that there still are some concerns/questions. We also welcome new suggestions/comments from you!
>
> Best regards,
>
> Authors of #3905

---

### Official Review · Reviewer_uGHp · 2022-10-25

**Confidence:** 3
**Correctness:** 4
**Technical Novelty And Significance:** 3
**Empirical Novelty And Significance:** 3
**Recommendation:** 6

**Clarity, Quality, Novelty And Reproducibility:**

I think the paper is well written and all experiments are performed well. The novelty seems to be good - even though its such a simple fix, the approach is general purpose and improves performance across the board on imbalanced datasets. It also seems reproducible. The authors have provided most details.

**Strength And Weaknesses:**

The method is really simple, yet elegant. The authors clearly describe the motivation and present the fix they propose. It is well explained.

One of the most appealing features about the method was its ability to work with many other techniques. The authors show experiments of four such methods - MSP, ODIN, Energy and GradNorm. The method improves performance across board on all experiments.

I also liked how extensive ablations were performed.

One question I had was what happens when you use your method on balanced dataset? I think it should go back to the original formulation and there should be no loss in performance compared to baseline. Does that happen?





**Summary Of The Paper:**

This paper proposes a very simple solution to an important problem of OOD detection. The authors study the effect of class imbalance in the training set. The idea is that if there is class imbalance, the current techniques that perform inference time OOD detection will fail as they assume uniform class probably distribution. To remedy this, the authors propose a simple fix in which: (1) If the methods explicitly use uniform prior, they replace it with training class probability distribution, or (2) For the methods that do not use class probabilities, they have a simple reweighting mechanism. Strong experimental results are obtained.

**Summary Of The Review:**

I think the paper proposes a simple solution, which they have demonstrated to be quite effective. Overall, I like the paper.

---

> ### Author Response · Authors · 2022-11-15
> **Response to your reviews**
>
> Thanks for your comments! We will answer them as follows:
>
> >Q: One question I had was what happens when you use your method on balanced dataset? I think it should go back to the original formulation and there should be no loss in performance compared to baseline. Does that happen?
>
> **Answer**: As for RP strategy, when the training dataset is balanced, RP+Method will be the same as the original method. Specifically, the class-prior distribution is uniform distribution in Eq. (10) and Eq. (11).
> As for RW strategy, we conduct experiments on full ImageNet dataset and the results are shown below. The performances of Energy and ODIN are quite close whether using RW strategy on the balanced dataset.
>
> | Method | RW Strategy | iNaturalist |        |   SUN  |        | Places |        | Textures |        | Average |        |
> |:------:|:-----------:|:-----------:|:------:|:------:|:------:|:------:|:------:|:--------:|:------:|:-------:|:------:|
> |        |             |    AUROC    |  FPR95 |  AUROC |  FPR95 |  AUROC |  FPR95 |   AUROC  |  FPR95 |  AUROC  |  FPR95 |
> |  ODIN  |     w/o     |      89.81% | 45.74% | 83.01% | 63.84% | 82.50% | 66.48% |   81.31% | 65.51% |  84.16% | 60.39% |
> |        |      w/     |      89.87% | 45.37% | 83.05% | 63.59% | 82.54% | 66.32% |   81.36% | 65.27% |  84.20% | 60.14% |
> | Energy |     w/o     |      93.28% | 37.62% | 88.60% | 49.48% | 87.36% | 54.61% |   86.80% | 53.40% |  89.01% | 48.78% |
> |        |      w/     |      93.48% | 35.86% | 88.48% | 50.13% | 87.29% | 54.80% |   86.72% | 53.24% |  88.99% | 48.51% |
>
> **General Response:** We have addressed your concerns about our paper. If you have more suggestions, please discuss them with us in the openreview system. We will try our best to address your further concerns and merge your comments into our revision as well!

---

> ### Author Response · Authors · 2022-11-17
> **Looking forward to your responses or further suggestions/comments!**
>
> Dear Reviewer uGHp,
>
> We have addressed your initial concerns regarding our paper. We are happy to discuss them with you in the openreview system if you feel that there still are some concerns/questions. We also welcome new suggestions/comments from you!
>
> Best regards,
>
> Authors of #3905

---

> ### Author Response · Authors · 2022-11-18
> **Looking forward to your responses or further suggestions/comments!**
>
> Dear Reviewer uGHp,
>
> We have addressed your initial concerns regarding our paper. We are happy to discuss them with you in the openreview system if you feel that there still are some concerns/questions. We also welcome new suggestions/comments from you!
>
> Best regards,
>
> Authors of #3905

---

> ### Author Response · Authors · 2022-11-19
> **Looking forward to your responses or further suggestions/comments!**
>
> Dear Reviewer uGHp,
>
> We have addressed your initial concerns regarding our paper. We are happy to discuss them with you in the openreview system if you feel that there still are some concerns/questions. We also welcome new suggestions/comments from you!
>
> Best regards,
>
> Authors of #3905

---

> ### Author Response · Authors · 2022-11-20
> **Looking forward to your responses or further suggestions/comments!**
>
> Dear Reviewer uGHp,
>
> We have addressed your initial concerns regarding our paper. We are happy to discuss them with you in the openreview system if you feel that there still are some concerns/questions. We also welcome new suggestions/comments from you!
>
> Best regards,
>
> Authors of #3905

---

> ### Author Response · Authors · 2022-11-21
> **Looking forward to your responses or further suggestions/comments!**
>
> Dear Reviewer uGHp,
>
> We have addressed your initial concerns regarding our paper. We are happy to discuss them with you in the openreview system if you feel that there still are some concerns/questions. We also welcome new suggestions/comments from you!
>
> Best regards,
>
> Authors of #3905

---

> ### Author Response · Authors · 2022-11-22
> **Looking forward to your responses or further suggestions/comments!**
>
> Dear Reviewer uGHp,
>
> We have addressed your initial concerns regarding our paper. We are happy to discuss them with you in the openreview system if you feel that there still are some concerns/questions. We also welcome new suggestions/comments from you!
>
> Best regards,
>
> Authors of #3905

---

> ### Author Response · Authors · 2022-11-23
> **Looking forward to your responses or further suggestions/comments!**
>
> Dear Reviewer uGHp,
>
> We have addressed your initial concerns regarding our paper. We are happy to discuss them with you in the openreview system if you feel that there still are some concerns/questions. We also welcome new suggestions/comments from you!
>
> Best regards,
>
> Authors of #3905

---

> ### Author Response · Authors · 2022-11-24
> **Looking forward to your responses or further suggestions/comments!**
>
> Dear Reviewer uGHp,
>
> We have addressed your initial concerns regarding our paper. We are happy to discuss them with you in the openreview system if you feel that there still are some concerns/questions. We also welcome new suggestions/comments from you!
>
> Best regards,
>
> Authors of #3905

---

> ### Author Response · Authors · 2022-11-25
> **Looking forward to your responses or further suggestions/comments!**
>
> Dear Reviewer uGHp,
>
> We have addressed your initial concerns regarding our paper. We are happy to discuss them with you in the openreview system if you feel that there still are some concerns/questions. We also welcome new suggestions/comments from you!
>
> Best regards,
>
> Authors of #3905

---

> ### Author Response · Authors · 2022-11-26
> **Looking forward to your responses or further suggestions/comments!**
>
> Dear Reviewer uGHp,
>
> We have addressed your initial concerns regarding our paper. We are happy to discuss them with you in the openreview system if you feel that there still are some concerns/questions. We also welcome new suggestions/comments from you!
>
> Best regards,
>
> Authors of #3905

---

### Official Review · Reviewer_dyEV · 2022-10-25

**Confidence:** 4
**Correctness:** 3
**Technical Novelty And Significance:** 3
**Empirical Novelty And Significance:** 3
**Recommendation:** 6

**Clarity, Quality, Novelty And Reproducibility:**

This paper is well-written and easy to understand, the motivation of this paper is clear and reasonable, and the results reported in this paper are promising. However, there are still some unresolved questions that need to be addressed.

**Strength And Weaknesses:**

Another branch of OOD detection is feature-space detection scores, such as KNN. I think a discussion of the potential extension of their method on feature space may be good since logit-space scores may not well fit some real-world applications, even since not all applications have a good assumption.

Fig 2 shows the performance of the balanced and imbalanced dataset, which, if an OOD set is close to the tailed categories of ID data, such as the OOD set highly close to the minor categories, and what's the performance of their method? I think reporting some per-category or hard case OOD is also interesting instead of reporting the full dataset score.

Assumption 1 seems good for natural images, while for those images with a huge domain gap between natural images, not the MINIST vs Fasion-MINIST one, how does assumption 1 fit for this case?

As for model selection, they only selected the ResNet for verification, it is reasonable for the standard setting, but considering that there are so many pretrained models existing, VIT, CLIP which was pretrained on large-scale data, reporting the scores for pretrained model with supervised or zero-shot setting is also necessary for the extension of this method since most of the recent OOD tasks are suing pretrained models.

What's the performance on other scores, such as Maxlogit ?

The performance of AUROC and FPR95 are pretty low on Textures, which is interesting; even though Detection methods with the help of their method on other more natural-image-like datasets look good, textures still fail to perform OOD detection.

Those selected datasets are pretty standard, and I wonder what the performance of zero-shot performance is with pretrained models (i.e, CLIP) when we use this method.

**Summary Of The Paper:**

This paper explores the importance of ID class prior for OOD detection; two methods are proposed, replace and reweight, which can enhance the current logit-space OOD detection scores and improve the performance a lot. Both medical and theatrical analyses are conducted and provided. The experiments on benchmark datasets support their motivation.

**Summary Of The Review:**

The idea of this paper is new and looks novel; the analysis of this paper supports their motivation. They also provide a good theatrical analysis of their discovery and idea. In general, I think this is a good finding for OOD detection.

---

> ### Author Response · Authors · 2022-11-15
> **Response to your reviews (part2)**
>
> >Q4:As for model selection, they only selected the ResNet for verification; it is reasonable for the standard setting, but considering that there are so many pre-trained models existing, VIT, CLIP, which was pre-trained on large-scale data, reporting the scores for pre-trained model with supervised or zero-shot setting is also necessary for the extension of this method since most of the recent OOD tasks are using pre-trained models.
>
> **A4**: That's a good question！ We will try ViT and update the results soon！
>
> >Q5: What's the performance on other scores, such as Maxlogit ?
>
> **A5**:  We search for Maxlogit and we think it is [Scaling Out-of-Distribution Detection for Real-World Settings]. The experiment results are shown below. It is clear that our strategy works well on Maxlogit.
> |    Method   | iNaturalist |        |   SUN  |        | Places |        | Textures |        | Average |        |
> |:-----------:|:-----------:|:------:|:------:|:------:|:------:|:------:|:--------:|:------:|:-------:|:------:|
> |             |    AUROC    |  FPR95 |  AUROC |  FPR95 |  AUROC |  FPR95 |   AUROC  |  FPR95 |  AUROC  |  FPR95 |
> |   Maxlogit  |    60.14%   | 98.70% | 70.64% | 93.13% | 70.15% | 91.96% |  41.83%  | 98.30% |  60.69% | 95.52% |
> | RW+Maxlogit |    78.16%   | 77.68% | 77.26% | 81.91% | 75.15% | 83.28% |  48.34%  | 94.06% |  69.73% | 84.23% |
>
> >Q6: The performance of AUROC and FPR95 are pretty low on Textures, which is interesting; even though Detection methods with the help of their method on other more natural-image-like datasets look good, textures still fail to perform OOD detection.
>
> **A6**:  As shown in Table 1 of the paper, the performance of other baselines on Textures is also consistent. We note that Textures is the worst-performing dataset [R1] in MSP and ODIN even when the data is balanced. This phenomenon is very interesting, and we will study it in the future. In the paper, we propose a general method to make OOD detectors adapt to imbalanced scenarios, and improve the performance on Textures.
>
> >Q7: Those selected datasets are pretty standard, and I wonder what the performance of zero-shot performance is with pretrained models (i.e, CLIP) when we use this method.
>
> **A7**: 1) In our setting, we need the training dataset to finetune the model to obtain better performance in the ID task. Then we collect the ID class prior distribution from the imbalanced training dataset. The imbalanced issue is rooted in the class-imbalance of training dataset. 2) In the zero-shot setting, CLIP is a pre-trained model designed for zero-shot learning and does not need the training set of ImageNet. The training dataset of ImagNet will not be used to finetune the model and it is impossible for us to obtain the ID-class-prior distribution. Moreover, there is no imbalanced issue in CLIP, since CLIP does not use the training dataset. Therefore, we think the zero-shot setting may not match our setting in the paper.
>
> **General Response:** We have addressed your concerns about our paper. If you have more suggestions, please discuss them with us in the openreview system. We will try our best to address your further concerns and merge your comments into our revision as well!
>
> **Reference**
>
> [R1] Rui Huang and Yixuan Li. MOS: towards scaling out-of-distribution detection for large semantic space. In CVPR, 2021.

---

> > ### Author Response · Authors · 2022-11-19
> > **Update for ViT experiment**
> >
> > We follow the paper [R1] to reproduce MSP and Maxlogit with ViT on our imbalanced dataset ImageNet-LT-a8. The results are shown in below table. The last two rows are the results after applying our strateges. Compared with the first two rows, our strateges significantly improve the performance of the previous method.
> >
> > | Model |    Method   | iNaturalist |        |   SUN  |        | Places |        | Textures |        | Average |        |
> > |:-----:|:-----------:|:-----------:|:------:|:------:|:------:|:------:|:------:|:--------:|:------:|:-------:|:------:|
> > |       |             |    AUROC    |  FPR95 |  AUROC |  FPR95 |  AUROC |  FPR95 |   AUROC  |  FPR95 |  AUROC  |  FPR95 |
> > | ViT-B |     MSP     |    79.03%   | 87.90% | 75.86% | 86.60% | 73.03% | 87.30% |  77.70%  | 81.70% |  76.41% | 85.88% |
> > | ViT-B |   Maxlogit  |    60.47%   | 99.10% | 69.46% | 99.70% | 67.09% | 98.70% |  64.03%  | 91.00% |  65.27% | 97.13% |
> > | ViT-B |    RP+MSP   |    79.23%   | 87.10% | 75.88% | 86.60% | 73.08% | 87.10% |  77.78%  | 81.70% |  76.49% | 85.63% |
> > | ViT-B | RW+Maxlogit |    79.03%   | 87.90% | 75.86% | 86.60% | 73.03% | 87.30% |  77.70%  | 81.70% |  76.41% | 85.88% |
> >
> > [R1] Dan Hendrycks, Steven Basart, Mantas Mazeika, Andy Zou, Joe Kwon, Mohammadreza Mostajabi, Jacob Steinhardt, and Dawn Song. Scaling out-of-distribution detection for real-world settings. ICML, 2022

---

> ### Author Response · Authors · 2022-11-15
> **Response to your reviews (part1)**
>
> Thanks for your comments! We will answer them as follows:
>
> >Q1: Another branch of OOD detection is feature-space detection scores, such as KNN. I think a discussion of the potential extension of their method on feature space may be good since logit-space scores may not well fit some real-world applications, even since not all applications have a good assumption.
>
> **A1**: Feature-based methods, like KNN, need a training set to generate class prototypes, i.e., an average feature vector for each category. Under class-imbalanced situations, prototypes of tailed classes would be more unreliable than the majority due to the limitation of training samples. We think using ID-class-prior distribution to reweight features may be an effective way to solve the imbalanced problem in feature space. We will add some discussions in our revised paper and explore this problem in our future researches.
>
> >Q2: Fig 2 shows the performance of the balanced and imbalanced dataset, which, if an OOD set is close to the tailed categories of ID data, such as the OOD set highly close to the minor categories, and what's the performance of their method? I think reporting some per-category or hard case OOD is also interesting instead of reporting the full dataset score.
>
> **A2**: Thanks for your suggestion! We try to analyze tailed categories using a confusion matrix, and there are some cases below. $[\begin{smallmatrix} A & B & C & D \end{smallmatrix} ]$ is the result, where A represents the number of ID samples in the current class that are correctly classified as ID while C represents the number of ID samples in the current class that are misclassified as ID. D represents the number of OOD samples close to ID samples in the current class that are correctly classfied as OOD while B represents the number of OOD samples close to ID samples in the current class that are misclassfied as OOD.
>
> In Class #88 and #94, more OOD samples are correctly classified after applying our strategies, while the performance of Class #671 remains stable. In Class #671, more ID samples are correctly classified, while the performance of Class #94 of ID samples is sacrificed for more performance improvement of OOD samples.
>
>
> | Class id | GradNorm                                                   | RP+GradNorm |
> |----------|------------------------------------------------------------|-------------|
> | 88       | $[\begin{smallmatrix} 47 & 14  & 3 & 8 \end{smallmatrix} ]$ |   $[\begin{smallmatrix} 47 & 4  & 3 & 18 \end{smallmatrix} ]$         |
> | 94       |   $[\begin{smallmatrix} 33 & 106  & 17 & 182 \end{smallmatrix} ]$                                                         |    $[\begin{smallmatrix} 29 & 61  & 21 & 227 \end{smallmatrix} ]$         |
> | 671      |    $[\begin{smallmatrix} 35 & 2  & 15 & 1 \end{smallmatrix} ]$                                                        |     $[\begin{smallmatrix} 41 & 2  & 9 & 1 \end{smallmatrix} ]$        |
>
>
> >Q3: Assumption 1 seems good for natural images, while for those images with a huge domain gap between natural images, not the MINIST vs Fasion-MINIST one, how does assumption 1 fit for this case?
>
> **A3**: For those images with a huge domain gap between natural images, we think they are shown in Figure1 (b), where images have different styles. Our assumption1 still holds under this situation. Furthermore, we could consider a more complex situation: both $X_{in}$ and $X_{out}$ have various styles. We can divide them into two parts: shared style and different style. This situation is a mixture of Figure1 $($b$)$ and $($c$)$. Our assumption1 still holds under this situation.

---

> ### Author Response · Authors · 2022-11-17
> **Looking forward to your responses or further suggestions/comments!**
>
> Dear Reviewer dyEV,
>
> We have addressed your initial concerns regarding our paper. We are happy to discuss them with you in the openreview system if you feel that there still are some concerns/questions. We also welcome new suggestions/comments from you!
>
> Best regards,
>
> Authors of #3905

---

> ### Author Response · Authors · 2022-11-18
> **Looking forward to your responses or further suggestions/comments!**
>
> Dear Reviewer dyEV,
>
> We have addressed your initial concerns regarding our paper. We are happy to discuss them with you in the openreview system if you feel that there still are some concerns/questions. We also welcome new suggestions/comments from you!
>
> Best regards,
>
> Authors of #3905

---

> ### Author Response · Authors · 2022-11-19
> **Looking forward to your responses or further suggestions/comments!**
>
> Dear Reviewer dyEV,
>
> We have addressed your initial concerns regarding our paper. We are happy to discuss them with you in the openreview system if you feel that there still are some concerns/questions. We also welcome new suggestions/comments from you!
>
> Best regards,
>
> Authors of #3905

---

> ### Author Response · Authors · 2022-11-20
> **Looking forward to your responses or further suggestions/comments!**
>
> Dear Reviewer dyEV,
>
> We have addressed your initial concerns regarding our paper. We are happy to discuss them with you in the openreview system if you feel that there still are some concerns/questions. We also welcome new suggestions/comments from you!
>
> Best regards,
>
> Authors of #3905

---

> ### Author Response · Authors · 2022-11-21
> **Looking forward to your responses or further suggestions/comments!**
>
> Dear Reviewer dyEV,
>
> We have addressed your initial concerns regarding our paper. We are happy to discuss them with you in the openreview system if you feel that there still are some concerns/questions. We also welcome new suggestions/comments from you!
>
> Best regards,
>
> Authors of #3905

---

> ### Author Response · Authors · 2022-11-22
> **Looking forward to your responses or further suggestions/comments!**
>
> Dear Reviewer dyEV,
>
> We have addressed your initial concerns regarding our paper. We are happy to discuss them with you in the openreview system if you feel that there still are some concerns/questions. We also welcome new suggestions/comments from you!
>
> Best regards,
>
> Authors of #3905

---

> ### Author Response · Authors · 2022-11-23
> **Looking forward to your responses or further suggestions/comments!**
>
> Dear Reviewer dyEV,
>
> We have addressed your initial concerns regarding our paper. We are happy to discuss them with you in the openreview system if you feel that there still are some concerns/questions. We also welcome new suggestions/comments from you!
>
> Best regards,
>
> Authors of #3905

---

> ### Author Response · Authors · 2022-11-24
> **Looking forward to your responses or further suggestions/comments!**
>
> Dear Reviewer dyEV,
>
> We have addressed your initial concerns regarding our paper. We are happy to discuss them with you in the openreview system if you feel that there still are some concerns/questions. We also welcome new suggestions/comments from you!
>
> Best regards,
>
> Authors of #3905

---

> ### Author Response · Authors · 2022-11-25
> **Looking forward to your responses or further suggestions/comments!**
>
> Dear Reviewer dyEV,
>
> We have addressed your initial concerns regarding our paper. We are happy to discuss them with you in the openreview system if you feel that there still are some concerns/questions. We also welcome new suggestions/comments from you!
>
> Best regards,
>
> Authors of #3905

---

> ### Author Response · Authors · 2022-11-26
> **Looking forward to your responses or further suggestions/comments!**
>
> Dear Reviewer dyEV,
>
> We have addressed your initial concerns regarding our paper. We are happy to discuss them with you in the openreview system if you feel that there still are some concerns/questions. We also welcome new suggestions/comments from you!
>
> Best regards,
>
> Authors of #3905

---

### Author Response · Authors · 2022-11-18
**Revision Update and Summary of Changes**

We thank all the reviewers for their great effort and insightful reviews - they wil certainly help improve the clarity, presentation, and completeness of our paper! We are also glad that all reviewers are in agreement about the novelty insights, and good empirical performance of the paper.

We have uploaded a revised version of our paper following the suggestions/comments from all the reviewers. some major changes are:

* We add more discussion about feature-based methods. Thanks for dyEV's advice!
* We add aditional experimental analysis about tailed categories using a confusion matrix.  Thanks for dyEV's advice!
* We add Maxlogit as new baseline and apply our strategy on it. Thanks for dyEV's advice!
* We add aditional experimental results of appling our methods on balanced dataset. Thanks for uGHp and G6VC's advice!
* We add some discussion about two related work in Appendix B.7 and Appendix C. Thanks for vuMZ and G6VC's advice!
* We rewrite the symbol of the joint distribution. Thanks for vuMZ and G6VC's advice!
*  We add some discussion about the possibility for aligning the minimizer of the score function with the class priors in Appendix B.8. Thanks for G6VC's advice!
* We revise the Figure 4 to show only one of the similar metrics AUROC and FPR, and use the space to show all datasets. Thanks for G6VC's advice!

We thank the reviewers again and look forward to hearing their thoughts after reading the response!

---

### Comment · Reviewer_vuMZ · 2022-12-08
**Question to authors after reviewer discussion**

Dear authors,

After the virtual meeting between reviewers and AC for borderline papers, we still see two issues in the methodology section that we are reasonably sure are important inconsistencies.
Since there was disagreement in the discussion period, we want to give you the chance to show that those concerns are invalid.
We collaboratively check and formulate the relevant points below.

### The definition of $S_\text{RW+Method}$:

The definitions given in Equations (5) and (12) have different signs.
It is not clear to us which one of them is used in the experiments and if either of them can be correct.
From the more detailed discussion before (12) and the comments to the reviews it seems like the version in (12) is the intended one.

The issue with (12) as an OOD score can be seen in the following example:

Assume that we have inputs $x_1$ and $x_2$ where $softmax(f_\Theta(x_1)) = softmax(f_\Theta(x_2))$.\
Also assume that $S_\text{Method}(f_\Theta,x_1) > S_\text{Method}(f_\Theta,x_2) > 0$ .\
We further note that $\cos(softmax(f_\Theta(x_1)), P_{Y^{in}}) = \cos(softmax(f_\Theta(x_2)), P_{Y^{in}}) > 0$.\
Thus: $ - S_\text{Method}(f_\Theta,x_1) \cdot \cos(softmax(f_\Theta(x_1)), P_{Y^{in}})$
$< -S_\text{Method}(f_\Theta,x_2) \cdot \cos(softmax(f_\Theta(x_1)), P_{Y^{in}})$.

I.e: $S_\text{RW+Method}(f_\Theta,x_1) < S_\text{RW+Method}(f_\Theta,x_2)$

In other words, given equal softmax scores and positive $S_\text{Method}$, a *higher* score $S_\text{Method}(f_\Theta,x)$ leads to a *lower* score $S_\text{RW+Method}(f_\Theta,x)$. This consequence is counter-intuitive and does not seem to follow from the explanations in the paper.

On the other hand, if we consider omitting the minus sign as in (5), a similar issue arises if we assume the same  $S_\text{Method}(f_\Theta,x)$ on two inputs. As correctly stated in the paper, a *lower* $\cos(softmax(f_\Theta(x)), P_{Y^{in}})$ corresponds to a *higher* in-distribution score (e.g consider the simple case of a uniform $P_{Y^{in}}$), suggesting the *presence* of the minus sign is necessary.

The reason that leads to this issue might be that for two (not necessarily independent) random variables which are each positively correlated with the input being in-distribution, their product is not necessarily positively correlated with the input being in-distribution. The product not being positively correlated happens particularly easily if the variables have value ranges of different signs (as is the case with RW+ODIN for example).

### The energy OOD detection score:

The energy of a prediction is $-T·\log \sum_{i=1}^K e^{{f_{\Theta i}}(x)/T}$, as for example defined in Eq. (4) of [1].\
As [1] state, "Examples with higher energies are considered as OOD inputs and vice versa." and in their Eq. (7), they define the OOD detection score as the negative energy. This means $S_\text{Energy}(f_\Theta,x_1) = T·\log \sum_{i=1}^K e^{f_{\Theta_i}(x)/T}$ (*i.e no minus sign*).
The opposite is stated in this [author's comment](https://openreview.net/forum?id=72lzvXrKqqd&noteId=cwyFURUfuaG) without further explanation, so it would be interesting to see the reasoning here.

An important issue is that with the correct energy OOD score, the scores in Eq. (13) and (14) are inconsistent; this means the choice of (5) or (12) cannot fix the problem.
For a consistent theoretical foundation of the paper's proposed methodology, either both or neither need to have a minus sign, since $S_\text{Energy}(f_\Theta,x_1) = T·\log \sum_{i=1}^K e^{{f_{\Theta i}}(x)/T}$ has to be replaceable with $S_\text{ODIN}(f_\Theta ,x_1) = \max e^{f_{\Theta i}(x)/T} / \sum_{j=1}^K e^{f_{\Theta j}(x)/T}$.

[1] Weitang Liu, Xiaoyun Wang, John D. Owens, and Yixuan Li. Energy-based out-of-distribution detection. In NeurIPS, 2020.

---

> ### Author Response · Authors · 2022-12-10
> **Thanks for your internal discussion and further comments! (part2)**
>
>
> | Method |    RW    | iNaturalist |        |   SUN  |        | Places |        | Textures |        | Average |        |
> |:------:|:--------:|:-----------:|:------:|:------:|:------:|:------:|:------:|:--------:|:------:|:-------:|:------:|
> |        |          |    AUROC    |  FPR95 |  AUROC |  FPR95 |  AUROC |  FPR95 |   AUROC  |  FPR95 |  AUROC  |  FPR95 |
> |  ODIN  |    w/o   |    59.33%   | 98.59% | 72.57% | 91.68% | 70.84% | 90.83% |  42.00%  | 98.16% |  61.19% | 94.81% |
> |        |  -cosine |    89.18%   | 51.36% | 81.55% | 73.18% | 77.47% | 78.30% |  62.38%  | 87.57% |  77.65% | 72.60% |
> |        | 1-cosine |    89.07%   | 51.65% | 81.64% | 73.18% | 77.58% | 78.34% |  61.81%  | 87.61% |  77.52% | 72.69% |
> | Energy |    w/o   |    56.25%   | 98.95% | 73.60% | 91.55% | 71.33% | 90.38% |  42.68%  | 98.10% |  60.96% | 94.75% |
> |        |  -cosine |    90.56%   | 42.69% | 80.46% | 75.34% | 76.27% | 80.48% |  64.79%  | 85.90% |  78.02% | 71.10% |
> |        | 1-cosine |    81.11%   | 69.14% | 81.48% | 74.73% | 77.57% | 79.45% |  51.14%  | 90.69% |  72.83% | 78.50% |
>
> **It should also be noted that the counter examples do not apply for all implementations of the RW strategy.** For example, if we choose the $S_{Method}$ to be MSP, then $S_{Method}(f,x_1)$ = $S_{Method}(f,x_2)$ once $softmax(f(x_1))$ and $softmax(f(x_2))$ are the same. Namely, the assumptions made in the counter examples do not hold. Thus, we **cannot deny the contributions of the whole paper based on some implementations of one strategy (especially in the case where these implementations have very good performance)**. Our paper mainly wants to show the importance of ID class prior in OOD detection, which has been verified by implementations of **two** strategies.
>
> Finally, we want to express that **we cannot expect a perfect method** (i.e., a method can perfectly detect OOD data). Each method has its own drawbacks and cannot cover every perspective. **What we can do is to propose a method that can perform better and make some conceptual contributions to the field** (like "ID class prior is important in OOD detection" in our paper). The provided counter examples are very important when using the RW strategy (we very appriciate this!). A deep discussion regarding RW strategy (one of our proposed strategies) will be added into our main paper. We will demonstrate it clearly, especially the difference between negative-value RW and non-negative-value RW, which can make RW more solid and well-understood. We will discuss the limitations of RW and what we should be careful of when using RW in this deep discussion as well.
> >The reason that leads to this issue might be that for two (not necessarily independent) random variables which are each positively correlated with the input being in-distribution, their product is not necessarily positively correlated with the input being in-distribution. The product not being positively correlated happens particularly easily if the variables have value ranges of different signs (as is the case with RW+ODIN for example).
>
> **Response:**  Thanks for the comments! This possible explanation makes sense and would make our paper better. We will combine this into our deep discussion regarding RW strategy as well. This deep discussion mainly shows the potential failure cases regarding the RW strategy. The user of the RW strategy should be careful when combining -cosine or 1-cosine and $S_{Method}$.
>
> > The sign of Energy score
>
> **Response:**
> Thanks for your comments, it would make our paper better! There is a typo and the Energy score should be $S_{Energy}(f, x)=T \cdot \log \sum_k e^{f_i(x) / T}$, so Eq. (14) should have a minus here. This is consistent with the content before Eq. (14) and codes (function Energy_RW() on page 27) in Appendix D. The core idea of the RW strategy is correctly expressed in Eq. (12). Experiments about Energy are also based on the correct score. We are very sorry for the confusion caused by this typo! We will correct this typo in the latest version.

---

> ### Author Response · Authors · 2022-12-10
> **Thanks for your internal discussion and further comments! (part1)**
>
> >The definitions given in Equations (5) and (12) have different signs. It is not clear to us which one of them is used in the experiments and if either of them can be correct. From the more detailed discussion before (12) and the comments to the reviews it seems like the version in (12) is the intended one.
>
> **Response:** Thanks for your comments, and it would make our paper better! There is a typo, and Eq. (5) should be the same as Eq. (12). The codes in Appendix D also follow Eq. (12) and experiments are based on the correct fomula. We are very sorry for the confusion caused by this typo. We will correct this typo in the latest version.
>
>
> >The issue with (12) as an OOD score can be seen in the following example.
> >Assume that we have inputs $x_1$ and $x_2$ where softmax(f(x_1)) and softmax(f(x_2)).
> >Also assume that $S_{Method}(f,x_1)>S_{Method}(f,x_2)>0.$
> >We further note that $cos(softmax(f(x_1)),P_{Y_{in}})=cos(softmax(f(x_2)),P_{Y_{in}})>0$.
> >Thus, $-S_{Method}(f,x_1)\cdot cos(softmax(f(x_1)),P_{Y_{in}}) < -S_{Method}(f,x_2)\cdot cos(softmax(f(x_2)),P_{Y_{in}})$
> >I.e., $S_{RW+Method}(f,x_1)<S_{RW+Method}(f,x_2)$.
> >In other words, given equal softmax scores and positive $S_{Method}$, a higher score $S_{Method}$ leads to a lower score $S_{RW+Method}$. This consequence is counter-intuitive and does not seem to follow from the explanations in the paper.
> >On the other hand, if we consider omitting the minus sign as in (5), a similar issue arises if we assume the same $S_{Method}$ on two inputs. As correctly stated in the paper, a lower $cos(softmax(f(x)), P_{Y_{in}})$ corresponds to a higher in-distribution score (e.g consider the simple case of a uniform $P_{Y_{in}}$), suggesting the presence of the minus sign is necessary.
>
> **Response:** The proposed counter example is very interesting and important to analyze the limitation of our RW strategy. Thanks for providing this.
>
> However, the RW strategy mainly depends on the similarity between $softmax(f(x))$ and $P_{Y_{in}}$. If we meet the situation corresponding to the counter example, we actually cannot expect the performance gain obtained by RW strategy. In this situation, we are looking at data whose consine similarity with $P_{Y_{in}}$ is the same. Thus, the success/failure of RW strategy directly depends on the failure/success of $S_{Method}$. Namely, if $S_{Method}$'s results are correct, then $RW+S_{Method}$'s results are wrong. If S_Method's results are wrong, $RW+S_{Method}$'s results are correct.
> Even if we consider "1-cosine" as the reweighting function (1-cosine is non-negative), the success/failure of the RW strategy also directly depends on the success/failure of $S_{Method}$. Namely, if $S_{Method}$'s results are correct, then $RW+S_{Method}$'s results are correct. If $S_{Method}$'s results are wrong, $RW+S_{Method}$'s are wrong.
>
> Hence, the performance of $RW+S_{Method}$ on these counter examples is just decided by the performance of $S_{Method}$ on these counter examples, instead of the RW strategy. We cannot deny RW's contributions using these counter examples.
>
> The non-negative-value design (like 1-cosine) believes that $S_{Method}$ should be trusted on these counter examples. The negative-value design (like -cosine) believes that $S_{Method}$ should not be trusted on these counter examples. Because some of the previous S_Methods have an implicit assumption regarding the uniform ID-class prior distribution, it is hard to say whether we should trust $S_{Method}$ or not on these counter examples. For the other examples (except for these counter examples), based on our experiments, believing $S_{Method}$ might be a good option.
>
> Based on the above demonstration, we **cannot** say the negative-value-RW strategy does not make sense. It considers the other view to evaluate these counter examples only. Empirically, we count the number of these counter examples (i.e., examples whose softmax vector is the same with at least one example's softmax vector). We find that the number of such counter examples is very small compared to the number of data in the dataset (less than 0.16%). Thus, the main idea behind the RW strategy is still working. The OOD detection performance on these counter examples is not a core part of the RW strategy (since the similarity is the same, RW cannot provide more useful information). We also implement RW strategy based on 1-cosine, and the results are shown below. From this table, it is clear that RW works as well. The -cosine based RW+ODIN is similar with 1-cosine based RW+ODIN. The -cosine based RW+Energy is better than 1-cosine based RW+Energy. Thus, for Energy, its performance on the counter examples might not be trusted.

---

### Decision · Program_Chairs · 2023-01-20

**Decision:**

Reject

**Justification For Why Not Higher Score:**

Reviewers pointed out a critical conceptual flaw in the method that the authors didn't address.

**Justification For Why Not Lower Score:**

NA

**Metareview: Summary, Strengths And Weaknesses:**

This paper studies the problem of OOD detection in the non-uniform / long-tail class probability distribution setting and propose a re-weighting strategy to boost performance on a number of baselines.

The reviewers agreed that the paper tackles an important problem of practical significance, and that the empirical results look strong.

However, one reviewer pointed out a mistake in the equations and derivation of the procedure, leading to a critical issue in the conceptual and theoretical validity of the proposed re-weighting procedure. During discussions, multiple other reviewers also agreed with this. During a virtual meeting held, the reviewers were unable to resolve that theory was correct, despite a number of back-and-forths where the authors insisted that there were no mistakes. Based on this meeting, the reviewers carefully constructed a counter-example as well as detailed explanation for their concerns and presented it to the authors. The authors could not disprove what the reviewers presented. Therefore, the paper could use more clarification and thought before it is ready for publication.

**Summary Of Ac-Reviewer Meeting:**

I asked vuMZ and G6VC to meet since they were the reviewers who gave the highest and lowest rating, and both of them noted a possible inconsistency in the paper's math. Moreover, both of them were in early timezones (i.e. Indian and Central European time) so it was easier to just schedule a meeting with them.

vuMZ liked the paper and results, and wanted to give the authors a chance to address the possible error in the equations. During the meeting, we could not resolve these concerns. vuMZ and G6VC prepared a detailed description of their concerns and posted it to OpenReview.